# Unifying Masked Diffusion Models with Various Generation Orders and Beyond

Chunsan Hong [1]   Sanghyun Lee [1 2]   Jong Chul Ye [1]

## Abstract

Masked diffusion models (MDMs) are a potential alternative to autoregressive models (ARMs) for language generation, but generation quality depends critically on the generation order. Prior work either hard-codes an ordering (e.g., blockwise left-to-right) or learns an ordering policy for a pretrained MDM, which incurs extra cost and can yield suboptimal solutions due to the two-stage optimization. Motivated by this, we propose order-expressive masked diffusion model (OeMDM) for a broad class of diffusion generative processes with various generation orders, enabling the interpretation of MDM, ARM, and block diffusion in a single framework. Furthermore, building on OeMDM, we introduce learnable-order masked diffusion model (LoMDM), which jointly learns the generation ordering and diffusion backbone through a single objective from scratch, enabling the diffusion model to generate text in context-dependent ordering. Empirically, we confirm that LoMDM outperforms various discrete diffusion models across multiple language modeling benchmarks. Code is available at this link.

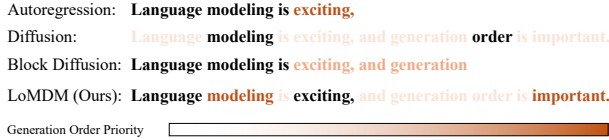

*Figure 1.* Conceptual illustration of learnable-order masked diffusion model (LoMDM) and other language models. Black text denotes already generated tokens, while the colored tokens indicate the generation candidates, with a lower color represents low generation order priority. In training time, LoMDM jointly learns **what to generate** and **where to generate next**, and in inference-time, LoMDM selects where to unmask next and predict a token.

## 1. Introduction

Diffusion models have recently been extended to language modeling in the form of masked diffusion models (MDMs) (Lou et al., 2024; Sahoo et al., 2024a), which are emerging as a potential alternative to autoregressive models (ARMs). MDMs define a forward process that replaces tokens in *random positions* with [MASK] and learn a corresponding reverse process. Training typically minimizes a negative evidence lower bound (NELBO) (Shi et al., 2024; Sahoo et al., 2024a) to fit the reverse dynamics.

One of the most actively studied directions in MDMs is the probabilistic modeling design with improved generation orderings beyond purely random orderings. However, classical MDMs are inherently *order-agnostic*: both the corruption process and the denoising objective are defined over randomly masked subsets, yielding a training signal that does not explicitly favor any particular generation ordering. Alternatively, structured orderings are incorporated through hand-designed schedulers or by redesigning the framework for each specific ordering. A representative example is block diffusion (BD3LM) (Arriola et al., 2025), which adopts a blockwise left-to-right ordering with random permutations within each block. GenMD4 (Shi et al., 2024) learns a token-dependent scheduler, differentiating the noising and denoising ratio by vocabulary. Both approaches require a modified modeling, which redefines the corruption/reverse processes to accommodate the imposed ordering.

Another line of work focuses on post-training the unmasking position sampler for improved orderings. Given an MDM trained with random ordering, various works (Hong et al., 2025; Peng et al., 2025a; Huang et al., 2025; Liu et al., 2026) learn a sampler that determines the unmasking order. However, such post-training approaches suffer from two key limitations: they incur additional training costs, and they may converge to suboptimal solutions due to the two-stage optimization between the MDM backbone and the unmasking ordering.

These observations raise two fundamental questions: 1) how to model various generation orders in the masked diffusion framework, and 2) how to learn generation order and the diffusion model jointly in a principled way.

[1] Graduate School of AI, KAIST, South Korea [2] KRAFTON, Seoul, South Korea. Correspondence to: Jong Chul Ye <jong.ye@kaist.ac.kr>.

*Proceedings of the 43rd International Conference on Machine Learning*, Seoul, South Korea. PMLR 306, 2026. Copyright 2026 by the author(s).

To address this, we introduce **order-expressive masked diffusion model (OeMDM)** that provides a lens for understanding diverse generation orderings in MDMs with a generalized NELBO. Recall that in classical MDM formulation (Lou et al., 2024; Sahoo et al., 2024a), a position-invariant noise schedule makes the forward masking rate uniform across positions, which in turn yields a uniform denoising rate in the reverse process and results in a primarily random ordering in both training and inference. In contrast, our OeMDM makes the generation order explicit by treating the *scheduler as a modeling component*, enabling the generation order to be considered in both training and inference. Specifically, the key contribution of OeMDM is to show that:

- Training an order-aware diffusion model requires sampling masked sequences from an *order-induced* corruption distribution, rather than using uniform random masking.

- The generalized NELBO for OeMDM can be decomposed into a reconstruction loss and a mismatch loss between corruption and unmasking order, yielding a principled mechanism for order-aware learning.

- This lens enables systematic analysis of different paradigms (e.g., ARMs, standard MDMs, BD3LM, and GenMD4) under a single formulation.

Additionally, our OeMDM framework naturally leads to **learnable-order masked diffusion model (LoMDM)**, which considers context-aware generation order. In contrast to the prior approaches that either hand-design the generation ordering or learn only restricted schedulers, LoMDM parameterizes a position-dependent scheduler conditioned on the full sequence so that we can learn full-context aware generation order and the diffusion model jointly by minimizing a single NELBO. This unified learning provides two benefits: (i) the learned scheduler is directly usable at generation time to decide where to unmask next, improving sample quality; and (ii) training focuses tokens with higher generation priority, so the scheduler simultaneously shapes a more order-aware training signal for the diffusion model. Empirically, LoMDM achieves lower test perplexity than a range of discrete diffusion baselines, including BD3LM and GenMD4.

## 2. Background

**Discrete diffusion.** Discrete diffusion models (Austin et al., 2021; Hoogeboom et al., 2021) have emerged as a competitive paradigm for *discrete* data, e.g., text. The main objective of discrete diffusion modeling is to model the discrete data distribution via a continuous-time diffusion process. There are broadly two types of forward corruption processes in discrete diffusion models: (1) uniform corruption (Lou et al.,

2024; Sahoo et al., 2025), which replaces tokens with uniformly random tokens, and (2) masking corruption (Sahoo et al., 2024a; Shi et al., 2024), which replaces tokens with a special [MASK] token. Among these, masked diffusion models (MDMs) have emerged as a leading class of discrete diffusion models for text generation (Sahoo et al., 2024a). Our work falls into this line of research and aims to improve MDMs in a continuous-time setting. Further details on related work are provided in Appendix A.

**Notation.** Let the vocabulary size be $V$ and define the token space $\mathcal{X} := \{\mathbf{v} \in [0,1]^{V+1} \mid \sum_{j=1}^{V+1} \mathbf{v}_j = 1, \mathbf{v}_{V+1} = 0\}$, where each word is represented by a one-hot vector. Let the mask token be $\mathbf{m} := \mathbf{e}_{V+1} \notin \mathcal{X}$. Let $\mathrm{Cat}(\cdot; \boldsymbol{\pi})$ denote the categorical distribution over $V+1$ classes with $\boldsymbol{\pi} \in \Delta^{V+1}$, the $(V+1)$-simplex. Let a sequence $\mathbf{x} = (\mathbf{x}^{(1)}, \ldots, \mathbf{x}^{(L)}) \in \mathcal{X}^L$, where $\mathbf{x}^{(i)}$ denotes the $i$-th token in a sequence. We denote a noised sequence by $\mathbf{z} \in \mathcal{Z}^L$ and, at time $t$, by $\mathbf{z}_t \in \mathcal{Z}_t^L$, where $\mathcal{Z} := \mathcal{Z}_t := \mathcal{X} \cup \{\mathbf{m}\}$ (we keep the subscript $t$ to emphasize time). For vectors $\mathbf{a}, \mathbf{b}$, $\langle \mathbf{a}, \mathbf{b} \rangle$ denotes the dot product.

**Masked diffusion language modeling (MDLM).** In continuous-time masked diffusion modeling, the most representative work is MDLM (Sahoo et al., 2024a). MDLM defines the forward corruption process using the absorbing mask strategy: once a token is masked, it remains masked throughout the remaining process. For the diffusion process, define the time interval as $t \in \mathcal{T} = [0, 1]$ where we corrupt the data from $t = 0$ (least noisy) to $t = 1$ (most noisy). Formally, the forward process at time $t$ is given as follows:

$$q(\mathbf{z}_t^{(i)} \mid \mathbf{x}) = q(\cdot \mid \mathbf{x}^{(i)}) = \mathrm{Cat}\left(\cdot; \alpha_t \mathbf{x}^{(i)} + (1 - \alpha_t)\mathbf{m}\right),$$

where the forward process gradually adds noise as $t$ grows. In this regard, the noise scheduler should satisfy $\alpha_0 \approx 1$, $\alpha_1 \approx 0$, $\alpha_t' < 0$, and is typically set to $\alpha_t = 1-t$. Following Sahoo et al. (2024a), discretize the time interval $\mathcal{T}$ with $T + 1$ steps, and define $s(\tau) = \tau/(T+1)$ and $t(\tau) = (\tau+1)/(T+1)$ such that generative distribution is divided into $T$ diffusion reverse steps ($\mathbf{z}_{t(T)} \to \cdots \to \mathbf{z}_{t(0)}$) and 1 reconstruction step ($\mathbf{z}_{t(0)} \to \mathbf{x}$). Then, the true reverse posterior can be derived as follows:

$$q(\mathbf{z}_s^{(i)} \mid \mathbf{z}_t, \mathbf{x}) = q(\mathbf{z}_s^{(i)} \mid \mathbf{z}_t^{(i)}, \mathbf{x})$$
$$= \begin{cases} \mathrm{Cat}(\mathbf{z}_s^{(i)}; \mathbf{z}_t^{(i)}), & \text{if } \mathbf{z}_t^{(i)} \neq \mathbf{m}, \\ \mathrm{Cat}\left(\mathbf{z}_s^{(i)}; \frac{(1-\alpha_s)\mathbf{m} + (\alpha_s - \alpha_t)\mathbf{x}^{(i)}}{1 - \alpha_t}\right), & \text{if } \mathbf{z}_t^{(i)} = \mathbf{m}. \end{cases}$$

where we drop $\tau$ in $s(\tau)$ and $t(\tau)$ for brevity. Hereafter, when the argument of a categorical distribution is clear from context, we omit it and write $\mathrm{Cat}(\boldsymbol{\pi})$ instead of $\mathrm{Cat}(\cdot; \boldsymbol{\pi})$. To mimic the true reverse posterior, Sahoo et al. (2024a) propose a parametrized reverse process as follows:

$$p_\theta(\mathbf{z}_s^{(i)} \mid \mathbf{z}_t) = q\Big(\mathbf{z}_s^{(i)} \mid \mathbf{z}_t, \mathbf{x} = \mathbf{x}_\theta(\mathbf{z}_t, t)\Big) \qquad (1)$$

$$= \begin{cases} \text{Cat}(\mathbf{z}_t^{(i)}), & \text{if } \mathbf{z}_t^{(i)} \neq \mathbf{m}, \\ \text{Cat}\left( \frac{(1-\alpha_s)\mathbf{m}+(\alpha_s-\alpha_t)\mathbf{x}_\theta^{(i)}(\mathbf{z}_t,t)}{1-\alpha_t} \right), & \text{if } \mathbf{z}_t^{(i)} = \mathbf{m}. \end{cases}$$

where $s < t$ and $\mathbf{x}_\theta^{(i)}(\mathbf{z}_t, t) = \text{Softmax}(\theta^{(i)}(\mathbf{z}_t, t))$ : $\mathcal{Z}_t^L \times \mathcal{T} \to \Delta^{V+1}$ predicts token $\mathbf{x}^{(i)}$. MDLM models the reverse process as a token-wise conditionally independent distribution, *i.e.*, $p_\theta(\mathbf{z}_s \mid \mathbf{z}_t) = \prod_{i=1}^L p_\theta(\mathbf{z}_s^{(i)} \mid \mathbf{z}_t)$. In the discrete-time case, the model distribution is expressed as $p_\theta(\mathbf{x}) = \sum_{\mathbf{z}_{t(0:T)}} p_\theta(\mathbf{z}_{t(T)}) p_\theta(\mathbf{x}|\mathbf{z}_{t(0)}) \prod_{\tau=1}^T p_\theta(\mathbf{z}_s|\mathbf{z}_t)$. Finally, with $T \to \infty$, the NELBO is given as follows:

$$\mathcal{L}_{\text{mdlm}} = \int_0^1 \mathbb{E}\Big[\sum_{i=1}^L \langle \mathbf{z}_t^{(i)}, \mathbf{m}\rangle \frac{\alpha_t'}{1-\alpha_t} \log\langle \mathbf{x}_\theta^{(i)}(\mathbf{z}_t,t), \mathbf{x}^{(i)}\rangle\Big] dt.$$

Note that we intentionally introduced MDLM with a multi-dimensional case for developing our method; yet, all the equations match those of MDLM. In the rest of the paper, we will denote the linear scheduler $\alpha_t = 1 - t$ as $\alpha_{\text{mdlm}}(t)$.

## 3. Unifying Various Orderings in MDMs

MDLM defines a single shared scheduler $\alpha_{\text{mdlm}}$ that is applied uniformly across all positions in a sequence. Therefore, in the model parametrized reverse process (Eq. 1), the denoising ratio is all equal as $(\alpha_s - \alpha_t)/(1-\alpha_t)$, so that the fundamental generation order is completely random. This implies that *modeling the generation order first requires rethinking the forward noise scheduler itself.*

Based on this observation, we introduce a generalized noise scheduler that allows different amounts of noise to be applied at different positions, thereby allowing different generation priorities across positions. Specifically, we provide a order-expressive masked diffusion model (OeMDM) defined under a generalized NELBO for various noise schedulers, and show how OeMDM can express various generation orders through a specific choice of scheduler.

### 3.1. Order-Expressive Masked Diffusion Model

We start by defining a scheduler function class that can represent diverse generation orderings in masked diffusion:

**Definition 3.1** (free-form scheduler function class)**.** For an arbitrary and fixed input domain $\mathcal{I}$ (e.g., $\mathcal{X}^L$, $\mathcal{Z}_t^L$, or $\emptyset$), we define the class of free-form schedulers with $\mathcal{I}$ as

$$\mathcal{F}[\mathcal{I}] := \Big\{\alpha : \mathcal{I} \times \mathcal{T} \to [0,1]^L \mid \forall u \in \mathcal{I}, \forall i \in [L] :$$
$$\alpha^{(i)}(u, \cdot) \in AC\big([0,1]\big) \cap C^1\big((0,1]\big), \ \alpha^{(i)}(u, 0) = 1,$$
$$\alpha^{(i)}(u, 1) = 0, \quad \partial_t \alpha^{(i)}(u, t) < 0, \forall t \in (0, 1]\Big\},$$

where $\mathcal{T} = [0, 1]$ is the time domain. Any scheduler in $\mathcal{F}[\mathcal{I}]$ is referred to as a free-form scheduler with input domain $\mathcal{I}$ and denoted by $\alpha_{\mathcal{F}[\mathcal{I}]}$[1].

The definition is motivated by two objectives: 1) The boundary and regularity conditions are essential in defining the diffusion process and deriving NELBO. 2) An arbitrary input domain allows the scheduler to be *context-aware*. By designing an appropriate scheduler, we can develop an MDM that decides where to unmask next, given the context.

**Forward process and true reverse process.** We define the forward process under $\alpha_{\mathcal{F}[\mathcal{I}]}$ as follows:

$$q_{\alpha_{\mathcal{F}}}(\mathbf{z}_t^{(i)} \mid \mathbf{x}) = \text{Cat}\Big( \alpha_{\mathcal{F}}^{(i)}(u, t)\mathbf{x}^{(i)} + (1 - \alpha_{\mathcal{F}}^{(i)}(u, t))\mathbf{m} \Big),$$

where $u \in \mathcal{I}$ and the input domain $\mathcal{I}$ specifies the information available to the scheduler in the forward process. In particular, the scheduler input for $\alpha_{\mathcal{F}}^{(i)}$ may be chosen from $u \in \big\{ \mathbf{x}, \ \mathbf{x}^{(i)}, \ \emptyset\big\}$, corresponding to a fully input-dependent scheduler, a coordinate-wise scheduler, or an unconditional (input-agnostic) scheduler, respectively. Then, the true reverse posterior is given by:

$$q_{\alpha_{\mathcal{F}}}(\mathbf{z}_s^{(i)} \mid \mathbf{z}_t, \mathbf{x}) = q_{\alpha_{\mathcal{F}}}(\mathbf{z}_s^{(i)} \mid \mathbf{z}_t^{(i)}, \mathbf{x})$$

$$= \begin{cases} \text{Cat}(\mathbf{z}_t^{(i)}), & \text{(I)}, \\ \text{Cat}\left( \frac{(1-\alpha_{\mathcal{F}}^{(i)}(u,s))\mathbf{m}+(\alpha_{\mathcal{F}}^{(i)}(u,s)-\alpha_{\mathcal{F}}^{(i)}(u,t))\mathbf{x}^{(i)}}{1-\alpha_{\mathcal{F}}^{(i)}(u,t)} \right), & \text{(II)}, \end{cases}$$

where (I) corresponds to $\mathbf{z}_t^{(i)} \neq \mathbf{m}$ and (II) to $\mathbf{z}_t^{(i)} = \mathbf{m}$. One can wonder how we can directly obtain $q_{\alpha_{\mathcal{F}}}(\mathbf{z}_s^{(i)}|\mathbf{z}_t, \mathbf{x})$ as above. This is because 1) we can just treat $\alpha_{\mathcal{F}}^{(i)}$ as fixed scheduler when $u$ is given, *i.e.* once $u$ is fixed, the map $r \mapsto \alpha_{\mathcal{F}}^{(i)}(u, r)$ is evaluated, and 2) the forward process $q_{\alpha_{\mathcal{F}}}(\mathbf{z}_t^{(i)} \mid \mathbf{x})$ is independent of $\mathbf{z}_t^{(j)}$ for all $j \neq i$. Therefore, $q_{\alpha_{\mathcal{F}}}(\mathbf{z}_s^{(i)} \mid \mathbf{z}_t, \mathbf{x})$ has the same structure as in MDLM, and its derivation can be done with the same process.

**Model-parametrized reverse process.** Unfortunately, the information available during parametrized reverse-time denoising differs from that of the true forward and reverse posterior. While $\alpha_{\mathcal{F}[\mathcal{I}]}$ may depend on inputs such as the full sequence $\mathbf{x}$ or a coordinate $\mathbf{x}^{(i)}$, the reverse process has access only to the current state $\mathbf{z}_t$ and the model outputs. Hence, the input domain $\mathcal{I}$ used in the forward construction generally does not match the information available in the denoising process. To reflect this mismatch, we introduce a separate input $\hat{u} \in \hat{\mathcal{I}}$ for model-parameterized reverse process, and let $\hat{\alpha}_{\mathcal{F}[\hat{\mathcal{I}}]} \in \mathcal{F}[\hat{\mathcal{I}}]$ as a scheduler for parametrized reverse process. In particular, $\hat{u}$ may belong to the following set: $\hat{u} \in \big\{ \mathbf{z}_t, \ \mathbf{z}_t^{(i)}, \mathbf{x}_\theta(\mathbf{z}_t, t), \ \emptyset\big\}$. Accordingly, we

---

[1]For brevity, we omit the explicit domain notation $[\mathcal{I}]$ and write $\alpha_{\mathcal{F}}$ when it is clear from context.

parametrize the reverse transition as follows:

$$
p_{\theta,\hat{\alpha}_{\mathcal{F}}}(\mathbf{z}_s^{(i)} \mid \mathbf{z}_t)
$$
$$
= \begin{cases} \mathrm{Cat}(\mathbf{z}_t^{(i)}), & \text{(I)}, \\ \mathrm{Cat}\left(\frac{(1-\hat{\alpha}_{\mathcal{F}}^{(i)}(\hat{u},s))\mathbf{m}+(\hat{\alpha}_{\mathcal{F}}^{(i)}(\hat{u},s)-\hat{\alpha}_{\mathcal{F}}^{(i)}(\hat{u},t))\mathbf{x}_\theta^{(i)}(\mathbf{z}_t,t)}{1-\hat{\alpha}_{\mathcal{F}}^{(i)}(\hat{u},t)}\right), & \\ & \text{(II)}, \end{cases}
$$

where $p_{\theta,\hat{\alpha}_{\mathcal{F}}}(\mathbf{z}_s \mid \mathbf{z}_t) = \prod_{i=1}^{L} p_{\theta,\hat{\alpha}_{\mathcal{F}}}(\mathbf{z}_s^{(i)} \mid \mathbf{z}_t)$. We follow SUBS parametrization (Sahoo et al., 2024a) of $\mathbf{x}_\theta$ (detailed in Appendix C). To avoid the potential confusion, note that $\hat{u}$ is restricted to the information available at reverse-time generation/denoising, *e.g.*, it can depend on the current state $\mathbf{z}_t$ but *not* on future/unknown states such as $\mathbf{z}_s$ for $s < t$. Equivalently, we view $\hat{\alpha}_{\mathcal{F}}^{(i)}(\hat{u},\cdot)$ as a time function *selected* (or "instantiated") by conditioning on $\hat{u}$ at time $t$; once $\hat{u}$ is fixed, the map $r \mapsto \hat{\alpha}_{\mathcal{F}}^{(i)}(\hat{u},r)$ is evaluated at $r = s$ or $r = t$ without introducing any additional dependence on $\mathbf{z}_s$. Note that this is a specific choice of parametrization that gives simple and intuitive NELBO; yet other parameterizations are also possible.

**Reinterpretation of reverse process with velocity.** While it is evident that the scheduler affects the generation order, how it does so in an intuitive, operational sense is somewhat unclear. To make this explicit, we define $A : \mathcal{I} \times (0,1] \to \mathbb{R}_+^L$ and $\hat{A} : \hat{\mathcal{I}} \times (0,1] \to \mathbb{R}_+^L$ as follows:

$$
A(u,t) := -\partial_t \alpha_{\mathcal{F}[\mathcal{I}]}(u,t) \oslash (1 - \alpha_{\mathcal{F}[\mathcal{I}]}(u,t)),
$$
$$
\hat{A}(\hat{u},t) := -\partial_t \hat{\alpha}_{\mathcal{F}[\hat{\mathcal{I}}]}(\hat{u},t) \oslash (1 - \hat{\alpha}_{\mathcal{F}[\hat{\mathcal{I}}]}(\hat{u},t)),
$$

where $\oslash$ refers to element-wise division. We will refer to $A$ as a ***velocity*** of discrete diffusion throughout the rest of the paper. Then, the true reverse posterior and denoising process in infinitesimal $dt$ can be rewritten as follows:

$$
q_{\alpha_{\mathcal{F}}}(\mathbf{z}_{t-dt}^{(i)} \mid \mathbf{z}_t, \mathbf{z}_t^{(i)} = \mathbf{m}, \mathbf{x})
$$
$$
= \mathrm{Cat}\left((1 - A^{(i)}(u,t)dt)\mathbf{m} + A^{(i)}(u,t)dt \cdot \mathbf{x}^{(i)}\right),
$$
$$
p_{\theta,\hat{\alpha}_{\mathcal{F}}}(\mathbf{z}_{t-dt}^{(i)} \mid \mathbf{z}_t, \mathbf{z}_t^{(i)} = \mathbf{m})
$$
$$
= \mathrm{Cat}\left((1 - \hat{A}^{(i)}(\hat{u},t)dt)\mathbf{m} + \hat{A}^{(i)}(\hat{u},t)dt \cdot \mathbf{x}_\theta^{(i)}(\mathbf{z}_t,t)\right),
$$

where $o(dt)$ is omitted in both terms. It is now clear how $\alpha_{\mathcal{F}}$ and $\hat{\alpha}_{\mathcal{F}}$ influence the generation order. By appropriately designing the scheduler, we can assign different velocities across indices, *i.e.*, we can determine which indices are prioritized and denoised earlier. Equipped with all ingredients above, we provide the corresponding NELBO of OeMDM:

**Proposition 3.2** (NELBO of OeMDM in continuous time). *Under SUBS parametrization, the NELBO of OeMDM in*

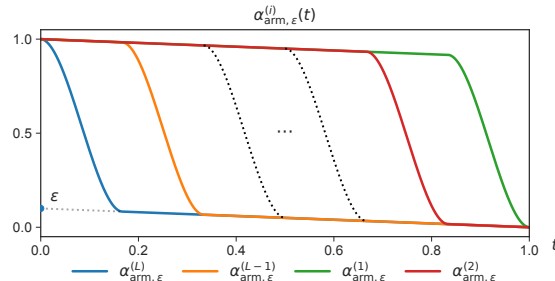

*Figure 2.* Illustration of $\alpha_{\mathrm{arm},\epsilon}(t)$ that makes OeMDM to generate in L2R order. The explicit function formulation is in Appendix D.1

*continuous time is given as follows:*

$$
-\log p_{\theta,\hat{\alpha}_{\mathcal{F}}}(\mathbf{x}) \leq \mathcal{L}_{\mathrm{OeMDM}}(\mathbf{x},\theta,\alpha_{\mathcal{F}},\hat{\alpha}_{\mathcal{F}})
$$
$$
= \int_0^1 \mathbb{E}_{q_{\alpha_{\mathcal{F}}}} \left[ \sum_{i=1}^L \langle \mathbf{z}_t^{(i)}, \mathbf{m} \rangle \left\{ \underbrace{-A^{(i)} \log\langle \mathbf{x}_\theta^{(i)}(\mathbf{z}_t,t), \mathbf{x}^{(i)} \rangle}_{\mathcal{L}_{\mathrm{main}}} \right. \right.
$$
$$
\left. \left. + \underbrace{A^{(i)}(\log A^{(i)} - \log \hat{A}^{(i)}) - (A^{(i)} - \hat{A}^{(i)})}_{\mathcal{L}_{\mathrm{velocity}}} \right\} \right] dt, \quad (2)
$$

*where the structure of $\mathcal{L}_{\mathrm{main}}$ is equal to $\mathcal{L}_{\mathrm{mdlm}}$ and $\mathcal{L}_{\mathrm{velocity}} \geq 0$ achieves 0 when $A = \hat{A}$.*

In $\mathcal{L}_{\mathrm{OeMDM}}$, since the forward process is defined by $q_{\alpha_{\mathcal{F}}}$, the expectations in the objective are taken with respect to $q_{\alpha_{\mathcal{F}}}$; in other words, training requires *order-aware* sampling of masked sequences. The NELBO decomposes into a diffusion-model reconstruction loss and a mismatch loss between the true posterior velocity and the parametrized reverse process velocity. Specifically, the reconstruction loss for each token ($\mathcal{L}_{\mathrm{main}}$ in Eq. 2) is weighted by its velocity. Furthermore, $\mathcal{L}_{\mathrm{velocity}}$ quantifies the gap between the unmasking order and the forward noise process. Consequently, when the corruption and unmasking orders are well aligned through $A$ and $\hat{A}$, yielding a small $\mathcal{L}_{\mathrm{velocity}}$, this encourages parameter updates of the diffusion model $\theta$ that focus $\mathcal{L}_{\mathrm{main}}$ on tokens with higher generation-order priority. The complete derivation and its finiteness condition can be found in Appendix C.

### 3.2. OeMDM Can Express Various Generation Orders

We provide here how we can understand the generation order and NELBO of MDLMs, ARMs, BD3LMs, and GenMD4 within our OeMDM. Trivially, if the free-form scheduler coincide with those of MDLM, i.e., $\alpha_{\mathcal{F}[\mathcal{I}]} := \alpha_{\mathrm{mdlm}}$ and $\hat{\alpha}_{\mathcal{F}[\hat{\mathcal{I}}]}(v,t) := \alpha_{\mathrm{mdlm}}$, then $\mathcal{L}_{\mathrm{velocity}} = 0$ such that $\mathcal{L}_{\mathrm{OeMDM}}(\theta,\alpha_{\mathrm{mdlm}},\alpha_{\mathrm{mdlm}}) = \mathcal{L}_{\mathrm{mdlm}}$. Furthermore, we show that OeMDM can also encompass ARMs:

**Proposition 3.3** (Autoregressive models as a special case of OeMDM). *If $\mathbf{x}_\theta$ is time-agnostic as typical ARMs, there*

*exists* $\alpha_{\mathrm{arm},\epsilon} \in \mathcal{F}[\emptyset]$ *that makes* $p_{\theta,\hat{\alpha}_{\mathcal{F}}}$ *becomes approximately equal to ARMs. Formally, the generative distribution induced by the reverse kernel* $p_{\theta,\alpha_{\mathrm{arm},\epsilon}}(\mathbf{z}_s|\mathbf{z}_t)$ *satisfies:*

$$p_{\theta,\alpha_{\mathrm{arm},\epsilon}}(\mathbf{x}) = \prod_{i=1}^{L} \langle \mathbf{x}_\theta^{(i)}(\mathbf{y}_i), \mathbf{x}^{(i)} \rangle + O(\epsilon),$$

*where* $\mathbf{y}_i = [\mathbf{x}^{(1:i-1)} : \mathbf{m}^{L-i+1}]$. *In continuous-time,*

$$\mathcal{L}_{\mathrm{OeMDM}}(\mathbf{x}, \theta, \alpha_{\mathrm{arm},\epsilon}, \alpha_{\mathrm{arm},\epsilon})$$
$$= -\log \prod_{i=1}^{L} \langle \mathbf{x}_\theta^{(i)}(\mathbf{y}_i), \mathbf{x}^{(i)} \rangle + O(\epsilon),$$

*such that OeMDM converges to ARM closely as* $\epsilon \to 0+$.

*proof sketch.* A specific form of the scheduler $\alpha_{\mathrm{arm},\epsilon}$ is given in Definition D.1, and its conceptual illustration is shown in Figure 2. Intuitively, OeMDM approximating ARM is trivial since such a scheduler would give a masked sequence with right-most filled masks and yield reconstruction loss weighted on the first mask in training time, and generation will likely occur in L2R order in inference-time. We rigorously show that this is true, and the NELBO of OeMDM becomes the negative log-likelihood of ARM. □

Note that this result can also be extended to auto-regressive modeling of any fixed ordering (see Corollary D.4). For BD3LM as well, by designing the scheduler, we can arrive at the same conclusion (Appendix D.2). Furthermore, we show that GenMD4, which considers a vocabulary-wise forward scheduler, *i.e.*, $\alpha_{\mathcal{F}[\mathcal{X}]}$, exactly falls into our OeMDM framework with the same NELBO in Appendix D.3.

## 4. MDMs with Learnable Order

Through OeMDM, we have observed how the generation order shapes both training via the NELBO and inference via the parameterized reverse process. However, existing schedulers do not fully consider the information given in the denoising/reverse processes, *i.e.*, $\mathcal{I} = \emptyset$ or $\mathcal{I} = \mathcal{X}$. This might be suboptimal since there might exist a better context-aware generation order. In this section, we provide learnable-order masked diffusion model (LoMDM), which learns *where to unmask next* and *what to generate next*.

### 4.1. NELBO for LoMDM

In LoMDM, we set $\mathcal{I} = \mathcal{X}^L$ and $\hat{\mathcal{I}} = \mathcal{Z}_t^L$ to fully leverage the information given in the forward and parametrized reverse process. That is, the forward process decides where to corrupt first, given the full sentence, and the parametrized reverse process determines where to generate first, given the masked sentence. With $\alpha_{\mathcal{F}[\mathcal{X}^L]}(\mathbf{x}, t) = \alpha_\phi(\mathbf{x}, t)$,

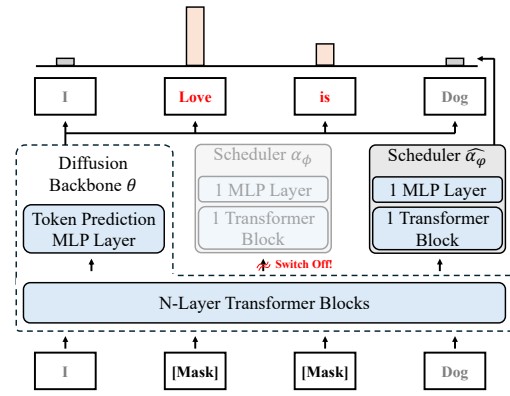

*Figure 3.* Model structure of LoMDM. We view backbone of diffusion model $\theta$ as a feature extractor of $\mathbf{z}_t$ or $\mathbf{x}$, and train $\theta, \alpha_\phi$, and $\hat{\alpha}_\psi$ jointly. Depending on the input type, final layers are switched off or on. For example in the above figure, the input is $\mathbf{z}_t$ so the final diffusion MLP layer and $\hat{\alpha}_\psi$ is activated. Meanwhile, if input was $\mathbf{x}$, only $\alpha_\phi$ would be activated. We detach the gradient of $\alpha_\phi$ and $\hat{\alpha}_\psi$ from flowing to the diffusion backbone (N-Layer transformer blocks in figure.)

$$A(\mathbf{x}, t) = A_\phi(\mathbf{x}, t), \text{ and } \hat{A}(\mathbf{z}_t, t) = \hat{A}_\psi(\mathbf{z}_t, t),$$

$$\mathcal{L}_{\mathrm{LoMDM}} = \mathcal{L}_{\mathrm{OeMDM}}(\mathbf{x}, \theta, \alpha_\phi, \hat{\alpha}_\psi) =$$

$$\int_{t=0}^{1} \mathbb{E}_{q_{\alpha_\phi}} \left[ \sum_{i=1}^{L} \langle \mathbf{z}_t^{(i)}, \mathbf{m} \rangle \left\{ \underbrace{-A_\phi^{(i)} \log \langle \mathbf{x}_\theta^{(i)}(\mathbf{z}_t, t), \mathbf{x}^{(i)} \rangle}_{\mathcal{L}_{\mathrm{main}}} \right. \right.$$

$$\left. \left. + \underbrace{A_\phi^{(i)}(\log A_\phi^{(i)} - \log \hat{A}_\psi^{(i)}) - (A_\phi^{(i)} - \hat{A}_\psi^{(i)})}_{\mathcal{L}_{\mathrm{velocity}}} \right\} \right] dt. \quad (3)$$

Note that the main difference from Eq. 2 is that the forward and reverse velocity ($A_\phi^{(i)}$ and $\hat{A}_\psi^{(i)}$) are now function of the neural networks $\phi$ and $\psi$ that should be learned by optimizing NELBO. More specifically, the interpretation of NELBO for LoMDM can be summarized as follows:

- To minimize $\mathcal{L}_{\mathrm{main}}$, diffusion model $\theta$ tries to reconstruct data, and it concentrates on the tokens with higher velocity. The velocity $A_\phi$ is trained to have a higher value for the token that the diffusion model predicts correctly.

- To minimize $\mathcal{L}_{\mathrm{velocity}}$, the velocity $A_\phi$ and velocity $\hat{A}_\psi$ is trained to have equal values. That is, the true reverse posterior velocity and the parametrized reverse process velocity are trained to become identical.

- Combining the above two results, $A_\phi$ is trained on two objectives as follows: 1) To *guide* diffusion backbone $\theta$ to the easiest path, 2) while keeping it *tractable* for $\hat{A}_\psi$.

### 4.2. Model Structure

However, introducing entirely new networks $\phi$ and $\psi$ for the scheduler would require a large number of additional

parameters, making training inefficient in both speed and memory, and would also make optimization more unstable. Therefore, inspired by Hong et al. (2025), we view the Transformer layers of the diffusion network $\theta$ as a fixed feature extractor for the scheduler networks $\phi$ and $\psi$. We utilize the time-agnostic diffusion network following prior works including MDLM (Sahoo et al., 2024a; Xie et al., 2025), given by $\mathbf{x}_\theta(\mathbf{z}_t, t) = \text{Softmax}(\theta_{\text{MLP}}(\theta_{\text{TF}}(\mathbf{z}_t)))$. We then use $f(\cdot) = \text{Sgd}(\theta_{\text{TF}}(\cdot))$ as the feature extractor, where Sgd refers to stop-gradient. We then parameterize (i) the forward scheduler $\alpha_\phi(\mathbf{x}, t)$, (ii) use the analytic relation to obtain $A_\phi$ from $\alpha_\phi$, and (iii) the reverse-time scheduler and velocity $\hat{\alpha}_\psi(\mathbf{z}_t, t), \hat{A}_\psi(\mathbf{z}_t, t)$ to have same functional form:

$$\alpha_\phi^{(i)}(\mathbf{x}, t) := 1 - t^{c_1 + c_2 \cdot [\text{NormSig}(g_\phi(f(\mathbf{x})))]_i} \tag{4}$$

$$A_\phi^{(i)}(\mathbf{x}, t) = \frac{c_1 + c_2 \cdot [\text{NormSig}(g_\phi(f(\mathbf{x})))]_i}{t}, \tag{5}$$

$$\hat{\alpha}_\psi^{(i)}(\mathbf{z}_t, t) := 1 - t^{c_1 + c_2 \cdot [\text{NormSig}(g_\psi(f(\mathbf{z}_t)))]_i} \tag{6}$$

$$\hat{A}_\psi^{(i)}(\mathbf{z}_t, t) = \frac{c_1 + c_2 \cdot [\text{NormSig}(g_\psi(f(\mathbf{z}_t)))]_i}{t}. \tag{7}$$

In particular, we choose $c_1 > c_2$ such that both $\alpha_\phi$ and $\alpha_\psi$ satisfy the condition of free-form scheduler class, and NELBO is always finite by Proposition C.3. We denote NormSig as the normalized Sigmoid defined on the overall sequence, *i.e.*, $[\text{NormSig}(\mathbf{v})]_i = \sigma(\mathbf{v}_i) - \sum_{j=1}^{L} \sigma(\mathbf{v}_j)/L$ for vector $\mathbf{v} \in \mathbb{R}^L$. As shown in Figure 3, each $\phi$ and $\psi$ is composed of 1 transformer layer followed by 1 MLP layer. This simple parametrization helps to learn generation order effectively, since $g_\phi$ and $g_\psi$ can be directly optimized toward reconstruction loss. Furthermore, normalization helps to forces overall velocity to be regularized, such that $g_\psi$ and $g_\phi$ just modulates relative generation order priority. See further detail of LoMDM parametrization in Appendix F.1,

### 4.3. Training Algorithm

In $\mathcal{L}_{\text{LoMDM}}$ (Eq. 3), we can see that $\alpha_\phi$ is included in the expectation term. This means that we should sample $\mathbf{z}_t \sim q_{\alpha_\phi}(\cdot \mid \mathbf{x})$, and it also requires gradient descent to be performed. In this section, we provide how we handle this.

**Gradient of $\phi$.** The naive gradient estimator for $\phi$ is:

$$\nabla_\phi \mathcal{L}_{\text{LoMDM}} = \mathbb{E}_{t \sim \text{Uniform}([0,1])} \big[ \mathbb{E}_{q_{\alpha_\phi}}[\nabla_\phi(\mathcal{L}_{\text{main}} + \mathcal{L}_{\text{velocity}})$$
$$+ \mathbb{E}_{q_{\alpha_\phi}}[\nabla_\phi \log q_{\alpha_\phi} \cdot (\mathcal{L}_{\text{main}} + \mathcal{L}_{\text{velocity}})]\big].$$

However, the estimator $\mathbb{E}_{q_{\alpha_\phi}}[\nabla_\phi \log q_{\alpha_\phi} \cdot (\mathcal{L}_{\text{main}} + \mathcal{L}_{\text{velocity}})]$ has high-variance in reinforcement learning perspective (Shi et al., 2024). In this regard, we sample $\mathbf{z}_t^1, \mathbf{z}_t^2 \sim q_{\alpha_\phi}(\cdot|\mathbf{x})$ independently and utilize a non-biased low variance estimator (Kool et al., 2019) as follows:

$$\mathcal{L}_{\text{rloo}} = \frac{1}{2} \log \frac{q_{\alpha_\phi}(\mathbf{z}_t^1|\mathbf{x})}{q_{\alpha_\phi}(\mathbf{z}_t^2|\mathbf{x})} \Big( \text{Sgd}(\mathcal{L}_{\mathbf{z}_t^1}) - \text{Sgd}(\mathcal{L}_{\mathbf{z}_t^2}) \Big), \tag{8}$$

$$\nabla_\phi \mathcal{L}_{\text{LoMDM}} = \mathbb{E}_t[\mathbb{E}_{\mathbf{z}_t^1, \mathbf{z}_t^2 \sim q_{\alpha_\phi}}[\nabla_\phi(\frac{1}{2}(\mathcal{L}_{\mathbf{z}_t^1} + \mathcal{L}_{\mathbf{z}_t^2}) + \mathcal{L}_{\text{rloo}})]],$$

---

**Algorithm 1** Training algorithm of LoMDM

1: **while** Until converge **do**
2:     Sample $\mathbf{x} \sim p(\mathbf{x})$ and $t \sim \text{Uniform}([0, 1])$
3:     $f \leftarrow \text{Sgd}(\theta_{\text{TF}}(\mathbf{x}))$
4:     $\alpha_\phi(\mathbf{x}, t), \quad A_\phi(\mathbf{x}, t) \leftarrow$ Eq. 4 and 5 using $f$
5:     Sample $\mathbf{z}_t^1, \mathbf{z}_t^2 \sim q_{\alpha_\phi}(\mathbf{z}_t|\mathbf{x})$
6:     **for** $i \in \{0, 1\}$ **do**
7:        $\mathbf{x}_\theta^i \leftarrow \text{Softmax}(\theta_{\text{MLP}}(\theta_{\text{TF}}(\mathbf{z}_t^i)))$
8:                 where $\theta_{\text{TF}}(\cdot)$ is cached
9:        $f_{\mathbf{z}^i} \leftarrow \text{Sgd}(\theta_{\text{TF}}(\mathbf{z}_t^i))$
10:       $\hat{A}_\psi(\mathbf{z}_t^i, t) \leftarrow$ Eq. 7 using $f_{\mathbf{z}^i}$
11:       $\mathcal{L}_{\mathbf{z}_t^i} \leftarrow$ Eq. 3 using $A_\phi, \hat{A}_\psi, \mathbf{x}_\theta$ and $\mathbf{x}$
12:     **end for**
13:     $\mathcal{L}_{\text{rloo}} \leftarrow$ Eq. 8
14:     $\widehat{\mathcal{L}}_{\text{LoMDM}} \leftarrow \frac{1}{2}(\mathcal{L}_{\mathbf{z}_t^1} + \mathcal{L}_{\mathbf{z}_t^2}) + \mathcal{L}_{\text{rloo}}$
15:     Perform gradient descent on $\widehat{\mathcal{L}}_{\text{LoMDM}}$ for $\{\theta, \phi, \psi\}$ simultaneously
16: **end while**

---

where $\mathcal{L}_{\mathbf{z}_t^i}$ refers to $\mathcal{L}_{\text{main}} + \mathcal{L}_{\text{velocity}}$ for $\mathbf{z}_t^i$. Here, $i$ is different from $(i)$ which indicates $i$-th token, *i.e.* $\mathbf{z}_t^i$ itself is full sequence such that $\dim(\mathbf{z}_t^i) = (V + 1) \times L$. Further details are given in Appendix E.1, and note that $\mathcal{L}_{\text{rloo}}$ is invariant to gradient of $\phi$ and $\theta$.

**Training algorithm.** Combining all above, we provide Algorithm 1, which includes sampling $\mathbf{z}_t^1, \mathbf{z}_t^2 \sim q_{\alpha_\phi}(\cdot|\mathbf{x})$ and directly optimize the single NELBO for three parameters. We have intentionally omitted the batch process to intensify readability, yet we sample $t$ number of $B//2$, and learn $\mathbf{z}_t^1, \mathbf{z}_t^2$ for each $t$, so a total of $B$ samples can be learn within one batch. Our training algorithm requires 1 more forward pass of $\theta$, and requires 2 more forward/backward passes of $\phi, \psi$ (composed of 1 transformer/MLP layer) than conventional MDLM, so that the number of tokens seen per second is slightly lower than that of MDLM. However, we observed that our LoMDM substantially outperforms MDLM within the same trained hours, and we report these observations in following section.

## 5. Experimental results

### 5.1. Main Results

**Experimental settings.** We evaluate LoMDM following the widely adopted experimental settings in continuous-time discrete diffusion language modeling (Sahoo et al., 2025; Arriola et al., 2025). We train LoMDM on three datasets, including One Billion Words dataset (LM1B) (Chelba et al., 2014) with/without sentence packing and OpenWebText (OWT) (Gokaslan et al., 2019). We utilize three metrics: test perplexity, zero-shot test perplexity, and generative perplexity. Every discrete diffusion model, including ours are

*Table 1.* Test perplexities (PPL; ↓) on LM1B and OpenWebText. Best diffusion value is bolded. ¶Denotes the dataset didn't incorporate sentence packing. For diffusion models, we report the bound on the likelihood. In GenMD4, ‡ denotes our trained model due to the absence of experiments. All diffusion models were trained with a batch size of 512, whereas reported PPL in OWT of GenMD4 (Shi et al., 2024) was that of model trained with a batch size of 1024, so that we marked it as ≥. Otherwise, reported values are imported from Sahoo et al. (2025). $L'$ in BD3LM refers to length of each block.

| | | LM1B¶ | LM1B | OWT |
|---|---|---|---|---|
| *Autoregressive* | | | | |
| Transformer | | 22.3 | 22.8[†] | 17.5 |
| *Diffusion (Uniform-state / Gaussian)* | | | | |
| D3PM Uniform (Austin et al., 2021) | | - | 137.9 | - |
| Diffusion-LM (Li et al., 2022) | | - | 118.6 | - |
| SEDD Uniform (Lou et al., 2024) | | 40.3 | - | 29.7 |
| UDLM (Schiff et al., 2025) | | 31.3 | 36.7 | 27.4 |
| Duo (Sahoo et al., 2025) | | 29.9 | 33.7 | 25.2 |
| *Diffusion (Absorbing state)* | | | | |
| BERT-Mouth (Wang & Cho, 2019) | | - | 142.9 | - |
| D3PM Absorb (Austin et al., 2021) | | - | 76.9 | - |
| DiffusionBert (He et al., 2023) | | - | 63.8 | - |
| SEDD Absorb (Lou et al., 2024) | | 32.7 | - | 24.1 |
| MDLM (Sahoo et al., 2024a) | | 27.0 | 31.8 | 23.2 |
| *Autoregressive + Diffusion* | | | | |
| BD3LM (Arriola et al., 2025) | $L'$=16 | - | 30.6 | 22.3 |
| | $L'$=8 | - | 29.8 | 21.7 |
| | $L'$=4 | - | 28.2 | 20.7 |
| *Diffusion (Absorbing state + Learnable scheduler)* | | | | |
| GenMD4 (Shi et al., 2024) | | 26.9[‡] | 30.0[‡] | 21.8[≥] |
| **LoMDM (Ours)** | | **25.4** | **27.2** | **20.4** |

trained for 1M steps with a batch size of 512 unless specified. Note that we actually sample 256 texts for each batch since LoMDM utilizes a two-sample estimator, so the experimental setting is fair. Finally, we set $c_1 = 0.7, c_2 = 0.65$ for LoMDM for every dataset. Further details are provided in Appendix G.1, and ablation study on $c_1, c_2$ is given in Appendix F.2

**Likelihood evaluation.** On LM1B and OWT (Table 1), our LoMDM outperforms every other discrete diffusion models with large margin. For LM1B with sentence packing and OWT, LoMDM achieves PPL values that are more than 3 points lower than those of MDLM. Furthermore, we observe that LoMDM achieves lower PPL across all benchmarks than BD3LM even when BD3LM uses a block size of $L' = 4$, which injects an almost autoregressive L2R bias. Finally, for GenMD4, which also learns the scheduler like our method, LoMDM outperforms in every benchmark. Notably, in OWT, even the batch size is set to 1024 for GenMD4 meaning it sees twice as many training samples as LoMDM, yet our LoMDM achieves much lower PPL.

**Zero-shot likelihood evaluation.** We measure the zero-shot generalization of the models trained on OWT by evaluating their PPL on 7 other datasets. Following Sahoo et al. (2024a; 2025), our zero-shot datasets include the validation splits

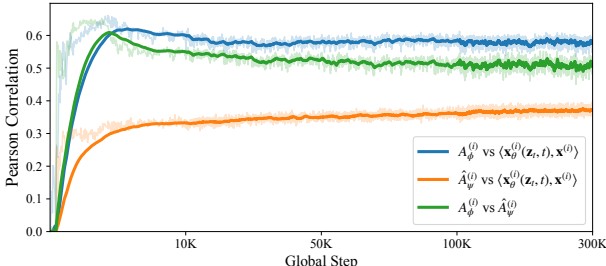

*Figure 4.* Pearson correlation per training step for LoMDM trained on OWT. We report correlations among $A_\phi^{(i)}(\mathbf{x}, t)$, $\hat{A}_\psi^{(i)}(\mathbf{z}_t, t)$, and $\langle \mathbf{x}_\theta^{(i)}(\mathbf{z}_t, t), \mathbf{x}^{(i)} \rangle$. When measuring correlation with $\langle \mathbf{x}_\theta^{(i)}(\mathbf{z}_t, t), \mathbf{x}^{(i)} \rangle$, we compute it only over masked positions in $\mathbf{z}_t$, since $\mathbf{x}_\theta^{(i)}(\mathbf{z}_t, t)$ is zero at unmasked positions.

of Penn Tree Bank (PTB; Marcus et al. (1993)), WikiText (Merity et al., 2017), LM1B, Lambada (Paperno et al., 2016), AG News (Zhang et al., 2015), and Scientific papers from ArXiv and Pubmed (Cohan et al., 2018). We observe that LoMDM achieves a lower PPL than MDLM on 7/7, notably, outperforming an autoregressive transformer on 4/7 datasets, and state-of-the-art among all discrete diffusion models on 6/7 datasets. In addition, Shi et al. (2024) report that their GenMD4 model trained on OWT exhibited a bias toward certain words, which in turn degraded its zero-shot PPL; for this reason, zero-shot PPL metric was not reported.

**Generative perplexity.** Finally, we test the effectiveness of LoMDM for the quality of generated texts. We employed ancestral sampling (Sahoo et al., 2024a) with reverse transition $p_{\theta, \hat{\alpha}_\psi}(\mathbf{z}_s | \mathbf{z}_t)$. As shown in Table 3, our LoMDM outperforms the MDLM baseline with a large margin, which indicates our velocity $\hat{A}_\psi$ actually improves the text generation ability of MDM. Furthermore, LoMDM achieves lower generative PPL across various NFE settings, indicating that it improves text generation quality while retaining the fast generation enabled by parallel decoding. We further conduct an ablation study to isolate the effect of the learned scheduler at inference time. While keeping the training setup fixed at $(c_1, c_2) = (0.7, 0.65)$ for all models, we disable the scheduler effect during generation by setting $c_2 = 0$ (Table 3, †). This consistently degrades the generative PPL compared to the matched train–inference setting, indicating that the learned velocity $\hat{A}_\psi$ provides a beneficial generation path rather than merely acting as a training-time regularizer.

### 5.2. Training Dynamics of LoMDM

**Correlation between $\theta$, $\phi$, $\psi$.** To understand how the learned scheduler interacts with the diffusion backbone, we track Pearson correlations between the forward velocity $A_\phi^{(i)}(\mathbf{x}, t)$, the reverse-time velocity predicted by $\psi$, $\hat{A}_\psi^{(i)}(\mathbf{z}_t, t)$, and the backbone's token-level reconstruction

*Table 2.* Zero-shot perplexities (↓) of models trained for 1M steps on OpenWebText. All perplexities for diffusion models are upper bounds. [†] Taken from Arriola et al. (2025). Otherwise, reported values are imported from Sahoo et al. (2025). Best diffusion values are **bolded** and diffusion values better than AR are underlined.

|  | PTB | Wikitext | LM1B | Lambada | AG News | Pubmed | Arxiv |
|---|---|---|---|---|---|---|---|
| *Autoregressive* |  |  |  |  |  |  |  |
| Transformer | 82.05 | 25.75 | 51.25 | 51.28 | 52.09 | 49.01 | 41.73 |
| *Diffusion (Uniform-state / Gaussian)* |  |  |  |  |  |  |  |
| SEDD Uniform | 105.51 | 41.10 | 82.62 | 57.29 | 82.64 | 55.89 | 50.86 |
| Plaid | 142.60 | 50.86 | 91.12 | 57.28 | - | - | - |
| UDLM | 112.82 | 39.42 | 77.59 | 53.57 | 80.96 | 50.98 | 44.08 |
| Duo | 89.35 | 33.57 | 73.86 | 49.78 | 67.81 | 44.48 | 40.39 |
| *Diffusion (Absorbing state)* |  |  |  |  |  |  |  |
| SEDD Absorb | 100.09 | 34.28 | 68.20 | 49.86 | 62.09 | 44.53 | 38.48 |
| D3PM Absorb | 200.82 | 50.86 | 138.92 | 93.47 | - | - | - |
| MDLM | 95.26 | 32.83 | 67.01 | 47.52 | 61.15 | 41.89 | 37.37 |
| *Autoregressive + Diffusion* |  |  |  |  |  |  |  |
| BD3LM[†] ($L' = 4$) | 96.81 | 31.31 | **60.88** | 50.03 | 61.67 | 42.52 | 39.20 |
| *Diffusion (Absorbing state + Learnable scheduler)* |  |  |  |  |  |  |  |
| **LoMDM (Ours)** | **80.40** | **27.82** | 61.19 | **36.32** | **53.53** | **37.73** | **32.88** |

*Table 3.* Generative PPL (↓) of models trained on OWT, computed by a pre-trained GPT-2 Large on 256 generated samples with length of 1024. LoMDM uses matched train–inference scheduling at test time is marked as bold. † denotes *no-scheduler* ablations where the scheduler effect is disabled at inference by setting $c_2=0$.

| #NFE | 128 | 256 | 512 | 1024 |
|---|---|---|---|---|
| MDLM | 116.71 | 79.43 | 55.50 | 42.56 |
| **LoMDM** ($c_1=0.7$, $c_2=0.65$) | **92.78** | **73.98** | **48.29** | **38.87** |
| *Ablation study* |  |  |  |  |
| LoMDM† ($c_1=0.7$, $c_2=0$) | 107.95 | 78.05 | 59.34 | 44.42 |
| LoMDM† ($c_1=1$, $c_2=0$) | 122.48 | 83.68 | 59.88 | 48.30 |

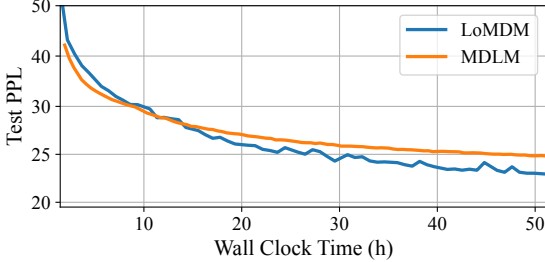

*Figure 5.* Test PPL per wall-clock-time during training on OWT. We truncate the curves at the point where LoMDM matches the 1M-step MDLM performance (PPL = 23.0). At this cutoff, MDLM had reached PPL = 24.9 with ∼0.30M steps, while our method had reached PPL = 23.0 with ∼0.18M steps.

confidence toward ground truth $\langle \mathbf{x}_\theta^{(i)}(\mathbf{z}_t, t), \mathbf{x}^{(i)} \rangle$. Figure 4 shows that $A_\phi^{(i)}(\mathbf{x}, t)$ quickly becomes positively correlated with reconstruction confidence and remains stable throughout training. This suggests that the learned $\phi$ assigns higher denoising emphasis to tokens that the backbone $\theta$ can reconstruct more reliably at the current time $t$. We also observe a positive correlation between $\hat{A}_\psi^{(i)}(\mathbf{z}_t, t)$ and reconstruction confidence, indicating that $\psi$ learns to infer which masked tokens are currently easier to predict from the partially observed context $\mathbf{z}_t$. Finally, the strong correlation between $A_\phi$ and $\hat{A}_\psi$ supports that the velocity-matching term $\mathcal{L}_{\text{velocity}}$ effectively aligns the forward and reverse velocities.

**LoMDM is a fast and efficient learner.** Figure 5 reports test PPL on OWT with wall-clock time, where the curves are truncated when LoMDM reaches the 1M-step MDLM reference performance (PPL=23.0). Notably, LoMDM attains this target after only 0.18M steps, compared to 1M steps for MDLM, indicating that comparable performance can be achieved after seeing only ∼18% of tokens. In terms of

wall-clock time, this corresponds to ∼ 50 hours for LoMDM versus ∼ 150 hours for the 1M-step MDLM reference, making LoMDM **about 3× faster in practice**. Moreover, across the shared wall-clock budget in Figure 5, LoMDM consistently achieves lower PPL than MDLM, further suggesting that it learns faster and more efficiently in practice.

### 5.3. Ablation Studies

We conduct ablation studies for two purposes. First, we test whether the gains of LoMDM persist under different backbone scales. Second, we diagnose plausible but less effective scheduler design choices, including alternative gradient flow, scheduler-head architecture, and scheduler parametrization. All experiments are conducted on LM1B[¶] without sentence packing for 0.5M training steps and evaluated by test PPL (↓), and all other experimental settings are the same as in Section 5.1. Refer to Appendix F.2 for ablation study on $c_1$ and $c_2$.

*Table 4.* Model-size ablation. Test PPL (↓) on LM1B¶ at 0.5M steps with different Transformer backbone configurations. Section 5.1 uses 768 hidden dimensions, 12 blocks, and 12 heads.

| Hidden size | Blocks | Heads | MDLM | LoMDM |
|---|---|---|---|---|
| 512 | 8 | 8 | 40.1 | **34.3** |
| 1024 | 24 | 16 | 27.5 | **24.6** |

*Table 5.* Scheduler-architecture ablation. Test PPL (↓) on LM1B¶ at 0.5M steps.

| Variant | PPL |
|---|---|
| **LoMDM (default)** | **29.2** |
| LoMDM w/o stop-gradient | 36.3 |
| LoMDM w/o Transformer layer | 31.3 |
| MDLM | 32.5 |

*Table 6.* Scheduler-parametrization ablation. Test PPL (↓) on LM1B¶ at 0.5M steps.

| Param. | $(c_1, c_2)$ | PPL |
|---|---|---|
| **NormSig** | **(0.7, 0.65)** | **29.2** |
| Softmax | (0.7, 1) | 34.3 |
| Softmax | (0.7, 3) | 34.1 |
| Softmax | (0.7, 10) | 34.0 |

*Table 7.* Generation-trajectory analysis. Proportion of newly generated tokens in each POS category at early and late decoding stages.

| Model | Steps | Nouns | Verbs | Articles | Prep. |
|---|---|---|---|---|---|
| MDLM | $1 \rightarrow 16$ | 0.30 | 0.18 | 0.10 | 0.15 |
|  | $112 \rightarrow 128$ | 0.29 | 0.18 | 0.09 | 0.14 |
| LoMDM | $1 \rightarrow 16$ | 0.25 | 0.15 | 0.14 | 0.19 |
|  | $112 \rightarrow 128$ | 0.37 | 0.20 | 0.07 | 0.10 |

**LoMDM with different model sizes.** We first vary the Transformer backbone size with various choices of hidden size and number of blocks/heads. In this experiment, LoMDM uses the default $(c_1, c_2) = (0.7, 0.65)$. As shown in Table 4, LoMDM consistently improves over MDLM under both smaller and larger backbones, suggesting that LoMDM with the current hyperparameter choice is robust to backbone scale.

**LoMDM with different scheduler architectures.** We next ablate the scheduler-head architecture in Table 5. Removing stop-gradient substantially worsens PPL, suggesting that directly coupling token prediction and order-priority learning makes optimization difficult. Replacing the Transformer scheduler head with an MLP-only head also underperforms the default, supporting the role of context-aware scheduler modeling. Overall, the default stop-gradient feature sharing with a lightweight Transformer scheduler performs best.

**LoMDM with different scheduler parametrizations.** Finally, we test whether Softmax parametrization can replace our normalized sigmoid design. For Softmax parametrization, the NormSig in Eq. 4–7 is replaced by the Softmax function, *e.g.*, $\alpha_\phi^{(i)}(\mathbf{x}, t) := 1 - t^{c_1 + c_2 \cdot [\text{SoftMax}(g_\phi(f(\mathbf{x})))]_i}$. Table 6 compares NormSig with Softmax variants. Across several Softmax configurations, PPL remains around 34 and is clearly worse than NormSig. We speculate the reason behind the failure mode of Softmax-style parameterization is that it can over-concentrate priority on a small number of positions, whereas NormSig regularizes relative generation priorities and stabilizes training.

### 5.4. Analysis of Generation Trajectories of LoMDM

To examine whether the learned reverse process induces a meaningful generation order, we analyze the POS composition of newly generated tokens along the sampling trajectory. Specifically, we generate 256 length-128 sentences over 128 reverse steps and measure the proportion of newly gener-

ated tokens in each POS category at early and late stages of decoding.

As shown in Table 7, MDLM exhibits a nearly unchanged POS distribution from the early stage to the late stage. In contrast, LoMDM generates more structural words early: articles and prepositions account for 0.33 of newly generated tokens in steps $1 \rightarrow 16$, but only 0.17 in steps $112 \rightarrow 128$. Conversely, semantic content becomes more prominent later, as nouns and verbs increase from 0.40 to 0.57. This suggests that LoMDM first establishes a reliable sentence structure under highly masked contexts and then fills in more content-bearing words once more context is available. We speculate that, for unconditional generation, a coarse-to-fine generation order may be more effective than a left-to-right order. In contrast, what generation order is preferable for conditional generation tasks, such as QA, remains an interesting direction for future work. See Appendix G.2 for the visualization of the generation trajectory of LoMDM.

## 6. Conclusion

We introduced OeMDM, a unified framework of masked diffusion models with various orderings, which treats the noise scheduler as a minimal-constrained and position-dependent object and yields a generalized NELBO. Building on this framework, we proposed LoMDM, which jointly learns a sequence-dependent scheduler and the diffusion backbone through a single NELBO, so that the learned scheduler is directly exploitable at generation time and concentrates learning on more tractable prediction paths. Empirically, LoMDM achieves substantially lower test perplexities than prior discrete diffusion baselines. Moreover, on OWT, LoMDM matches the 1M-step MDLM at only 180K steps, highlighting that LoMDM is a fast and efficient learner.

## Impact Statement

The objective of our work is to advance the discrete diffusion-based language modeling. Potential societal consequences are similar to those of other text generation methods, and we do not anticipate impacts beyond those already well established for generative language models.

## Acknowledgements

This work was supported by the National Research Foundation of Korea under Grant RS-2024-00336454. Additionally, it was supported by Institute for Information & communications Technology Planning & Evaluation(IITP) grant funded by the Korea government(MSIT) (RS-2019-II190075, Artificial Intelligence Graduate School Program(KAIST)) and the Institute of Information & Communications Technology Planning & Evaluation(IITP) grant funded by the Korea government(MSIT) (RS-2025-02304967, AI Star Fellowship(KAIST)).

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

# A. Related works

**Discrete diffusion models.** Diffusion probabilistic models have become a dominant approach for continuous domains such as images, audio, and video (Ho et al., 2020; Song et al., 2021). This success has motivated extensions to discrete domains, including text, leading to discrete diffusion models (Austin et al., 2021; Hoogeboom et al., 2021). The forward corruption process is typically designed in one of two ways: (i) *uniform* corruption, which replaces tokens with random vocabulary elements (Lou et al., 2024; Sahoo et al., 2025), or (ii) *masking*-based corruption, which maps tokens to an absorbing [MASK] state (Sahoo et al., 2024a; Shi et al., 2024). Notably, Lou et al. (2024) propose SEDD, a continuous-time formulation based on a score-entropy objective that accommodates both uniform and masking corruptions.

**Continuous-time masked diffusion models.** Masked diffusion models (MDMs) form a widely used subclass of discrete diffusion for text generation, built around an absorbing mask state (Lou et al., 2024; Sahoo et al., 2024a; Shi et al., 2024). Empirically, masking-based formulations often yield stronger likelihood bounds than their uniform-corruption counterparts, and they have become a common baseline for discrete diffusion language modeling. In the continuous-time setting, a representative instantiation is MDLM (Sahoo et al., 2024a), which defines an absorbing forward process where once a position becomes masked, it remains masked thereafter. This design substantially simplifies the forward dynamics and enables a principled continuous-time training objective based on an NELBO derivation.

**Time-agnostic masked diffusion and its relation to continuous-time formulations.** Beyond continuous-time formulations that rely on explicit time-conditioned schedulers, several works study time-agnostic training for masking-based diffusion models and show that the resulting objectives can be implemented with simple cross-entropy losses (Zheng et al., 2025; Ou et al., 2025). The key intuition is that a scheduler primarily determines the fraction of masked tokens over time; hence many scheduler-dependent terms can be rewritten as functions of the current mask ratio, yielding a practically convenient framework that is widely adopted in recent large-scale MDM implementations (Ye et al., 2025; Nie et al., 2025).

Importantly, the distinction between time-agnostic and continuous-time MDMs does not lie in whether the denoiser is time-conditioned: many continuous-time MDMs, including MDLM, BD3LM, and VADD, keep the denoiser itself time-agnostic (Sahoo et al., 2024a; Arriola et al., 2025; Xie et al., 2025). Rather, the difference is whether the diffusion process is specified via an explicit time-conditioned scheduler. Zheng et al. (2025) derives a scheduler-free NELBO and proposes the First-Hitting sampler, which enables sampling without an explicit scheduler by repeatedly selecting random positions to update. However, time-agnostic formulations effectively assume that the corruption strength is identical across positions and make time implicit via the number (or ratio) of masked tokens. As a result, constructing order-dependent, position-varying MDMs naturally calls for revisiting the continuous-time formulation with explicit, time-conditioned schedulers.

**Importance of generation order in masked diffusion models.** A central determinant of MDM generation quality is the *unmasking order* at inference time. Prior studies have repeatedly observed that naive random unmasking can be suboptimal (Ou et al., 2025; Shih et al., 2022; Kim et al., 2025). In continuous-time MDMs, BD3LM (Arriola et al., 2025) injects a block-wise left-to-right bias and substantially improves over MDLM. Similar phenomena appear in large-scale time-agnostic MDMs: when using random ordering, large-scale MDMs (Ye et al., 2025; Nie et al., 2025) often lag behind large-scale ARMs such as LLaMA-3 (Grattafiori et al., 2024). However, adopting structured decoding—e.g., block-wise L2R generation, within each block, revealing the highest-confidence positions first—can close this gap and even surpass ARMs in some settings. Azangulov et al. (2026) further show that safe parallel unmasking can be achieved without additional training by testing conditional independence among candidate tokens, improving generation performance under a few-step generation setting. Overall, these results highlight that the choice of unmasking order matters in MDMs.

**Learning unmasking order in masked diffusion models.** In the continuous-time MDMs setting, Peng et al. (2025a) introduce a planner, an unmasking module, and post-train this planner to improve a fixed diffusion backbone. Shi et al. (2024) propose GenMD4, which defines vocabulary-dependent learnable schedulers. In contrast to ours, such vocabulary-dependent designs primarily capture global token preferences (e.g., whether certain token types tend to be sampled earlier), rather than adapting the ordering to the specifi sentence context. In the time-agnostic MDMs setting, learning unmasking strategies has been actively explored due to the simplicity of time-agnostic training objectives (Hong et al., 2025; Peng et al., 2025b; Huang et al., 2025). Peng et al. (2025b) propose PAPL, which integrates a planner into the time-agnostic MDM framework, and derive a corresponding NELBO. However, optimizing the NELBO with an explicit planner involves sampling along the Markov chain (i.e., generating intermediate states up to $t = 0$ and an intermediate time $t'$). To avoid this infeasible sampling cost, they replace the planner with the diffusion network's confidence scores and instead train a surrogate objective that samples text from a uniform distribution. Hong et al. (2025) propose post-training an unmasking policy module using GRPO, and prove that the resulting policy can sample closer to the true data distribution. Similarly, Liu et al. (2026) address the

inefficiency of confidence-based unmasking by training a neural indicator that predicts which tokens can be safely unmasked at each step, thereby merging redundant sampling steps and accelerating generation through an optimized token order.

**Connection to learning-order autoregressive models (LO-ARM).** LO-ARM (Wang et al., 2025) studies a closely related problem in the autoregressive framework: jointly learning the generation order and token prediction model. In this sense, LO-ARM and our LoMDM share the same high-level goal, and their training objectives have an intuitive similarity. More specifically, both objectives decompose into an order-weighted token-prediction loss and an order-matching loss: the former concentrates training on high-priority positions, while the latter aligns the oracle order from the full sequence with the order inferred from a partially observed sequence during generation.

However, the two methods are built on fundamentally different modeling foundations. LO-ARM is derived from an autoregressive factorization and learns the order through a variational order model, whereas LoMDM integrates learnable order into the masked diffusion framework through a learnable scheduler. This distinction makes LoMDM the first framework, to our knowledge, that theoretically incorporates jointly learned generation order into MDM training from scratch. Moreover, OeMDM provides a unified view of standard MDMs, ARMs, block diffusion, and learned-scheduler MDMs, while preserving *the key advantage of MDMs: parallel and few-step generation*. Thus, while LO-ARM and LoMDM share a similar motivation, they make parallel contributions in different generative modeling paradigms.

**Summary.** MDMs are a promising alternative to ARMs, but their generation quality depends strongly on the unmasking order. However, existing approaches typically 1) improve the order only via post-training (Hong et al., 2025; Peng et al., 2025a), 2) rely on surrogate objectives (Peng et al., 2025b), or 3) fail to fully capture context-dependent ordering over the entire sequence (Shi et al., 2024). To address these limitations, we return to the canonical continuous-time MDLM formulation and focus on the scheduler as the central mechanism that governs the unmasking process. This perspective enables a principled integration of MDM training with learnable, context-dependent generation orders within a unified theoretical framework.

# B. Preliminaries and Notation

We first introduce basic objects, masked sequences, absorbing mask (monotone-masking) trajectories, and the corresponding path measures that will be used throughout the appendix.

## B.1. Masked sequences and absorbing trajectories

**Definition B.1** (Masked sequence set). For a length-$L$ sequence $\mathbf{x} \in \mathcal{X}^L$, define the masked sequence set as

$$\mathcal{S}(\mathbf{x}) := \left\{ \mathbf{z} \in (\mathcal{X} \cup \{\mathbf{m}\})^L \mid \mathbf{z}^{(i)} \in \{\mathbf{x}^{(i)}, \mathbf{m}\}, \ \forall i \in [L] \right\}. \tag{9}$$

**Definition B.2** (Absorbing mask trajectory). Fix an endpoint $\mathbf{x} \in \mathcal{X}^L$ and a discrete time grid $t(0) < \cdots < t(T)$. Define the absorbing mask trajectory set as

$$\mathcal{S}_{\text{absorb}}(\mathbf{x}, T) := \Big\{ (\mathbf{z}_{t(0)}, \ldots, \mathbf{z}_{t(T)}) \in \big((\mathcal{X} \cup \{\mathbf{m}\})^L\big)^{T+1} \mid \exists \, (M_\tau)_{\tau=0}^T \subseteq [L] \text{ s.t.}$$
$$M_T = [L], \quad M_{\tau-1} \subseteq M_\tau \ (\tau = 1, \ldots, T),$$
$$\mathbf{z}_{t(\tau)}^{(j)} = \mathbf{m} \, \mathbb{1}\{j \in M_\tau\} + \mathbf{x}^{(j)} \, \mathbb{1}\{j \notin M_\tau\}, \ \forall \tau \in \{0, \ldots, T\}, \ \forall j \in [L] \Big\}, \tag{10}$$

Equivalently, along any $\mathbf{z}_{t(0:T)} \in \mathcal{S}_{\text{absorb}}(\mathbf{x}, T)$ the mask is absorbing: $\mathbf{z}_{t(\tau)}^{(i)} = \mathbf{m} \ \Rightarrow \ \mathbf{z}_{t(\tau+1)}^{(i)} = \mathbf{m}$.

For brevity, we omit $T$ in $\mathcal{S}_{\text{absorb}}(\mathbf{x}, T)$.

## B.2. Forward and reverse path measures

The path distributions induced by the model-parameterized reverse process and the true forward/posterior process is defined as follows using chain rule:

$$p_{\theta,\hat{\alpha}_{\mathcal{F}}}(\mathbf{x}, \mathbf{z}_{t(0:T)}) = p_{\theta,\hat{\alpha}_{\mathcal{F}}}(\mathbf{x} \mid \mathbf{z}_{t(0)}) \Big( \prod_{\tau=1}^{T} p_{\theta,\hat{\alpha}_{\mathcal{F}}}(\mathbf{z}_{s(\tau)} \mid \mathbf{z}_{t(\tau)}) \Big) p_{\theta,\hat{\alpha}_{\mathcal{F}}}(\mathbf{z}_{t(T)}),$$

$$q_{\alpha_{\mathcal{F}}}(\mathbf{z}_{t(0:T)} \mid \mathbf{x}) = \Big( \prod_{\tau=1}^{T} q_{\alpha_{\mathcal{F}}}(\mathbf{z}_{s(\tau)} \mid \mathbf{z}_{t(\tau)}, \mathbf{x}) \Big) q_{\alpha_{\mathcal{F}}}(\mathbf{z}_{t(T)} \mid \mathbf{x}),$$

where we use the definition $s(\tau) = \tau/(T+1), t(\tau) = (\tau+1)/(T+1)$ such that $s(\tau) = t(\tau-1)$. Following conventional MDMs (Sahoo et al., 2024a; Shi et al., 2024), we take the terminal forward distribution to be fully masked: $q_{\alpha_{\mathcal{F}}}(\mathbf{z}_{t(T)}^{(i)} \mid \mathbf{x}) = \text{Cat}(\mathbf{m})$ for all $i \in [L]$. Likewise, the model's initial noise distribution is fully masked: $p_{\theta,\hat{\alpha}_{\mathcal{F}}}(\mathbf{z}_{t(T)}^{(i)}) = \text{Cat}(\mathbf{m})$ for all $i \in [L]$. Finally, the reconstruction distribution at $t(0)$ is defined tokenwise as

$$p_{\theta,\hat{\alpha}_{\mathcal{F}}}(\mathbf{x}^{(i)} \mid \mathbf{z}_{t(0)}) = \text{Cat}\Big( \mathbf{x}_{\theta}^{(i)}\big(\mathbf{z}_{t(0)}, t(0)\big) \Big),$$

where $\mathbf{x}_{\theta}^{(i)}(\mathbf{z}_{t(0)}, t(0)) \in \Delta^{V+1}$ denotes the model prediction of token $(i)$.

# C. Derivation of NELBO

In this section, we provide a complete derivation of the NELBO of OeMDM. Before deriving NELBO, we explain the SUBS parametrization (Sahoo et al., 2024a) of $\mathbf{x}_{\theta}$ that was omitted in the main paper:

**Zero Masking Probabilities.** By definition, $\langle \mathbf{x}^{(i)}, \mathbf{m} \rangle = 0$ holds for all $\mathbf{x} \in \mathcal{V}^{L}$ and $i \in [L]$. SUBS parametrization therefore design the diffusion backbone never to output mask prediction such that $\langle \mathbf{x}_{\theta}^{(i)}(\mathbf{z}_t, t), \mathbf{m} \rangle = 0$ for all $i$, i.e., to substitute the logit index corresponding to the mask token with $-\infty$.

**Carry-Over Unmasking.** Once the mask token is unmasked through the generation process, SUBS parametrization desires not to unmask it again. This is accomplished by substituting the output of the diffusion network to simply copy unmasked inputs. Formally, if $\mathbf{z}_t^{(i)} \neq \mathbf{m}$, then $\mathbf{x}_{\theta}^{(i)}(\mathbf{z}_t, t) = \mathbf{z}_t^{(i)}$.

To derive an NELBO, Kullback–Leibler (KL) divergence between the model path distribution and the true reverse-posterior path distribution should be defined, which requires that the model assigns zero probability outside the posterior's support. Under SUBS parametrization, we first prove a lemma that is required for deriving NELBO:

**Lemma C.1** (Conditional absolute continuity of OeMDM). *For any free-form schedulers $\alpha_{\mathcal{F}} \in \mathcal{F}[I]$, $\hat{\alpha}_{\mathcal{F}} \in \mathcal{F}[\hat{I}]$, any fixed $\mathbf{x}$, and any discretized trajectory $\mathbf{z}_{t(0:T)}$, the following statement holds under SUBS parametrization:*

$$p_{\theta,\hat{\alpha}_{\mathcal{F}}}(\mathbf{x}, \mathbf{z}_{t(0:T)}) > 0 \quad \implies \quad q_{\alpha_{\mathcal{F}}}(\mathbf{z}_{t(0:T)} \mid \mathbf{x}) > 0, \tag{11}$$

*or equivalently $p_{\theta,\hat{\alpha}_{\mathcal{F}}}(\mathbf{x}, \mathbf{z}_{t(0:T)}) \ll q_{\alpha_{\mathcal{F}}}(\mathbf{z}_{t(0:T)} \mid \mathbf{x})$ for every fixed $\mathbf{x}$.*

*Proof.* We divide the proof in two steps: 1) we first show that $q_{\alpha_{\mathcal{F}}}(\mathbf{z}_{t(0:T)} \mid \mathbf{x}) > 0$ for all $\mathbf{z}_{t(0:T)} \in \mathcal{S}_{\text{absorb}}(\mathbf{x})$, and 2) prove that $p_{\theta,\hat{\alpha}_{\mathcal{F}}}(\mathbf{x}, \mathbf{z}_{t(0:T)}) > 0$ implies $\mathbf{z}_{t(0:T)} \in \mathcal{S}_{\text{absorb}}(\mathbf{x})$.

**1.** $q_{\alpha_{\mathcal{F}}}(\mathbf{z}_{t(0:T)} \mid \mathbf{x}) > 0, \forall \mathbf{z}_{t(0:T)} \in \mathcal{S}_{\text{absorb}}(\mathbf{x})$.

Consider any $\mathbf{z}_{t(0:T)} \in \mathcal{S}_{\text{absorb}}(\mathbf{x})$. Since $\mathbf{z}_{t(T)} = \mathbf{m}^L$ by definition, $q_{\alpha_{\mathcal{F}}}(\mathbf{z}_{t(T)} \mid \mathbf{x}) = 1$ by definition. We now show that for any $\mathbf{z}_{s(\tau)}(= \mathbf{z}_{t(\tau-1)})$ and $\mathbf{z}_{t(\tau)}$ from path $\mathbf{z}_{t(0:T)} \in \mathcal{S}_{\text{absorb}}(\mathbf{x})$, the transition $q_{\alpha_{\mathcal{F}}}(\mathbf{z}_{s(\tau)} \mid \mathbf{z}_{t(\tau)}, \mathbf{x})$ is positive. There are three cases of $\mathbf{z}_{s(\tau)}^{(i)}, \mathbf{z}_{t(\tau)}^{(i)}$: 1) $\mathbf{z}_{s(\tau)}^{(i)} = \mathbf{m}, \mathbf{z}_{t(\tau)}^{(i)} = \mathbf{m}$, 2) $\mathbf{z}_{s(\tau)}^{(i)} = \mathbf{x}^{(i)}, \mathbf{z}_{t(\tau)}^{(i)} = \mathbf{m}$, and 3) $\mathbf{z}_{s(\tau)}^{(i)} = \mathbf{x}^{(i)}, \mathbf{z}_{t(\tau)}^{(i)} = \mathbf{x}^{(i)}$. To recap, $q_{\alpha_{\mathcal{F}}}(\mathbf{z}_s^{(i)} \mid \mathbf{z}_t^{(i)}, \mathbf{x})$ is given by

$$q_{\alpha_{\mathcal{F}}}(\mathbf{z}_s^{(i)} \mid \mathbf{z}_t, \mathbf{x}) = q_{\alpha_{\mathcal{F}}}(\mathbf{z}_s^{(i)} \mid \mathbf{z}_t^{(i)}, \mathbf{x}) = \begin{cases} \text{Cat}(\mathbf{z}_t^{(i)}), & \text{if } \mathbf{z}_t^{(i)} \neq \mathbf{m}, \\ \text{Cat}\left( \frac{(1-\alpha_{\mathcal{F}}^{(i)}(u,s))\mathbf{m} + (\alpha_{\mathcal{F}}^{(i)}(u,s) - \alpha_{\mathcal{F}}^{(i)}(u,t))\mathbf{x}^{(i)}}{1 - \alpha_{\mathcal{F}}^{(i)}(u,t)} \right), & \text{if } \mathbf{z}_t^{(i)} = \mathbf{m}, \end{cases}$$

Since $\alpha_{\mathcal{F}} \in \mathcal{F}[I]$, for any fixed input the map $t \mapsto \alpha_{\mathcal{F}}^{(i)}(u,t)$ is strictly decreasing with boundary values $\alpha_{\mathcal{F}}^{(i)}(u,0) = 1$ and $\alpha_{\mathcal{F}}^{(i)}(u,1) = 0$. Hence, for any $0 \leq s < t \leq 1$ we have $\alpha_{\mathcal{F}}^{(i)}(u^{(i)}, s) > \alpha_{\mathcal{F}}^{(i)}(u^{(i)}, t)$, and for any $t \in (0,1]$ we have $0 < \alpha_{\mathcal{F}}^{(i)}(u^{(i)}, t) < 1$. Therefore, all of three cases are strictly positive, such that for any $\mathbf{z}_{s(\tau)}(= \mathbf{z}_{t(\tau-1)})$ and $\mathbf{z}_{t(\tau)}$ from path $\mathbf{z}_{t(0:T)} \in \mathcal{S}_{\mathrm{absorb}}(\mathbf{x})$, $q_{\alpha_{\mathcal{F}}}(\mathbf{z}_{s(\tau)}|\mathbf{z}_{t(\tau)}, \mathbf{x})$ is positive, and so is their product $q_{\alpha_{\mathcal{F}}}(\mathbf{z}_{t(0:T)} \mid \mathbf{x})$.

**2.** $p_{\theta, \hat{\alpha}_{\mathcal{F}}}(\mathbf{x}, \mathbf{z}_{t(0:T)}) > 0 \Rightarrow \mathbf{z}_{t(0:T)} \in \mathcal{S}_{\mathrm{absorb}}(\mathbf{x})$.

Suppose $p_{\theta, \hat{\alpha}_{\mathcal{F}}}(\mathbf{x}, \mathbf{z}_{t(0:T)}) > 0$. Then, in particular, all factors in the joint are positive.

First, since the initial noise distribution is a fully-masked sequence, $\mathbf{z}_{t(T)} = \mathbf{m}^L$.

Next, carry-over unmasking implies that if $\mathbf{z}_t^{(i)} \neq \mathbf{m}$, then $\mathbf{x}_\theta^{(i)}(\mathbf{z}_t, t) = \mathbf{z}_t^{(i)}$, so the reverse transition copies that token with probability 1. Formally,

$$\left( \mathbf{z}_{t(\tau+1)}^{(i)} \neq \mathbf{m} \;\Rightarrow\; \mathbf{z}_{t(\tau)}^{(i)} = \mathbf{z}_{t(\tau+1)}^{(i)} \right) \;\Longleftrightarrow\; \left( \mathbf{z}_{t(\tau+1)}^{(i)} \neq \mathbf{m} \;\Rightarrow\; \mathbf{z}_{t(\tau)}^{(i)} \neq \mathbf{m} \right) \;\Longleftrightarrow\; \left( \mathbf{z}_{t(\tau)}^{(i)} = \mathbf{m} \;\Rightarrow\; \mathbf{z}_{t(\tau+1)}^{(i)} = \mathbf{m} \right)$$

satisfies for every trajectory from Markov process starting from $\mathbf{z}_{t(T)} = \mathbf{m}^L$. Therefore, if $p_{\theta, \hat{\alpha}_{\mathcal{F}}}(\mathbf{x}, \mathbf{z}_{t(0:T)}) > 0$, the trajectory $\mathbf{z}_{t(0:T)}$ satisfies absorbing mask property, and $\mathbf{z}_{t(\tau)}^{(i)} \in \{\mathbf{m}, \mathbf{v}\}$ for every $\tau \in [T]$ and fixed one-hot vector $\mathbf{v} \in \mathcal{V}$.

Finally, if there remains mask token in $\mathbf{z}_{t(0)}$, the variational distribution $p_{\theta, \hat{\alpha}_{\mathcal{F}}}(\mathbf{x}^{(i)}|\mathbf{z}_{t(0)}) = \mathrm{Cat}\left( \mathbf{x}_\theta^{(i)}(\mathbf{z}_{t(0)}, t(0)) \right)$ converts it into non-mask token by zero masking property of SUBS parametrization, such that resulting $\mathbf{x}$ is composed of $\mathcal{V}^L$. Since carry-over unmasking property of SUBS parametrization holds for every transition, we can conclude that $\mathbf{z}_{t(\tau)}^{(i)} \in \{\mathbf{m}, \mathbf{x}^{(i)}\}$ for every $\tau \in [T]$ and final state $\mathbf{x} \in \mathcal{V}^L$. Since $\mathbf{z}_{t(\tau)}^{(i)} \in \{\mathbf{x}^{(i)}, \mathbf{m}\}$ for all $i, \tau$, and the trajectory is absorbing, $p_{\theta, \hat{\alpha}_{\mathcal{F}}}(\mathbf{x}, \mathbf{z}_{t(0:T)}) > 0 \Rightarrow \mathbf{z}_{t(0:T)} \in \mathcal{S}_{\mathrm{absorb}}(\mathbf{x})$ holds.

Since $q_{\alpha_{\mathcal{F}}}(\mathbf{z}_{t(0:T)} \mid \mathbf{x}) > 0, \forall \mathbf{z}_{t(0:T)} \in \mathcal{S}_{\mathrm{absorb}}(\mathbf{x})$ and $p_{\theta, \hat{\alpha}_{\mathcal{F}}}(\mathbf{x}, \mathbf{z}_{t(0:T)}) > 0 \Rightarrow \mathbf{z}_{t(0:T)} \in \mathcal{S}_{\mathrm{absorb}}(\mathbf{x})$ hold, the given statement $p_{\theta, \hat{\alpha}_{\mathcal{F}}}(\mathbf{x}, \mathbf{z}_{t(0:T)}) > 0 \implies q_{\alpha_{\mathcal{F}}}(\mathbf{z}_{t(0:T)} \mid \mathbf{x}) > 0$ hold for every fixed $\mathbf{x}$. $\qquad\square$

With Lemma C.1, we now derive NELBO of OeMDM. To recap, NELBO is given as follows:

**Proposition 3.2** (NELBO of OeMDM in continuous time). *Under SUBS parametrization, the NELBO of OeMDM in continuous time is given as follows:*

$$-\log p_{\theta, \hat{\alpha}_{\mathcal{F}}}(\mathbf{x}) \leq \mathcal{L}_{\mathrm{OeMDM}}(\mathbf{x}, \theta, \alpha_{\mathcal{F}}, \hat{\alpha}_{\mathcal{F}})$$
$$= \int_0^1 \mathbb{E}_{q_{\alpha_{\mathcal{F}}}} \left[ \sum_{i=1}^L \langle \mathbf{z}_t^{(i)}, \mathbf{m} \rangle \left\{ \underbrace{-A^{(i)} \log \langle \mathbf{x}_\theta^{(i)}(\mathbf{z}_t, t), \mathbf{x}^{(i)} \rangle}_{\mathcal{L}_{\mathrm{main}}} + \underbrace{A^{(i)}(\log A^{(i)} - \log \hat{A}^{(i)}) - (A^{(i)} - \hat{A}^{(i)})}_{\mathcal{L}_{\mathrm{velocity}}} \right\} \right] dt,$$

*where the structure of $\mathcal{L}_{\mathrm{main}}$ is equal to $\mathcal{L}_{\mathrm{mdlm}}$ and $\mathcal{L}_{\mathrm{velocity}} \geq 0$ achieves 0 when $A = \hat{A}$.*

Following Sahoo et al. (2024a), discretize the time interval $\mathcal{T}$ with $T + 1$ steps, and define $s(\tau) = \tau/(T+1)$ and $t(\tau) = (\tau+1)/(T+1)$ such that generative distribution is divided into $T$ diffusion reverse steps $(\mathbf{z}_{t(T)} \to \cdots \to \mathbf{z}_{t(0)})$

and 1 reconstruction step ($\mathbf{z}_{t(0)} \to \mathbf{x}$). The negative evidence lower bound (NELBO) can be obtained as follows:

$$-\log p_{\theta,\hat{\alpha}_{\mathcal{F}}}(\mathbf{x}) = -\log \sum_{\mathbf{z}_{t(0:T)}} p_{\theta,\hat{\alpha}_{\mathcal{F}}}(\mathbf{x}, \mathbf{z}_{t(0:T)}) \tag{12}$$

$$= -\log \sum_{\mathbf{z}_{t(0:T)}} q_{\alpha_{\mathcal{F}}}(\mathbf{z}_{t(0:T)} \mid \mathbf{x}) \frac{p_{\theta,\hat{\alpha}_{\mathcal{F}}}(\mathbf{x}, \mathbf{z}_{t(0:T)})}{q_{\alpha_{\mathcal{F}}}(\mathbf{z}_{t(0:T)} \mid \mathbf{x})} \qquad \because \text{Lemma C.1} \tag{13}$$

$$= -\log \mathbb{E}_{\mathbf{z}_{t(0:T)} \sim q_{\alpha_{\mathcal{F}}}(\mathbf{z}_{t(0:T)}|\mathbf{x})} \left[ \frac{p_{\theta,\hat{\alpha}_{\mathcal{F}}}(\mathbf{x}, \mathbf{z}_{t(0:T)})}{q_{\alpha_{\mathcal{F}}}(\mathbf{z}_{t(0:T)} \mid \mathbf{x})} \right] \tag{14}$$

$$\leq -\mathbb{E}_{q_{\alpha_{\mathcal{F}}}(\mathbf{z}_{t(0:T)}|\mathbf{x})} \left[ \log \frac{p_{\theta,\hat{\alpha}_{\mathcal{F}}}(\mathbf{x}, \mathbf{z}_{t(0:T)})}{q_{\alpha_{\mathcal{F}}}(\mathbf{z}_{t(0:T)} \mid \mathbf{x})} \right] \qquad \because \text{Jensen's inequality} \tag{15}$$

$$= \mathbb{E}_{q_{\alpha_{\mathcal{F}}}} \Bigg[ \underbrace{-\log p_{\theta,\hat{\alpha}_{\mathcal{F}}}(\mathbf{x} \mid \mathbf{z}_{t(0)})}_{\mathcal{L}_{\text{reconstruction}}} + \underbrace{\sum_{\tau=1}^{T} D_{\text{KL}}\big(q_{\alpha_{\mathcal{F}}}(\mathbf{z}_{s(\tau)} \mid \mathbf{z}_{t(\tau)}, \mathbf{x}) \,\|\, p_{\theta,\hat{\alpha}_{\mathcal{F}}}(\mathbf{z}_{s(\tau)} \mid \mathbf{z}_{t(\tau)}))}_{\mathcal{L}_{\text{diffusion}}^{T}}$$

$$+ \underbrace{D_{\text{KL}}\big(q_{\alpha_{\mathcal{F}}}(\mathbf{z}_{t(T)} \mid \mathbf{x}) \,\|\, p_{\theta,\hat{\alpha}_{\mathcal{F}}}(\mathbf{z}_{t(T)}))}_{\mathcal{L}_{\text{prior}}} \Bigg]. \tag{16}$$

Hereafter, we omit $\tau$ in $t(\tau)$ and $s(\tau)$ for brevity. We first derive $\mathcal{L}_{\text{diffusion}}^{\infty} = \lim_{T \to \infty} \mathcal{L}_{\text{diffusion}}^{T}$.

## C.1. Breaking Sequence-Level Diffusion Loss into Token-Level Diffusion Loss

We breakdown $D_{\text{KL}}(q_{\alpha_{\mathcal{F}}}(\mathbf{z}_s \mid \mathbf{z}_t, \mathbf{x}) \,\|\, p_{\theta,\hat{\alpha}_{\mathcal{F}}}(\mathbf{z}_s \mid \mathbf{z}_t))$ into token-wsie KL divergence:

$$D_{\text{KL}}\big(q_{\alpha_{\mathcal{F}}}(\mathbf{z}_s \mid \mathbf{z}_t, \mathbf{x}) \,\|\, p_{\theta,\hat{\alpha}_{\mathcal{F}}}(\mathbf{z}_s \mid \mathbf{z}_t)\big)$$

$$= \sum_{\mathbf{z}_s} \left( q_{\alpha_{\mathcal{F}}}(\mathbf{z}_s \mid \mathbf{z}_t, \mathbf{x}) \log \frac{q_{\alpha_{\mathcal{F}}}(\mathbf{z}_s \mid \mathbf{z}_t, \mathbf{x})}{p_{\theta,\hat{\alpha}_{\mathcal{F}}}(\mathbf{z}_s \mid \mathbf{z}_t)} \right)$$

$$= \sum_{\mathbf{z}_s} \left( \Big( \prod_{j=1}^{L} q_{\alpha_{\mathcal{F}}}(\mathbf{z}_s^{(j)} \mid \mathbf{z}_t^{(j)}, \mathbf{x}) \Big) \log \frac{\prod_{i=1}^{L} q_{\alpha_{\mathcal{F}}}(\mathbf{z}_s^{(i)} \mid \mathbf{z}_t^{(i)}, \mathbf{x})}{\prod_{i=1}^{L} p_{\theta,\hat{\alpha}_{\mathcal{F}}}(\mathbf{z}_s^{(i)} \mid \mathbf{z}_t)} \right) \quad \because \begin{cases} q_{\alpha_{\mathcal{F}}}(\mathbf{z}_s \mid \mathbf{z}_t, \mathbf{x}) = \prod_{j=1}^{L} q_{\alpha_{\mathcal{F}}}(\mathbf{z}_s^{(j)} \mid \mathbf{z}_t^{(j)}, \mathbf{x}), \\ p_{\theta,\hat{\alpha}_{\mathcal{F}}}(\mathbf{z}_s \mid \mathbf{z}_t) = \prod_{i=1}^{L} p_{\theta,\hat{\alpha}_{\mathcal{F}}}(\mathbf{z}_s^{(i)} \mid \mathbf{z}_t) \end{cases}$$

$$= \sum_{\mathbf{z}_s} \left( \Big( \prod_{j=1}^{L} q_{\alpha_{\mathcal{F}}}(\mathbf{z}_s^{(j)} \mid \mathbf{z}_t^{(j)}, \mathbf{x}) \Big) \sum_{i=1}^{L} \log \frac{q_{\alpha_{\mathcal{F}}}(\mathbf{z}_s^{(i)} \mid \mathbf{z}_t^{(i)}, \mathbf{x})}{p_{\theta,\hat{\alpha}_{\mathcal{F}}}(\mathbf{z}_s^{(i)} \mid \mathbf{z}_t)} \right)$$

$$= \sum_{\mathbf{z}_s} \left( \sum_{i=1}^{L} \left( \Big( \prod_{j=1}^{L} q_{\alpha_{\mathcal{F}}}(\mathbf{z}_s^{(j)} \mid \mathbf{z}_t^{(j)}, \mathbf{x}) \Big) \log \frac{q_{\alpha_{\mathcal{F}}}(\mathbf{z}_s^{(i)} \mid \mathbf{z}_t^{(i)}, \mathbf{x})}{p_{\theta,\hat{\alpha}_{\mathcal{F}}}(\mathbf{z}_s^{(i)} \mid \mathbf{z}_t)} \right) \right)$$

$$= \sum_{i=1}^{L} \sum_{\mathbf{z}_s^{(i)}} \sum_{\mathbf{z}_s^{(1)}} \cdots \sum_{\mathbf{z}_s^{(i-1)}} \sum_{\mathbf{z}_s^{(i+1)}} \cdots \sum_{\mathbf{z}_s^{(L)}} \left( \Big( \prod_{j=1}^{L} q_{\alpha_{\mathcal{F}}}(\mathbf{z}_s^{(j)} \mid \mathbf{z}_t^{(j)}, \mathbf{x}) \Big) \log \frac{q_{\alpha_{\mathcal{F}}}(\mathbf{z}_s^{(i)} \mid \mathbf{z}_t^{(i)}, \mathbf{x})}{p_{\theta,\hat{\alpha}_{\mathcal{F}}}(\mathbf{z}_s^{(i)} \mid \mathbf{z}_t)} \right)$$

$$= \sum_{i=1}^{L} \sum_{\mathbf{z}_s^{(i)}} \sum_{\mathbf{z}_s^{(1)}} \cdots \sum_{\mathbf{z}_s^{(i-1)}} \sum_{\mathbf{z}_s^{(i+1)}} \cdots \sum_{\mathbf{z}_s^{(L)}} \left( q_{\alpha_{\mathcal{F}}}(\mathbf{z}_s^{(i)} \mid \mathbf{z}_t^{(i)}, \mathbf{x}) \Big( \prod_{j \neq i} q_{\alpha_{\mathcal{F}}}(\mathbf{z}_s^{(j)} \mid \mathbf{z}_t^{(j)}, \mathbf{x}) \Big) \log \frac{q_{\alpha_{\mathcal{F}}}(\mathbf{z}_s^{(i)} \mid \mathbf{z}_t^{(i)}, \mathbf{x})}{p_{\theta,\hat{\alpha}_{\mathcal{F}}}(\mathbf{z}_s^{(i)} \mid \mathbf{z}_t)} \right)$$

$$= \sum_{i=1}^{L} \sum_{\mathbf{z}_s^{(i)}} \left( q_{\alpha_{\mathcal{F}}}(\mathbf{z}_s^{(i)} \mid \mathbf{z}_t^{(i)}, \mathbf{x}) \log \frac{q_{\alpha_{\mathcal{F}}}(\mathbf{z}_s^{(i)} \mid \mathbf{z}_t^{(i)}, \mathbf{x})}{p_{\theta,\hat{\alpha}_{\mathcal{F}}}(\mathbf{z}_s^{(i)} \mid \mathbf{z}_t)} \underbrace{\sum_{\mathbf{z}_s^{(1)}} \cdots \sum_{\mathbf{z}_s^{(i-1)}} \sum_{\mathbf{z}_s^{(i+1)}} \cdots \sum_{\mathbf{z}_s^{(L)}} \Big( \prod_{j \neq i} q_{\alpha_{\mathcal{F}}}(\mathbf{z}_s^{(j)} \mid \mathbf{z}_t^{(j)}, \mathbf{x}) \Big)}_{=1 \text{ by } (*)} \right)$$

$$= \sum_{i=1}^{L} \sum_{\mathbf{z}_s^{(i)}} q_{\alpha_{\mathcal{F}}}(\mathbf{z}_s^{(i)} \mid \mathbf{z}_t^{(i)}, \mathbf{x}) \log \frac{q_{\alpha_{\mathcal{F}}}(\mathbf{z}_s^{(i)} \mid \mathbf{z}_t^{(i)}, \mathbf{x})}{p_{\theta,\hat{\alpha}_{\mathcal{F}}}(\mathbf{z}_s^{(i)} \mid \mathbf{z}_t)} = \sum_{i=1}^{L} D_{\text{KL}}\Big( q_{\alpha_{\mathcal{F}}}(\mathbf{z}_s^{(i)} \mid \mathbf{z}_t^{(i)}, \mathbf{x}) \,\|\, p_{\theta,\hat{\alpha}_{\mathcal{F}}}(\mathbf{z}_s^{(i)} \mid \mathbf{z}_t) \Big).$$

where (*) holds by marginalization (or sum of product):

$$\sum_{\mathbf{z}_s^{(1)}} \cdots \sum_{\mathbf{z}_s^{(i-1)}} \sum_{\mathbf{z}_s^{(i+1)}} \cdots \sum_{\mathbf{z}_s^{(L)}} \Big( \prod_{j \neq i} q_{\alpha_{\mathcal{F}}}(\mathbf{z}_s^{(j)} \mid \mathbf{z}_t^{(j)}, \mathbf{x}) \Big)$$

$$= \sum_{\mathbf{z}_s^{(1)}} \cdots \sum_{\mathbf{z}_s^{(i-1)}} \sum_{\mathbf{z}_s^{(i+1)}} \cdots \sum_{\mathbf{z}_s^{(L)}} \Big( q_{\alpha_{\mathcal{F}}}(\mathbf{z}_s^{(1)} \mid \mathbf{z}_t^{(1)}, \mathbf{x}) \cdots q_{\alpha_{\mathcal{F}}}(\mathbf{z}_s^{(i-1)} \mid \mathbf{z}_t^{(i-1)}, \mathbf{x})$$

$$\cdot q_{\alpha_{\mathcal{F}}}(\mathbf{z}_s^{(i+1)} \mid \mathbf{z}_t^{(i+1)}, \mathbf{x}) \cdots q_{\alpha_{\mathcal{F}}}(\mathbf{z}_s^{(L)} \mid \mathbf{z}_t^{(L)}, \mathbf{x}) \Big)$$

$$= \sum_{\mathbf{z}_s^{(1)}} \Big( q_{\alpha_{\mathcal{F}}}(\mathbf{z}_s^{(1)} \mid \mathbf{z}_t^{(1)}, \mathbf{x}) \sum_{\mathbf{z}_s^{(2)}} \Big( q_{\alpha_{\mathcal{F}}}(\mathbf{z}_s^{(2)} \mid \mathbf{z}_t^{(2)}, \mathbf{x}) \cdots$$

$$\cdots \sum_{\mathbf{z}_s^{(i-1)}} \Big( q_{\alpha_{\mathcal{F}}}(\mathbf{z}_s^{(i-1)} \mid \mathbf{z}_t^{(i-1)}, \mathbf{x}) \sum_{\mathbf{z}_s^{(i+1)}} \Big( q_{\alpha_{\mathcal{F}}}(\mathbf{z}_s^{(i+1)} \mid \mathbf{z}_t^{(i+1)}, \mathbf{x}) \cdots$$

$$\cdots \sum_{\mathbf{z}_s^{(L)}} q_{\alpha_{\mathcal{F}}}(\mathbf{z}_s^{(L)} \mid \mathbf{z}_t^{(L)}, \mathbf{x}) \Big) \cdots \Big) \cdots \Big)$$

$$= \Big( \sum_{\mathbf{z}_s^{(1)}} q_{\alpha_{\mathcal{F}}}(\mathbf{z}_s^{(1)} \mid \mathbf{z}_t^{(1)}, \mathbf{x}) \Big) \cdots \Big( \sum_{\mathbf{z}_s^{(i-1)}} q_{\alpha_{\mathcal{F}}}(\mathbf{z}_s^{(i-1)} \mid \mathbf{z}_t^{(i-1)}, \mathbf{x}) \Big)$$

$$\cdot \Big( \sum_{\mathbf{z}_s^{(i+1)}} q_{\alpha_{\mathcal{F}}}(\mathbf{z}_s^{(i+1)} \mid \mathbf{z}_t^{(i+1)}, \mathbf{x}) \Big) \cdots \Big( \sum_{\mathbf{z}_s^{(L)}} q_{\alpha_{\mathcal{F}}}(\mathbf{z}_s^{(L)} \mid \mathbf{z}_t^{(L)}, \mathbf{x}) \Big) = 1 \cdots 1 = 1,$$

where $\sum_{\mathbf{z}_s^{(\ell)}} q_{\alpha_{\mathcal{F}}}(\mathbf{z}_s^{(\ell)} \mid \mathbf{z}_t^{(\ell)}, \mathbf{x}) = 1$ for every $\ell$ since the definition of $q$ is categorical distribution.

### C.2. Deriving Token-level KL into Closed-Form Equation

From the previous decomposition, the total KL can be written as the sum of token-wise terms:

$$D_{\mathrm{KL}}\big( q_{\alpha_{\mathcal{F}}}(\mathbf{z}_s \mid \mathbf{z}_t, \mathbf{x}) \,\big\|\, p_{\theta, \hat{\alpha}_{\mathcal{F}}}(\mathbf{z}_s \mid \mathbf{z}_t) \big) = \sum_{i=1}^{L} D_{\mathrm{KL}}\big( q_{\alpha_{\mathcal{F}}}(\mathbf{z}_s^{(i)} \mid \mathbf{z}_t^{(i)}, \mathbf{x}) \,\big\|\, p_{\theta, \hat{\alpha}_{\mathcal{F}}}(\mathbf{z}_s^{(i)} \mid \mathbf{z}_t) \big). \tag{17}$$

For each token $i$, since $\mathbf{z}_t^{(i)} \in \{\mathbf{x}^{(i)}, \mathbf{m}\}$, we can separate the two cases as:

$$D_{\mathrm{KL}}\Big( q_{\alpha_{\mathcal{F}}}(\mathbf{z}_s^{(i)} \mid \mathbf{z}_t^{(i)}, \mathbf{x}) \,\big\|\, p_{\theta, \hat{\alpha}_{\mathcal{F}}}(\mathbf{z}_s^{(i)} \mid \mathbf{z}_t) \Big) = D_{\mathrm{KL}}\Big( q_{\alpha_{\mathcal{F}}}(\mathbf{z}_s^{(i)} \mid \mathbf{z}_t^{(i)} = \mathbf{x}^{(i)}, \mathbf{x}) \,\big\|\, p_{\theta, \hat{\alpha}_{\mathcal{F}}}(\mathbf{z}_s^{(i)} \mid \mathbf{z}_t, \mathbf{z}_t^{(i)} = \mathbf{x}^{(i)}) \Big) \langle \mathbf{z}_t^{(i)}, \mathbf{x}^{(i)} \rangle$$

$$+ D_{\mathrm{KL}}\Big( q_{\alpha_{\mathcal{F}}}(\mathbf{z}_s^{(i)} \mid \mathbf{z}_t^{(i)} = \mathbf{m}, \mathbf{x}) \,\big\|\, p_{\theta, \hat{\alpha}_{\mathcal{F}}}(\mathbf{z}_s^{(i)} \mid \mathbf{z}_t, \mathbf{z}_t^{(i)} = \mathbf{m}) \Big) \langle \mathbf{z}_t^{(i)}, \mathbf{m} \rangle. \tag{18}$$

For breaking the KL term, we denote elements of the arbitrary input domain $\mathcal{I}$ and $\hat{\mathcal{I}}$ as $u$ and $\hat{u}$ respectively, *i.e.*, $u \in \mathcal{I}$ and $\hat{u} \in \hat{\mathcal{I}}$.

**Case 1: $\mathbf{z}_t^{(i)} = \mathbf{x}^{(i)}$.** In this case, both posteriors collapse to the same categorical atom:

$$q_{\alpha_{\mathcal{F}}}(\mathbf{z}_s^{(i)} \mid \mathbf{z}_t^{(i)} = \mathbf{x}^{(i)}, \mathbf{x}) = p_{\theta, \hat{\alpha}_{\mathcal{F}}}(\mathbf{z}_s^{(i)} \mid \mathbf{z}_t, \mathbf{z}_t^{(i)} = \mathbf{x}^{(i)}) = \mathrm{Cat}(\mathbf{z}_s^{(i)}; \mathbf{x}^{(i)}),$$

thus

$$D_{\mathrm{KL}}\Big( q_{\alpha_{\mathcal{F}}}(\mathbf{z}_s^{(i)} \mid \mathbf{z}_t^{(i)} = \mathbf{x}^{(i)}, \mathbf{x}) \,\big\|\, p_{\theta, \hat{\alpha}_{\mathcal{F}}}(\mathbf{z}_s^{(i)} \mid \mathbf{z}_t, \mathbf{z}_t^{(i)} = \mathbf{x}^{(i)}) \Big) = 0. \tag{19}$$

**Case 2: $\mathbf{z}_t^{(i)} = \mathbf{m}$.** Using the definitions of the forward and reverse posteriors defined earlier, we have:

$$q_{\alpha_{\mathcal{F}}}(\mathbf{z}_s^{(i)} = \mathbf{x}^{(i)} \mid \mathbf{z}_t^{(i)} = \mathbf{m}, \mathbf{x}) = \frac{\alpha_{\mathcal{F}}^{(i)}(u,s) - \alpha_{\mathcal{F}}^{(i)}(u,t)}{1 - \alpha_{\mathcal{F}}^{(i)}(u,t)}, \tag{20}$$

$$q_{\alpha_{\mathcal{F}}}(\mathbf{z}_s^{(i)} = \mathbf{m} \mid \mathbf{z}_t^{(i)} = \mathbf{m}, \mathbf{x}) = \frac{1 - \alpha_{\mathcal{F}}^{(i)}(u,s)}{1 - \alpha_{\mathcal{F}}^{(i)}(u,t)}, \tag{21}$$

$$p_{\theta,\hat{\alpha}_{\mathcal{F}}}(\mathbf{z}_s^{(i)} = \mathbf{x}^{(i)} \mid \mathbf{z}_t, \mathbf{z}_t^{(i)} = \mathbf{m}) = \frac{(\hat{\alpha}_{\mathcal{F}}^{(i)}(\hat{u},s) - \hat{\alpha}_{\mathcal{F}}^{(i)}(\hat{u},t)) \left\langle \mathbf{x}_\theta^{(i)}(\mathbf{z}_t,t), \mathbf{x}^{(i)} \right\rangle}{1 - \hat{\alpha}_{\mathcal{F}}^{(i)}(\hat{u},t)}, \tag{22}$$

$$p_{\theta,\hat{\alpha}_{\mathcal{F}}}(\mathbf{z}_s^{(i)} = \mathbf{m} \mid \mathbf{z}_t, \mathbf{z}_t^{(i)} = \mathbf{m}) = \frac{1 - \hat{\alpha}_{\mathcal{F}}^{(i)}(\hat{u},s)}{1 - \hat{\alpha}_{\mathcal{F}}^{(i)}(\hat{u},t)}. \tag{23}$$

We now directly compute the KL divergence:

$$D_{\mathrm{KL}}\Big(q_{\alpha_{\mathcal{F}}}(\mathbf{z}_s^{(i)} \mid \mathbf{z}_t^{(i)} = \mathbf{m}, \mathbf{x}) \,\big\|\, p_{\theta,\hat{\alpha}_{\mathcal{F}}}(\mathbf{z}_s^{(i)} \mid \mathbf{z}_t, \mathbf{z}_t^{(i)} = \mathbf{m})\Big) \tag{24}$$

$$= \sum_{\mathbf{z}_s^{(i)} \in \{\mathbf{x}^{(i)}, \mathbf{m}\}} q_{\alpha_{\mathcal{F}}}(\mathbf{z}_s^{(i)} \mid \mathbf{z}_t^{(i)} = \mathbf{m}, \mathbf{x}) \log \frac{q_{\alpha_{\mathcal{F}}}(\mathbf{z}_s^{(i)} \mid \mathbf{z}_t^{(i)} = \mathbf{m}, \mathbf{x})}{p_{\theta,\hat{\alpha}_{\mathcal{F}}}(\mathbf{z}_s^{(i)} \mid \mathbf{z}_t, \mathbf{z}_t^{(i)} = \mathbf{m})}. \tag{25}$$

Explicitly expanding both terms:

$$D_{\mathrm{KL}}\Big(q_{\alpha_{\mathcal{F}}}(\mathbf{z}_s^{(i)} \mid \mathbf{z}_t^{(i)} = \mathbf{m}, \mathbf{x}) \,\big\|\, p_{\theta,\hat{\alpha}_{\mathcal{F}}}(\mathbf{z}_s^{(i)} \mid \mathbf{z}_t, \mathbf{z}_t^{(i)} = \mathbf{m})\Big)$$

$$= q_{\alpha_{\mathcal{F}}}(\mathbf{z}_s^{(i)} = \mathbf{x}^{(i)} \mid \mathbf{z}_t^{(i)} = \mathbf{m}, \mathbf{x}) \log \frac{q_{\alpha_{\mathcal{F}}}(\mathbf{z}_s^{(i)} = \mathbf{x}^{(i)} \mid \mathbf{z}_t^{(i)} = \mathbf{m}, \mathbf{x})}{p_{\theta,\hat{\alpha}_{\mathcal{F}}}(\mathbf{z}_s^{(i)} = \mathbf{x}^{(i)} \mid \mathbf{z}_t, \mathbf{z}_t^{(i)} = \mathbf{m})}$$

$$+ q_{\alpha_{\mathcal{F}}}(\mathbf{z}_s^{(i)} = \mathbf{m} \mid \mathbf{z}_t^{(i)} = \mathbf{m}, \mathbf{x}) \log \frac{q_{\alpha_{\mathcal{F}}}(\mathbf{z}_s^{(i)} = \mathbf{m} \mid \mathbf{z}_t^{(i)} = \mathbf{m}, \mathbf{x})}{p_{\theta,\hat{\alpha}_{\mathcal{F}}}(\mathbf{z}_s^{(i)} = \mathbf{m} \mid \mathbf{z}_t, \mathbf{z}_t^{(i)} = \mathbf{m})}. \tag{26}$$

Now substitute each probability definition. For the first ratio:

$$\frac{q_{\alpha_{\mathcal{F}}}(\mathbf{z}_s^{(i)} = \mathbf{x}^{(i)} \mid \mathbf{z}_t^{(i)} = \mathbf{m}, \mathbf{x})}{p_{\theta,\hat{\alpha}_{\mathcal{F}}}(\mathbf{z}_s^{(i)} = \mathbf{x}^{(i)} \mid \mathbf{z}_t, \mathbf{z}_t^{(i)} = \mathbf{m})} = \frac{\dfrac{\alpha_{\mathcal{F}}^{(i)}(u,s) - \alpha_{\mathcal{F}}^{(i)}(u,t)}{1 - \alpha_{\mathcal{F}}^{(i)}(u,t)}}{\dfrac{(\hat{\alpha}_{\mathcal{F}}^{(i)}(\hat{u},s) - \hat{\alpha}_{\mathcal{F}}^{(i)}(\hat{u},t)) \left\langle \mathbf{x}_\theta^{(i)}(\mathbf{z}_t,t), \mathbf{x}^{(i)} \right\rangle}{1 - \hat{\alpha}_{\mathcal{F}}^{(i)}(\hat{u},t)}}$$

$$= \frac{(\alpha_{\mathcal{F}}^{(i)}(u,s) - \alpha_{\mathcal{F}}^{(i)}(u,t))(1 - \hat{\alpha}_{\mathcal{F}}^{(i)}(\hat{u},t))}{(1 - \alpha_{\mathcal{F}}^{(i)}(u,t))(\hat{\alpha}_{\mathcal{F}}^{(i)}(\hat{u},s) - \hat{\alpha}_{\mathcal{F}}^{(i)}(\hat{u},t)) \left\langle \mathbf{x}_\theta^{(i)}(\mathbf{z}_t,t), \mathbf{x}^{(i)} \right\rangle}. \tag{27}$$

For the second ratio:

$$\frac{q_{\alpha_{\mathcal{F}}}(\mathbf{z}_s^{(i)} = \mathbf{m} \mid \mathbf{z}_t^{(i)} = \mathbf{m}, \mathbf{x})}{p_{\theta,\hat{\alpha}_{\mathcal{F}}}(\mathbf{z}_s^{(i)} = \mathbf{m} \mid \mathbf{z}_t, \mathbf{z}_t^{(i)} = \mathbf{m})} = \frac{\dfrac{1 - \alpha_{\mathcal{F}}^{(i)}(u,s)}{1 - \alpha_{\mathcal{F}}^{(i)}(u,t)}}{\dfrac{1 - \hat{\alpha}_{\mathcal{F}}^{(i)}(\hat{u},s)}{1 - \hat{\alpha}_{\mathcal{F}}^{(i)}(\hat{u},t)}}$$

$$= \frac{(1 - \alpha_{\mathcal{F}}^{(i)}(u,s))(1 - \hat{\alpha}_{\mathcal{F}}^{(i)}(\hat{u},t))}{(1 - \alpha_{\mathcal{F}}^{(i)}(u,t))(1 - \hat{\alpha}_{\mathcal{F}}^{(i)}(\hat{u},s))}. \tag{28}$$

Substituting Eq. 27 and Eq. 28 into Eq. 26, we have:

$$D_{\mathrm{KL}}\Big(q_{\alpha_{\mathcal{F}}}(\mathbf{z}_s^{(i)} \mid \mathbf{z}_t^{(i)} = \mathbf{m}, \mathbf{x}) \,\big\|\, p_{\theta,\hat{\alpha}_{\mathcal{F}}}(\mathbf{z}_s^{(i)} \mid \mathbf{z}_t, \mathbf{z}_t^{(i)} = \mathbf{m})\Big)$$

$$= \frac{\alpha_{\mathcal{F}}^{(i)}(u,s) - \alpha_{\mathcal{F}}^{(i)}(u,t)}{1 - \alpha_{\mathcal{F}}^{(i)}(u,t)} \log \frac{(\alpha_{\mathcal{F}}^{(i)}(u,s) - \alpha_{\mathcal{F}}^{(i)}(u,t))(1 - \hat{\alpha}_{\mathcal{F}}^{(i)}(\hat{u},t))}{(1 - \alpha_{\mathcal{F}}^{(i)}(u,t))(\hat{\alpha}_{\mathcal{F}}^{(i)}(\hat{u},s) - \hat{\alpha}_{\mathcal{F}}^{(i)}(\hat{u},t))\langle \mathbf{x}_\theta^{(i)}(\mathbf{z}_t,t), \mathbf{x}^{(i)}\rangle}$$

$$+ \frac{1 - \alpha_{\mathcal{F}}^{(i)}(u,s)}{1 - \alpha_{\mathcal{F}}^{(i)}(u,t)} \log \frac{(1 - \alpha_{\mathcal{F}}^{(i)}(u,s))(1 - \hat{\alpha}_{\mathcal{F}}^{(i)}(\hat{u},t))}{(1 - \alpha_{\mathcal{F}}^{(i)}(u,t))(1 - \hat{\alpha}_{\mathcal{F}}^{(i)}(\hat{u},s))}. \quad (29)$$

Finally, since this applies only for masked tokens ($\mathbf{z}_t^{(i)} = \mathbf{m}$), the overall KL divergence can be expressed as:

$$D_{\mathrm{KL}}\big(q_{\alpha_{\mathcal{F}}}(\mathbf{z}_s \mid \mathbf{z}_t, \mathbf{x}) \,\big\|\, p_{\theta,\hat{\alpha}_{\mathcal{F}}}(\mathbf{z}_s \mid \mathbf{z}_t)\big) = \sum_{i=1}^{L} \langle \mathbf{z}_t^{(i)}, \mathbf{m}\rangle \, D_{\mathrm{KL}}\Big(q_{\alpha_{\mathcal{F}}}(\mathbf{z}_s^{(i)} \mid \mathbf{z}_t^{(i)} = \mathbf{m}, \mathbf{x}) \,\big\|\, p_{\theta,\hat{\alpha}_{\mathcal{F}}}(\mathbf{z}_s^{(i)} \mid \mathbf{z}_t, \mathbf{z}_t^{(i)} = \mathbf{m})\Big). \quad (30)$$

### C.3. Diffusion Loss into Tractable Loss with Infinite Discretization Steps

The diffusion loss term is

$$\mathcal{L}_{\mathrm{diffusion}}^{T} = \mathbb{E}_{q_{\alpha_{\mathcal{F}}}}\left[ \sum_{\tau=1}^{T} D_{\mathrm{KL}}\big(q_{\alpha_{\mathcal{F}}}(\mathbf{z}_{s(\tau)} \mid \mathbf{z}_{t(\tau)}, \mathbf{x}) \,\big\|\, p_{\theta,\hat{\alpha}_{\mathcal{F}}}(\mathbf{z}_{s(\tau)} \mid \mathbf{z}_{t(\tau)}))\right] \quad (31)$$

$$= \mathbb{E}_{q_{\alpha_{\mathcal{F}}}}\left[ T \cdot \sum_{\tau=1}^{T} \frac{1}{T} D_{\mathrm{KL}}\big(q_{\alpha_{\mathcal{F}}}(\mathbf{z}_{s(\tau)} \mid \mathbf{z}_{t(\tau)}, \mathbf{x}) \,\big\|\, p_{\theta,\hat{\alpha}_{\mathcal{F}}}(\mathbf{z}_{s(\tau)} \mid \mathbf{z}_{t(\tau)}))\right] \quad (32)$$

$$= \mathbb{E}_{q_{\alpha_{\mathcal{F}}}}\left[ T \mathbb{E}_{t \in \{\frac{2}{T+1}, \frac{3}{T+1}, \ldots, 1\}} \Big[ D_{\mathrm{KL}}\big(q_{\alpha_{\mathcal{F}}}(\mathbf{z}_s \mid \mathbf{z}_t, \mathbf{x}) \,\big\|\, p_{\theta,\hat{\alpha}_{\mathcal{F}}}(\mathbf{z}_s \mid \mathbf{z}_t))\Big]\right] \quad (33)$$

$$= \mathbb{E}_{q_{\alpha_{\mathcal{F}}}}\left[ \mathbb{E}_{t \in \{\frac{2}{T+1}, \frac{3}{T+1}, \ldots, 1\}} \Big[ T \cdot D_{\mathrm{KL}}\big(q_{\alpha_{\mathcal{F}}}(\mathbf{z}_s \mid \mathbf{z}_t, \mathbf{x}) \,\big\|\, p_{\theta,\hat{\alpha}_{\mathcal{F}}}(\mathbf{z}_s \mid \mathbf{z}_t))\Big]\right] \quad (34)$$

$$= \mathbb{E}_{q_{\alpha_{\mathcal{F}}}}\left[ \mathbb{E}_{t \in \{\frac{2}{T+1}, \frac{3}{T+1}, \ldots, 1\}} \Big[ \sum_{i=1}^{L} \langle \mathbf{z}_t^{(i)}, \mathbf{m}\rangle \, T \cdot D_{\mathrm{KL}}\Big(q_{\alpha_{\mathcal{F}}}(\mathbf{z}_s^{(i)} \mid \mathbf{z}_t^{(i)} = \mathbf{m}, \mathbf{x}) \,\big\|\, p_{\theta,\hat{\alpha}_{\mathcal{F}}}(\mathbf{z}_s^{(i)} \mid \mathbf{z}_t, \mathbf{z}_t^{(i)} = \mathbf{m})\Big)\Big]\right] \quad (35)$$

$$= \mathbb{E}_{t \in \{\frac{2}{T+1}, \frac{3}{T+1}, \ldots, 1\}}\left[ \mathbb{E}_{q_{\alpha_{\mathcal{F}}}}\Big[ \sum_{i=1}^{L} \langle \mathbf{z}_t^{(i)}, \mathbf{m}\rangle \, T \cdot D_{\mathrm{KL}}\Big(q_{\alpha_{\mathcal{F}}}(\mathbf{z}_s^{(i)} \mid \mathbf{z}_t^{(i)} = \mathbf{m}, \mathbf{x}) \,\big\|\, p_{\theta,\hat{\alpha}_{\mathcal{F}}}(\mathbf{z}_s^{(i)} \mid \mathbf{z}_t, \mathbf{z}_t^{(i)} = \mathbf{m})\Big)\Big]\right] \quad (36)$$

We will now transform $T \cdot D_{\mathrm{KL}}\Big(q_{\alpha_{\mathcal{F}}}(\mathbf{z}_s^{(i)} \mid \mathbf{z}_t^{(i)} = \mathbf{m}, \mathbf{x}) \,\big\|\, p_{\theta,\hat{\alpha}_{\mathcal{F}}}(\mathbf{z}_s^{(i)} \mid \mathbf{z}_t, \mathbf{z}_t^{(i)} = \mathbf{m})\Big)$ into tractable loss when $T \to \infty$ that corresponds to continuous time. Define $A_T, \hat{A}_T$ as follows:

$$A_T(u,t) := \frac{\alpha_{\mathcal{F}}^{(i)}(u, t-\Delta) - \alpha_{\mathcal{F}}^{(i)}(u,t)}{\Delta(1 - \alpha_{\mathcal{F}}^{(i)}(u,t))}, \qquad \hat{A}_T(\hat{u},t) := \frac{\hat{\alpha}_{\mathcal{F}}^{(i)}(\hat{u}, t-\Delta) - \hat{\alpha}_{\mathcal{F}}^{(i)}(\hat{u},t)}{\Delta(1 - \hat{\alpha}_{\mathcal{F}}^{(i)}(\hat{u},t))},$$

where $\Delta = \frac{1}{T+1}$. Since $\alpha_{\mathcal{F}}^{(i)}, \hat{\alpha}_{\mathcal{F}}^{(i)}$ is $AC([0,1])$ by definition,

$$\alpha_{\mathcal{F}}^{(i)}(u, t-\Delta) - \alpha_{\mathcal{F}}^{(i)}(u,t) = -\int_{t-\Delta}^{t} \partial_r \alpha_{\mathcal{F}}^{(i)}(u,r)dr \quad \Rightarrow \quad \lim_{T \to \infty} \frac{\alpha_{\mathcal{F}}^{(i)}(u, t-\Delta) - \alpha_{\mathcal{F}}^{(i)}(u,t)}{\Delta} = -\partial_t \alpha_{\mathcal{F}}^{(i)}(u,t) \quad \text{a.e.}$$

such that

$$\lim_{T \to \infty} A_T^{(i)}(u, t) = A^{(i)}(u, t) = \frac{-\partial_t \alpha_{\mathcal{F}}^{(i)}(u, t)}{1 - \alpha_{\mathcal{F}}^{(i)}(u, t)}, \qquad t \in (0, 1] \tag{37}$$

and also $\lim_{T \to \infty} \hat{A}_T^{(i)}(\hat{u}, t) = \hat{A}^{(i)}(\hat{u}, t)$.

**First term** expansion of Eq. 29. The first term of Eq. 29 becomes

$$\frac{\alpha_{\mathcal{F}}^{(i)}(u, s) - \alpha_{\mathcal{F}}^{(i)}(u, t)}{1 - \alpha_{\mathcal{F}}^{(i)}(u, t)} \log \frac{(\alpha_{\mathcal{F}}^{(i)}(u, s) - \alpha_{\mathcal{F}}^{(i)}(u, t))(1 - \hat{\alpha}_{\mathcal{F}}^{(i)}(\hat{u}, t))}{(1 - \alpha_{\mathcal{F}}^{(i)}(u, t))(\hat{\alpha}_{\mathcal{F}}^{(i)}(\hat{u}, s) - \hat{\alpha}_{\mathcal{F}}^{(i)}(\hat{u}, t))\langle \mathbf{x}_\theta^{(i)}(\mathbf{z}_t, t), \mathbf{x}^{(i)} \rangle} \tag{38}$$

$$= \Delta A_T^{(i)}(u, t) \Big( \log A_T^{(i)}(u, t) - \log \hat{A}_T^{(i)}(\hat{u}, t) - \log \langle \mathbf{x}_\theta^{(i)}(\mathbf{z}_t, t), \mathbf{x}^{(i)} \rangle \Big). \tag{39}$$

**Second term** expansion of Eq. 29. Similarly, the second term of Eq. 29 becomes

$$\frac{1 - \alpha_{\mathcal{F}}^{(i)}(u, s)}{1 - \alpha_{\mathcal{F}}^{(i)}(u, t)} \log \frac{(1 - \alpha_{\mathcal{F}}^{(i)}(u, s))(1 - \hat{\alpha}_{\mathcal{F}}^{(i)}(\hat{u}, t))}{(1 - \alpha_{\mathcal{F}}^{(i)}(u, t))(1 - \hat{\alpha}_{\mathcal{F}}^{(i)}(\hat{u}, s))} = (1 - \Delta A_T^{(i)}(u, t)) \Big( \log(1 - \Delta A_T^{(i)}(u, t)) - \log(1 - \Delta \hat{A}_T^{(i)}(\hat{u}, t)) \Big) \tag{40}$$

where

$$\log(1 - \Delta A_T^{(i)}(u, t)) - \log(1 - \Delta \hat{A}_T^{(i)}(\hat{u}, t) = -\frac{\Delta}{1 - \Delta \zeta_T}(A_T^{(i)}(u, t) - \hat{A}_T^{(i)}(\hat{u}, t)), \qquad \zeta_T \in (A_T^{(i)}(u, t), \hat{A}_T^{(i)}(\hat{u}, t)), \tag{41}$$

holds by the mean value theorem.

**Combining both parts.** With $T \to \infty$ such that $\Delta = \frac{1}{T+1} \to 0+$, summing Eq. 37, Eq. 39, Eq. 40, and Eq. 41, the infinitesimal KL for token $i$ becomes

$$\lim_{T \to \infty} T \cdot D_{\mathrm{KL}}\Big( q_{\alpha_{\mathcal{F}}}(\mathbf{z}_s^{(i)} \mid \mathbf{z}_t^{(i)} = \mathbf{m}, \mathbf{x}) \, \big\| \, p_{\theta, \hat{\alpha}_{\mathcal{F}}}(\mathbf{z}_s^{(i)} \mid \mathbf{z}_t, \mathbf{z}_t^{(i)} = \mathbf{m}) \Big) \tag{42}$$

$$= \lim_{T \to \infty} (\Delta \cdot T) \cdot \frac{1}{\Delta} D_{\mathrm{KL}}\Big( q_{\alpha_{\mathcal{F}}}(\mathbf{z}_s^{(i)} \mid \mathbf{z}_t^{(i)} = \mathbf{m}, \mathbf{x}) \, \big\| \, p_{\theta, \hat{\alpha}_{\mathcal{F}}}(\mathbf{z}_s^{(i)} \mid \mathbf{z}_t, \mathbf{z}_t^{(i)} = \mathbf{m}) \Big) \tag{43}$$

$$= -A^{(i)} \log \langle \mathbf{x}_\theta^{(i)}(\mathbf{z}_t, t), \mathbf{x}^{(i)} \rangle + A^{(i)}(\log A^{(i)} - \log \hat{A}^{(i)}) - (A^{(i)} - \hat{A}^{(i)}) \Big). \tag{44}$$

**Final loss form.** By Eq. 36 and Eq. 44, $\mathcal{L}_{\text{diffusion}}^\infty = \lim_{T \to \infty} \mathcal{L}_{\text{diffusion}}^T$ gives:

$$\mathcal{L}_{\text{diffusion}}^\infty = \lim_{T \to \infty} \mathbb{E}_{t \in \{\frac{2}{T+1}, \frac{3}{T+1}, \dots, 1\}} \left[ \mathbb{E}_{q_{\alpha_{\mathcal{F}}}} \left[ \sum_{i=1}^L \langle \mathbf{z}_t^{(i)}, \mathbf{m} \rangle \, T \cdot D_{\mathrm{KL}}\Big( q_{\alpha_{\mathcal{F}}}(\mathbf{z}_s^{(i)} \mid \mathbf{z}_t^{(i)} = \mathbf{m}, \mathbf{x}) \, \big\| \, p_{\theta, \hat{\alpha}_{\mathcal{F}}}(\mathbf{z}_s^{(i)} \mid \mathbf{z}_t, \mathbf{z}_t^{(i)} = \mathbf{m}) \Big) \right] \right]$$

$$= \lim_{\delta \to 0+} \int_\delta^1 \mathbb{E}_{q_{\alpha_{\mathcal{F}}}} \left[ \sum_{i=1}^L \langle \mathbf{z}_t^{(i)}, \mathbf{m} \rangle \Big\{ -A^{(i)} \log \langle \mathbf{x}_\theta^{(i)}(\mathbf{z}_t, t), \mathbf{x}^{(i)} \rangle + A^{(i)}(\log A^{(i)} - \log \hat{A}^{(i)}) - (A^{(i)} - \hat{A}^{(i)}) \Big) \Big\} \right] \tag{45}$$

$$= \lim_{\delta \to 0+} \int_\delta^1 \mathbb{E}_{q_{\alpha_{\mathcal{F}}}} \left[ \sum_{i=1}^L \langle \mathbf{z}_t^{(i)}, \mathbf{m} \rangle \Big\{ \underbrace{\frac{\partial_t \alpha_{\mathcal{F}}^{(i)}(u, t)}{1 - \alpha_{\mathcal{F}}^{(i)}(u, t)} \log \langle \mathbf{x}_\theta^{(i)}(\mathbf{z}_t, t), \mathbf{x}^{(i)} \rangle}_{\text{original MDLM loss}} \right. \tag{46}$$

$$\left. - \underbrace{\frac{\partial_t \alpha_{\mathcal{F}}^{(i)}(u, t)}{1 - \alpha_{\mathcal{F}}^{(i)}(u, t)} \left( \log \frac{-\partial_t \alpha_{\mathcal{F}}^{(i)}(u, t)}{1 - \alpha_{\mathcal{F}}^{(i)}(u, t)} - \log \frac{-\partial_t \hat{\alpha}_{\mathcal{F}}^{(i)}(\hat{u}, t)}{1 - \hat{\alpha}_{\mathcal{F}}^{(i)}(\hat{u}, t)} \right) - \left( \frac{-\partial_t \alpha_{\mathcal{F}}^{(i)}(u, t)}{1 - \alpha_{\mathcal{F}}^{(i)}(u, t)} - \frac{-\partial_t \hat{\alpha}_{\mathcal{F}}^{(i)}(\hat{u}, t)}{1 - \hat{\alpha}_{\mathcal{F}}^{(i)}(\hat{u}, t)} \right)}_{\text{is minimized when } \partial_t \alpha_{\mathcal{F}}(u, t)/(1 - \alpha_{\mathcal{F}}(u, t)) = \partial_t \hat{\alpha}_{\mathcal{F}}(\hat{u}, t)/(1 - \hat{\alpha}_{\mathcal{F}}(\hat{u}, t))} \Big\} \right] dt. \tag{47}$$

where $A^{(i)}(\log A^{(i)} - \log \hat{A}^{(i)}) - (A^{(i)} - \hat{A}^{(i)}) \geq 0$ holds and minimized to 0 when $A = \hat{A}$. Since $A^{(i)}(t), \hat{A}^{(i)}(t)$ can be unbounded as $t \to 0^+$ and the discrete-time objective samples from $t \in \{\frac{2}{T+1}, \frac{3}{T+1} \ldots, 1\}$, we write as $\lim_{\delta \to 0^+} \int_\delta^1 (\cdot) \, dt$; we explain this detail in Appendix C.6 and prove in Appendix C.7 that the integrand is integrable near $t = 0$, so the limit is well-defined and finite.

**Remark. Why** $A^{(i)}(\log A^{(i)} - \log \hat{A}^{(i)}) - (A^{(i)} - \hat{A}^{(i)}) \geq 0$**?** Fix an index $i$ and write $a := A^{(i)} > 0$ and $b := \hat{A}^{(i)} > 0$. Consider the function

$$f(a; b) := a(\log a - \log b) - (a - b) = a \log \frac{a}{b} - a + b. \tag{48}$$

Let $r := \frac{a}{b} > 0$. Then

$$f(a; b) = b(r \log r - r + 1). \tag{49}$$

Since $b > 0$, it suffices to show $g(r) := r \log r - r + 1 \geq 0$ for all $r > 0$. We have

$$g'(r) = \log r, \qquad g''(r) = \frac{1}{r} > 0 \ (r > 0), \tag{50}$$

so $g$ is strictly convex on $(0, \infty)$ and its unique minimizer satisfies $g'(r) = 0$, i.e., $r = 1$. Evaluating at $r = 1$ gives $g(1) = 0$, hence $g(r) \geq 0$ for all $r > 0$, with equality if and only if $r = 1$. Therefore $f(a; b) \geq 0$ for all $a, b > 0$, and the minimum 0 is attained if and only if $a = b$, i.e., $A^{(i)} = \hat{A}^{(i)}$.

## C.4. Prior Loss

Recall that the prior loss is given as follows:

$$\mathcal{L}_{\text{prior}} = \mathbb{E}_{q_{\alpha_{\mathcal{F}}}}[D_{\text{KL}}(q_{\alpha_{\mathcal{F}}}(\mathbf{z}_{t(T)} \mid \mathbf{x}) \| p_{\theta, \hat{\alpha}_{\mathcal{F}}}(\mathbf{z}_{t(T)}))]$$

Since $t(\tau) = (\tau + 1)/(T + 1)$, $t(T)$ becomes 1. Therefore, $p_{\theta, \hat{\alpha}_{\mathcal{F}}}(\mathbf{z}_{t(T)})$ just becomes prior distribution, that is,

$$p_{\theta, \hat{\alpha}_{\mathcal{F}}}(\mathbf{z}_{t(T)}^{(i)}) = p_{\theta, \hat{\alpha}_{\mathcal{F}}}(\mathbf{z}_1^{(i)}) = \text{Cat}(\mathbf{m}).$$

Furthermore, substituting $t = 1$ into forward process $q_{\alpha_{\mathcal{F}}}(\mathbf{z}_t^{(i)} \mid \mathbf{x}) = \text{Cat}\left(\alpha_{\mathcal{F}}^{(i)}(\cdot, t)\mathbf{x}^{(i)} + (1 - \alpha_{\mathcal{F}}^{(i)}(\cdot, t))\mathbf{m}\right)$ gives:

$$q_{\alpha_{\mathcal{F}}}(\mathbf{z}_1^{(i)} \mid \mathbf{x}) = \text{Cat}(\mathbf{m}),$$

where $\alpha_{\mathcal{F}}^{(i)}(\cdot, 1) = 0$ by the definition of free-form scheduler. Since the above equations hold for every $i$, prior loss becomes zero:

$$\mathcal{L}_{\text{prior}} = \mathbb{E}_{q_{\alpha_{\mathcal{F}}}}[D_{\text{KL}}(q_{\alpha_{\mathcal{F}}}(\mathbf{z}_{t(T)} \mid \mathbf{x}) \| p_{\theta, \hat{\alpha}_{\mathcal{F}}}(\mathbf{z}_{t(T)}))] = 0. \tag{51}$$

## C.5. Reconstruction Loss

Recall that the reconstruction loss is given as follows:

$$\mathcal{L}_{\text{reconstruction}} = \mathbb{E}_{q_{\alpha_{\mathcal{F}}}}[-\log p_{\theta, \hat{\alpha}_{\mathcal{F}}}(\mathbf{x} \mid \mathbf{z}_{t(0)})].$$

Since the expectation is conditioned on $q_{\alpha_{\mathcal{F}}}$, the sampling of $\mathbf{z}_{t(0)}$ is conducted as follows:

$$\mathbf{z}_{t(0)}^{(i)} \sim \text{Cat}\left(\alpha_{\mathcal{F}}^{(i)}(\cdot, t(0))\mathbf{x}^{(i)} + (1 - \alpha_{\mathcal{F}}^{(i)}(\cdot, t(0)))\mathbf{m}\right).$$

Since $t(0) = 1/(T + 1)$ and by definition of free-form scheduler, in the continuous time case with $T \to \infty$, $\alpha_{\mathcal{F}}$ becomes 1:

$$\lim_{T \to \infty} \alpha_{\mathcal{F}}^{(i)}(\cdot, t(0)) = \alpha_{\mathcal{F}}^{(i)}(\cdot, 0) = 1,$$

where $\alpha^{(i)}_{\mathcal{F}[\mathcal{I}]}(u, \cdot) \in AC([0,1])$ and $\alpha^{(i)}_{\mathcal{F}[\mathcal{I}]}(u, 0) = 1$ hold for all $u \in \mathcal{I}$ and $i \in [1, \ldots, L]$ by definition. Therefore, $\lim_{T \to \infty} \mathbf{z}_{t(0)} = \mathbf{x}$ holds:

$$\mathbf{z}^{(i)}_{t(0)} \sim \lim_{T \to \infty} \mathrm{Cat}\Big( \alpha^{(i)}_{\mathcal{F}}(\cdot, t(0))\mathbf{x}^{(i)} + (1 - \alpha^{(i)}_{\mathcal{F}}(\cdot, t(0)))\mathbf{m} \Big)$$

$$\Rightarrow \mathbf{z}^{(i)}_{t(0)} \sim \mathrm{Cat}(\mathbf{x}^{(i)}) \Rightarrow \mathbf{z}^{(i)}_{t(0)} = \mathbf{x}^{(i)}.$$

Furthermore, by carry-over unmasking of SUBS parametrization, $p_{\theta, \hat{\alpha}_{\mathcal{F}}}(\mathbf{x}^{(i)}|\mathbf{z}_{t(0)}) = p_{\theta, \hat{\alpha}_{\mathcal{F}}}(\mathbf{x}^{(i)}|\mathbf{x}) = 1$ for all $i \in [1, \ldots, L]$. Combining the above results, the reconstruction loss in continuous time becomes zero:

$$\mathcal{L}_{\text{reconstruction}} = \mathbb{E}_{q_{\alpha_{\mathcal{F}}}}[- \log p_{\theta, \hat{\alpha}_{\mathcal{F}}}(\mathbf{x} \mid \mathbf{z}_{t(0)})] = 0. \tag{52}$$

Note that MDLM (Sahoo et al., 2024a) and BD3LM (Arriola et al., 2025) follow same derivation process with $\alpha_{\text{mdlm}}(t(0)) = T/(T+1)$.

## C.6. Final NELBO Objective

Combining all the results derived above, the final NELBO objective in continuous time can be rewritten as:

$$- \log p_{\theta, \hat{\alpha}_{\mathcal{F}}}(\mathbf{x}) \leq \underbrace{\mathcal{L}^{\infty}_{\text{diffusion}}}_{Eq.\ 47} + \underbrace{\mathcal{L}_{\text{prior}}}_{=0\ \text{from}\ Eq.\ 51} + \underbrace{\mathcal{L}_{\text{reconstruction}}}_{=0\ \text{from}\ Eq.\ 52} \tag{53}$$

$$= \lim_{\delta \to 0+} \int_{\delta}^{1} \mathbb{E}_{q_{\alpha_{\mathcal{F}}}} \left[ \sum_{i=1}^{L} \langle \mathbf{z}^{(i)}_t, \mathbf{m} \rangle \left\{ -A^{(i)} \log \langle \mathbf{x}^{(i)}_{\theta}(\mathbf{z}_t, t), \mathbf{x}^{(i)} \rangle + A^{(i)}(\log A^{(i)} - \log \hat{A}^{(i)}) - (A^{(i)} - \hat{A}^{(i)}) \right\} \right] dt. \tag{54}$$

$$= \int_{0}^{1} \underbrace{\mathbb{E}_{q_{\alpha_{\mathcal{F}}}} \left[ \sum_{i=1}^{L} \langle \mathbf{z}^{(i)}_t, \mathbf{m} \rangle \left\{ -A^{(i)} \log \langle \mathbf{x}^{(i)}_{\theta}(\mathbf{z}_t, t), \mathbf{x}^{(i)} \rangle + A^{(i)}(\log A^{(i)} - \log \hat{A}^{(i)}) - (A^{(i)} - \hat{A}^{(i)}) \right\} \right]}_{:= \ell(t)} dt, \tag{55}$$

where $A^{(i)}(\cdot, t)$ and $\hat{A}^{(i)}(\cdot, t)$ are defined only for $t \in (0, 1]$; yet, at $t = 0$, the forward process is deterministic since $\alpha^{(i)}_{\mathcal{F}}(u, 0) = 1$ implies $q_{\alpha_{\mathcal{F}}}(\mathbf{z}_0 \mid \mathbf{x}) = \delta_{\mathbf{x}}$, so that $\langle \mathbf{z}^{(i)}_0, \mathbf{m} \rangle = 0$ almost surely for all $i \in [L]$. Hence it is natural to extend the integrand to $[0, 1]$ by setting its value at $t = 0$ to be 0 without defining $A^{(i)}(0)$ or $\hat{A}^{(i)}(0)$. However, since the coefficient $A^{(i)}(\cdot, t) = -\partial_t \alpha^{(i)}(\cdot, t)/(1 - \alpha^{(i)}(\cdot, t))$ can diverge as $t \to 0+$ when $\alpha_{\mathcal{F}}(0) = 1$, we will separately prove that $\int_0^1 \ell(t)\, dt < \infty$ in the following section.

## C.7. Finiteness of NELBO

We show under what conditions the integral over $[0, 1]$ is finite. First, we can make a reasonable assumption [2] that the output of the diffusion network $\theta$ cannot take the values $+\infty$ or $-\infty$. Furthermore, since $\mathbf{x}^{(i)}_{\theta}(\mathbf{z}, t) \in \Delta^{V+1}$ is produced by a neural network followed by a softmax head, *i.e.* $\mathbf{x}^{(i)}_{\theta}(\mathbf{z}, t) = \mathrm{Softmax}(\theta^{(i)}(\mathbf{z}, t))$, for any fixed $\theta$, we can prove that $-\log \langle \mathbf{x}^{(i)}_{\theta}(\mathbf{z}, t), \mathbf{x}^{(i)} \rangle$ is finite:

$$-\log \langle \mathbf{x}^{(i)}_{\theta}(\mathbf{z}, t), \mathbf{x}^{(i)} \rangle = -\langle \theta^{(i)}(\mathbf{z}, t), \mathbf{x}^{(i)} \rangle + \log \sum_{\mathbf{x}' \in \mathcal{X}} \exp(\langle \theta^{(i)}(\mathbf{z}, t), \mathbf{x}' \rangle) \qquad \because \text{log-softmax}$$

$$\leq -\langle \theta^{(i)}(\mathbf{z}, t), \mathbf{x}^{(i)} \rangle + \log V \max_{\mathbf{x}' \in \mathcal{X}} \exp(\langle \theta^{(i)}(\mathbf{z}, t), \mathbf{x}' \rangle)$$

$$= -\langle \theta^{(i)}(\mathbf{z}, t), \mathbf{x}^{(i)} \rangle + \max_{\mathbf{x}' \in \mathcal{X}} \langle \theta^{(i)}(\mathbf{z}, t), \mathbf{x}' \rangle + \log V$$

$$< +\infty, \qquad \forall i \in [L], \forall \mathbf{x} \in \mathcal{X}^L, \forall \mathbf{z} \in \mathcal{S}(\mathbf{x}), \forall j \in [L]\}, \forall t \in [0, 1],$$

where $\mathcal{S}(\mathbf{x})$ is set of all possible masked sequence induced from $\mathbf{x}$ (Definition B.1). Since $-\log \langle \mathbf{x}^{(i)}_{\theta}(\mathbf{z}, t), \mathbf{x}^{(i)} \rangle$ is finite, there exists uniform bound $\varepsilon_{\theta}$ for fixed $\theta$:

$$-\log \langle \mathbf{x}^{(i)}_{\theta}(\mathbf{z}, t), \mathbf{x}^{(i)} \rangle \leq \varepsilon_{\theta}, \quad \forall i \in [L], \forall \mathbf{x} \in \mathcal{X}^L, \forall \mathbf{z} \in \mathcal{S}(\mathbf{x}), \forall t \in [0, 1]. \tag{56}$$

---

[2] Since the composition of finite linear transformations and continuous activation functions results in a continuous mapping, a neural network with a finite number of neurons and finite weights will always produce a finite output for any finite input.

where $\mathcal{X}^L$ and $\mathcal{S}(\mathbf{x})$ are finite.

**Proposition C.3** (Finiteness of the OeMDM NELBO). *Assume the diffusion model output is finite everywhere (as is typical for neural networks). Fix arbitrary $u$ and $\hat{u}$, and assume that $A^{(i)}(u,t)$ and $\hat{A}^{(i)}(\hat{u},t)$ are well-defined such that there exists a constant $k \geq 1$ satisfying:*

$$\frac{A^{(i)}(u,t)}{\hat{A}^{(i)}(\hat{u},t)} \in [k^{-1}, k], \qquad \forall t \in (0,1], \ \forall i \in [L].$$

*Then the OeMDM NELBO is finite:*

$$\mathcal{L}_{\text{OeMDM}} = \int_0^1 \mathbb{E}_{q_{\alpha_{\mathcal{F}}}} \left[ \sum_{i=1}^L \langle \mathbf{z}_t^{(i)}, \mathbf{m} \rangle \left\{ -A^{(i)} \log \langle \mathbf{x}_\theta^{(i)}(\mathbf{z}_t,t), \mathbf{x}^{(i)} \rangle + A^{(i)} (\log A^{(i)} - \log \hat{A}^{(i)}) - (A^{(i)} - \hat{A}^{(i)}) \right\} \right] dt < \infty,$$

*where we omit $u$ and $\hat{u}$ in $A^{(i)}$ and $\hat{A}^{(i)}$ for brevity.*

*Proof.* To prove the finiteness of NELBO, we will upper bound integrand $\ell(t)$ with an equation without the expectation term. Before that, we recall the input domain of $\alpha_{\mathcal{F}}$ and $\hat{\alpha}_{\mathcal{F}}$. Note that $\alpha_{\mathcal{F}}$ is the scheduler for forward and true posterior, *i.e.*, $q_{\alpha_{\mathcal{F}}}(\mathbf{z}_t|\mathbf{x})$ and $q_{\alpha_{\mathcal{F}}}(\mathbf{z}_s|\mathbf{x},\mathbf{z}_t)$ respectively. This means that $\mathcal{I}$ only can include $\mathcal{X}$: when the true sequence $\mathbf{x}$ is given, every $\alpha_{\mathcal{F}}$ is fixed such that the map $r \mapsto \alpha_{\mathcal{F}}^{(i)}(u,r)$ is evaluated. Furthermore, $\hat{\alpha}_{\mathcal{F}}$ is the scheduler for parametrized reverse posterior, *i.e.*, $p_{\theta,\hat{\alpha}_{\mathcal{F}}}(\mathbf{z}_s|\mathbf{z}_t)$. This means that $\hat{\mathcal{I}}$ can include $\mathcal{Z}_t$ and $\theta$: when they are fixed, every $\hat{\alpha}_{\mathcal{F}}$ is fixed such that the map $r \mapsto \hat{\alpha}_{\mathcal{F}}^{(i)}(u,r)$ is evaluated. When deriving $\ell(t)$, we will omit $u$ term in $A$ and $\alpha_{\mathcal{F}}$ for brevity, but note that $A$ and $\alpha$ cannot be exists alone without $u$ (which can be $\mathbf{x}$). This also should be applied to $\hat{A}$ and $\hat{\alpha}_{\mathcal{F}}$, they cannot stand alone without $\hat{u}$ (which can be $\mathbf{z}$ or $\theta$). But also note that $\mathbf{x}$ and $\theta$ are actually given for $\ell(t)$ since we are measuring $\log p_{\theta,\alpha_{\mathcal{F}}}(\mathbf{x})$; so that we only consider that $\mathbf{z}$ should be given when including $\hat{A}$ and $\hat{\alpha}_{\mathcal{F}}$. Within such property, we transform $\ell(t)$ as follows:

$$\ell(t) := \mathbb{E}_{q_{\alpha_{\mathcal{F}}}} \left[ \sum_{i=1}^L \langle \mathbf{z}_t^{(i)}, \mathbf{m} \rangle \left\{ -A^{(i)} \log \langle \mathbf{x}_\theta^{(i)}(\mathbf{z}_t,t), \mathbf{x}^{(i)} \rangle + A^{(i)} (\log A^{(i)} - \log \hat{A}^{(i)}) - (A^{(i)} - \hat{A}^{(i)}) \right) \right\} \right]$$

$$= \sum_{\mathbf{z} \in \mathcal{S}(\mathbf{x})} q_{\alpha_{\mathcal{F}}}(\mathbf{z}_t = \mathbf{z}|\mathbf{x}) \left[ \sum_{i=1}^L \langle \mathbf{z}^{(i)}, \mathbf{m} \rangle \left\{ -A^{(i)} \log \langle \mathbf{x}_\theta^{(i)}(\mathbf{z},t), \mathbf{x}^{(i)} \rangle + A^{(i)} (\log A^{(i)} - \log \hat{A}^{(i)}) - (A^{(i)} - \hat{A}^{(i)}) \right) \right\} \right]$$

$$= \sum_{\mathbf{z} \in \mathcal{S}(\mathbf{x})} \prod_{j=1}^L q_{\alpha_{\mathcal{F}}}(\mathbf{z}_t^{(j)} = \mathbf{z}^{(j)}|\mathbf{x}) \qquad \qquad \because \text{ Forward process is independent for every indices}$$

$$\cdot \left[ \sum_{i=1}^L \langle \mathbf{z}^{(i)}, \mathbf{m} \rangle \left\{ -A^{(i)} \log \langle \mathbf{x}_\theta^{(i)}(\mathbf{z},t), \mathbf{x}^{(i)} \rangle + A^{(i)} (\log A^{(i)} - \log \hat{A}^{(i)}) - (A^{(i)} - \hat{A}^{(i)}) \right) \right\} \right]$$

$$= \sum_{\mathbf{z} \in \mathcal{S}(\mathbf{x})} \left[ \sum_{i=1}^L \left\{ \left( (1 - \alpha_{\mathcal{F}}^{(i)}) \langle \mathbf{z}^{(i)}, \mathbf{m} \rangle \right) \cdot \left( \prod_{j \neq i} q_{\alpha_{\mathcal{F}}}(\mathbf{z}_t^{(j)} = \mathbf{z}^{(j)}|\mathbf{x}) \right) \right. \right.$$

$$\left. \left. \left( -A^{(i)} \log \langle \mathbf{x}_\theta^{(i)}(\mathbf{z},t), \mathbf{x}^{(i)} \rangle + A^{(i)} (\log A^{(i)} - \log \hat{A}^{(i)}) - (A^{(i)} - \hat{A}^{(i)}) \right) \right) \right\} \right]$$

$$\leq \sum_{\mathbf{z} \in \mathcal{S}(\mathbf{x})} \left[ \underbrace{\sum_{i=1}^L (1 - \alpha_{\mathcal{F}}^{(i)}) \left\{ -A^{(i)} \log \langle \mathbf{x}_\theta^{(i)}(\mathbf{z},t), \mathbf{x}^{(i)} \rangle + A^{(i)} (\log A^{(i)} - \log \hat{A}^{(i)}) - (A^{(i)} - \hat{A}^{(i)}) \right\}}_{:=\hat{\ell}(\mathbf{z},t)} \right].$$

We upper bound $\ell(t)$ with $\mathbf{z}^* = \arg\max_{\mathbf{z} \in \mathcal{S}(\mathbf{x})} \int_0^1 \hat{\ell}(\mathbf{z},t) dt$ as follows:

$$-\log p_{\theta,\alpha_{\mathcal{F}}}(\mathbf{x}) \leq \int_0^1 \ell(t) dt \leq \int_0^1 \sum_{\mathbf{z} \in \mathcal{S}(\mathbf{x})} \hat{\ell}(\mathbf{z},t) dt \leq 2^L \int_0^1 \hat{\ell}(\mathbf{z}^*,t) dt. \tag{57}$$

Now our goal is to show that $\int_0^1 \hat{\ell}(\mathbf{z}^*, t) dt$ is finite. Divide $\hat{\ell}(\mathbf{z}^*, t)$ as follows:

$$\hat{\ell}_1(\mathbf{z}^*, t) = -\sum_{i=1}^{L} (1 - \alpha_{\mathcal{F}}^{(i)}) A^{(i)} \log \langle \mathbf{x}_\theta^{(i)}(\mathbf{z}, t), \mathbf{x}^{(i)} \rangle dt,$$

$$\hat{\ell}_2(\mathbf{z}^*, t) = \sum_{i=1}^{L} (1 - \alpha_{\mathcal{F}}^{(i)}) \Big( A^{(i)} (\log A^{(i)} - \log \hat{A}^{(i)}) - (A^{(i)} - \hat{A}^{(i)}) \Big) dt$$

such that $\hat{\ell}(\mathbf{z}^*, t) = \hat{\ell}_1(\mathbf{z}^*, t) + \hat{\ell}_2(\mathbf{z}^*, t)$ We first show that $\int_0^1 \hat{\ell}_1(\mathbf{z}^*, t) dt$ is finite:

$$\int_{t=0}^{1} \sum_{i=1}^{L} -(1 - \alpha_{\mathcal{F}}^{(i)}) A^{(i)} \log \langle \mathbf{x}_\theta^{(i)}(\mathbf{z}^*, t), \mathbf{x}^{(i)} \rangle dt = \sum_{i=1}^{L} \int_{t=0}^{1} -(1 - \alpha_{\mathcal{F}}^{(i)}) A^{(i)} \log \langle \mathbf{x}_\theta^{(i)}(\mathbf{z}^*, t), \mathbf{x}^{(i)} \rangle dt$$

$$= \sum_{i=1}^{L} \int_{\alpha_{\mathcal{F}}^{(i)}=1}^{0} \log \langle \mathbf{x}_\theta^{(i)}(\mathbf{z}^*, t), \mathbf{x}^{(i)} \rangle d\alpha_{\mathcal{F}}^{(i)} = \sum_{i=1}^{L} \int_{\alpha_{\mathcal{F}}^{(i)}=0}^{1} -\log \langle \mathbf{x}_\theta^{(i)}(\mathbf{z}^*, t), \mathbf{x}^{(i)} \rangle d\alpha_{\mathcal{F}}^{(i)} \le L\epsilon_\theta \qquad (58)$$

We now show that $\int_0^1 \hat{\ell}_2(\mathbf{z}^*, t) \, dt$ is finite. Fix $i \in [L]$. For $t \in (0, 1]$, define

$$A^{(i)}(t) := -\frac{\partial_t \alpha_{\mathcal{F}}^{(i)}(u, t)}{1 - \alpha_{\mathcal{F}}^{(i)}(u, t)}, \qquad \hat{A}^{(i)}(t) := -\frac{\partial_t \hat{\alpha}_{\mathcal{F}}^{(i)}(\hat{u}, t)}{1 - \hat{\alpha}_{\mathcal{F}}^{(i)}(\hat{u}, t)},$$

and

$$r^{(i)}(t) := \frac{A^{(i)}(t)}{\hat{A}^{(i)}(t)} > 0, \qquad \kappa(r) := r \log r - (r - 1) \ge 0.$$

Then for $t \in (0, 1]$,

$$(1 - \alpha_{\mathcal{F}}^{(i)}(u, t)) \Big( A^{(i)} (\log A^{(i)} - \log \hat{A}^{(i)}) - (A^{(i)} - \hat{A}^{(i)}) \Big) = (1 - \alpha_{\mathcal{F}}^{(i)}(u, t)) \hat{A}^{(i)}(t) \kappa(r^{(i)}(t)).$$

By the assumption, for fixed $u$ and $\hat{u}$, there exists $k \ge 1$ such that $r^{(i)}(t) \in [k^{-1}, k]$ for all $t \in (0, 1]$. Since the map $r \mapsto \log r - 1 + \frac{1}{r}$ is continuous on $(0, \infty)$ and $r$ is bounded on $[k^{-1}, k]$; let

$$M := \sup_{r \in [k^{-1}, k]} \left( \log r - 1 + \frac{1}{r} \right) < \infty.$$

Using $A^{(i)}(t) = r^{(i)}(t) \hat{A}^{(i)}(t)$, we have for all $t \in (0, 1]$,

$$0 \le (1 - \alpha_{\mathcal{F}}^{(i)}(u, t)) \hat{A}^{(i)}(t) \kappa(r^{(i)}(t))$$

$$= (1 - \alpha_{\mathcal{F}}^{(i)}(u, t)) A^{(i)}(t) \frac{\kappa(r^{(i)}(t))}{r^{(i)}(t)}$$

$$= (-(\partial_t \alpha_{\mathcal{F}}^{(i)}(u, t))) \left( \log r^{(i)}(t) - 1 + \frac{1}{r^{(i)}(t)} \right)$$

$$\le M(-\partial_t \alpha_{\mathcal{F}}^{(i)}(u, t)),$$

where we used $(1 - \alpha_{\mathcal{F}}^{(i)}(u, t)) A^{(i)}(t) = -\partial_t \alpha_{\mathcal{F}}^{(i)}(u, t)$. Since $\alpha_{\mathcal{F}}^{(i)}(u, \cdot) \in AC([0, 1])$ with $\alpha_{\mathcal{F}}^{(i)}(u, 0) = 1$ and $\alpha_{\mathcal{F}}^{(i)}(u, 1) = 0$, the fundamental theorem of calculus for absolutely continuous functions yields

$$\int_0^1 -\partial_t \alpha_{\mathcal{F}}^{(i)}(u, t) \, dt = \alpha_{\mathcal{F}}^{(i)}(u, 0) - \alpha_{\mathcal{F}}^{(i)}(u, 1) = 1.$$

Therefore,

$$\int_0^1 (1 - \alpha_{\mathcal{F}}^{(i)}(u, t)) \hat{A}^{(i)}(t) \kappa(r^{(i)}(t)) \, dt \le M \int_0^1 -\partial_t \alpha_{\mathcal{F}}^{(i)}(u, t) \, dt = M < \infty. \qquad (59)$$

Finally, by Eq. 57–Eq. 59, the continuous-time NELBO of OeMDM is finite on $t \in [0, 1]$ provided that there exists $k \ge 1$ such that $r^{(i)}(t) \in [k^{-1}, k]$ for all $t \in (0, 1]$ and all $i \in [L]$. $\qquad \square$

This condition should not be viewed as an artificial assumption introduced solely for the proof; rather, it provides practical intuition on how schedulers should be designed so that the NELBO remains a valid (finite) training objective for masked diffusion models. For example, under the linear scheduler of MDLM, both $\alpha$ and $\hat{\alpha}$ take the form $1 - t$, so $r^{(i)}(t) \equiv 1$ and the finiteness of the NELBO follows immediately. More broadly, when constructing input-dependent schedulers, this condition highlights a potential pitfall: if $\alpha$ and $\hat{\alpha}$ are parameterized by overly different function classes so that $r^{(i)}(t)$ cannot be uniformly controlled, then finiteness of the NELBO is no longer guaranteed. In this sense, the bounded-ratio condition serves as a mild regularity guideline for scheduler parameterizations that are expressive yet remain compatible with stable NELBO-based training.

**Remark. Why NELBO of LoMDM is finite?** Recall that the true posterior and reverse velocity of LoMDM are given as follows:

$$A_\phi^{(i)}(\mathbf{x}, t) = \frac{c_1 + c_2 \cdot [\text{NormSig}(g_\phi(f(\mathbf{x})))]_i}{t}, \qquad \hat{A}_\psi^{(i)}(\mathbf{z}_t, t) = \frac{c_1 + c_2 \cdot [\text{NormSig}(g_\psi(f(\mathbf{z}_t)))]_i}{t},$$

where $c_1 > c_2$. Therefore, the inequality

$$\frac{c_1 - c_2}{c_1 + c_2} \leq \frac{A_\phi^{(i)}(\mathbf{x}, t)}{\hat{A}_\psi^{(i)}(\mathbf{z}_t, t)} \leq \frac{c_1 + c_2}{c_1 - c_2}$$

always holds for any $\mathbf{x} \in \mathcal{X}^L$ and $\mathbf{z}_t \in \mathcal{S}(\mathbf{x})$, such that NELBO of LoMDM is always finite by Proposition C.3.

## D. Other Proofs

### D.1. OeMDM Can Express Autoregerssive Models

In this section, we provide the complete proof of Proposition 3.3 omitted in the main paper. We first define the ARM scheduler that satisfies the statements in Proposition 3.3, and further provide the definition of the trajectory sets and one lemma required for the proof.

**Definition D.1** (ARM scheduler $\alpha_{\text{arm},\varepsilon}$). Fix $L \in \mathbb{N}$ and define time windows

$$t_i^{\text{start}} := 1 - \frac{i}{L}, \qquad t_i^{\text{end}} := 1 - \frac{i-1}{L}, \qquad \Delta := t_i^{\text{end}} - t_i^{\text{start}} = \frac{1}{L}, \quad i \in [L]. \tag{60}$$

Let $S : \mathbb{R} \to [0, 1]$ be the $C^1$ smoothstep

$$S(u) := \begin{cases} 0, & u \leq 0, \\ 3u^2 - 2u^3, & 0 < u < 1, \\ 1, & u \geq 1. \end{cases} \tag{61}$$

For $\varepsilon \in (0, 1)$, define $\alpha_{\text{arm},\varepsilon} \in \mathcal{F}[\emptyset]$ coordinate-wise by

$$\alpha_{\text{arm},\varepsilon}^{(i)}(t) := 1 - \varepsilon t - (1 - \varepsilon) S\left(\frac{t - t_i^{\text{start}}}{\Delta}\right), \qquad t \in [0, 1], \; i \in [L]. \tag{62}$$

**Definition D.2** (Autoregressive trajectory sets conditioned on $\mathbf{x}$). Fix a target sequence $\mathbf{x} \in \mathcal{X}^L$. Fix $T \gg L$ and a time grid $t(\tau) = (\tau + 1)/(T + 1)$. For each $i \in [L]$, define the window-index set

$$\mathcal{W}_i := \{\tau \in [T] : t(\tau) \in [t_i^{\text{start}}, t_i^{\text{end}}]\}, \qquad \tau_i^{\text{start}} := \min \mathcal{W}_i, \quad \tau_i^{\text{end}} := \max \mathcal{W}_i,$$

Define the autoregressive trajectory set as the subset of $\mathcal{S}_{\text{absorb}}(\mathbf{x}; T)$ (Definition B.2), whose masking times fall inside the designated windows:

$$\mathcal{S}_{\text{arm}}(\mathbf{x}, T) := \Big\{ (\mathbf{z}_{t(0)}, \ldots, \mathbf{z}_{t(T)}) \in \mathcal{S}_{\text{absorb}}(\mathbf{x}; L, T) \Big| \exists (\kappa_i)_{i=2}^L \text{ with } \kappa_i \in \mathcal{W}_i \text{ s.t.}$$
$$\mathbf{z}_{t(\tau)}^{(i)} = \mathbf{x}^{(i)} \mathbb{1}\{\tau < \kappa_i\} + \mathbf{m} \mathbb{1}\{\tau \geq \kappa_i\}, \; \forall i \in [L], \; \forall \tau \in \{0, \ldots, T\} \Big\}. \tag{63}$$

That is, each coordinate $i \in [2, \ldots, L]$ switches from $\mathbf{x}^{(i)}$ to $\mathbf{m}$ exactly once, and the switch index lies within its own window. For $i = 1$, $\mathbf{z}_{t(\tau)}^{(1)}$ may transform into $\mathbf{x}^{(1)}$ by reverse process in $\tau \in \mathcal{W}_1$ or by $p_{\theta,\alpha_{\mathrm{arm},\varepsilon}}(\mathbf{x}^{(1)} \mid \mathbf{z}_{t(0)}) = \mathrm{Cat}\Big(\mathbf{x}_\theta^{(1)}\big(\mathbf{z}_{t(0)}, t(0)\big)\Big)$. Let $\mathcal{S}_{\mathrm{rest}}(\mathbf{x}, T) := \mathcal{S}_{\mathrm{absorb}}(\mathbf{x}, T) \setminus \mathcal{S}_{\mathrm{arm}}(\mathbf{x}, T)$.

**Lemma D.3.** *In discrete-time OeMDM, for every $i \in [L]$, every grid index $\tau \in \{0, \ldots, T\}$ where $t(0) = 0 < t(1) < \cdots < t(T) = 1$, every input-agnostic free-form scheduler $\hat{\alpha}_{\mathcal{F}[\emptyset]} \in \mathcal{F}[\emptyset]$, and for generative model distribution $p_{\theta,\hat{\alpha}_{\mathcal{F}[\emptyset]}}(\mathbf{x})$ induced by reverse kernel $p_{\theta,\hat{\alpha}_{\mathcal{F}[\emptyset]}}(\mathbf{z}_s|\mathbf{z}_t)$,*

$$p_{\theta,\hat{\alpha}_{\mathcal{F}[\emptyset]}}\big(\mathbf{z}_{t(\tau)}^{(i)} \neq \mathbf{m}\big) = \hat{\alpha}_{\mathcal{F}[\emptyset]}^{(i)}(t(\tau)). \tag{64}$$

*Proof.* At time $t(T) = 1$, the prior is $\mathrm{Cat}(\mathbf{m})$ at each coordinate, hence $p_{\theta,\hat{\alpha}_{\mathcal{F}[\emptyset]}}\big(\mathbf{z}_{t(T)}^{(i)} \neq \mathbf{m}\big) = 0$. Also, by the boundary condition of a free-form scheduler, $\hat{\alpha}_{\mathcal{F}[\emptyset]}^{(i)}(t(T)) = \hat{\alpha}_{\mathcal{F}[\emptyset]}^{(i)}(1) = 0$. Therefore the claim holds for $\tau = T$. Assume the equality holds at some $\tau \in \{1, \ldots, T\}$, *i.e.*, $p_{\theta,\hat{\alpha}_{\mathcal{F}[\emptyset]}}\big(\mathbf{z}_{t(\tau)}^{(i)} \neq \mathbf{m}\big) = \hat{\alpha}_{\mathcal{F}[\emptyset]}^{(i)}(t(\tau))$. We prove it for $\tau - 1$:

$$\begin{aligned}
p_{\theta,\hat{\alpha}_{\mathcal{F}[\emptyset]}}\big(\mathbf{z}_{t(\tau-1)}^{(i)} \neq \mathbf{m}\big) &= p_{\theta,\hat{\alpha}_{\mathcal{F}[\emptyset]}}\big(\mathbf{z}_{t(\tau)}^{(i)} \neq \mathbf{m}\big) + p_{\theta,\hat{\alpha}_{\mathcal{F}[\emptyset]}}\big(\mathbf{z}_{t(\tau)}^{(i)} = \mathbf{m}\big) p_{\theta,\hat{\alpha}_{\mathcal{F}[\emptyset]}}\big(\mathbf{z}_{t(\tau-1)}^{(i)} \neq \mathbf{m} \mid \mathbf{z}_{t(\tau)}^{(i)} = \mathbf{m}\big) \\
&= p_{\theta,\hat{\alpha}_{\mathcal{F}[\emptyset]}}\big(\mathbf{z}_{t(\tau)}^{(i)} \neq \mathbf{m}\big) + \Big(1 - p_{\theta,\hat{\alpha}_{\mathcal{F}[\emptyset]}}\big(\mathbf{z}_{t(\tau)}^{(i)} \neq \mathbf{m}\big)\Big) p_{\theta,\hat{\alpha}_{\mathcal{F}[\emptyset]}}\big(\mathbf{z}_{t(\tau-1)}^{(i)} \neq \mathbf{m} \mid \mathbf{z}_{t(\tau)}^{(i)} = \mathbf{m}\big) \\
&= p_{\theta,\hat{\alpha}_{\mathcal{F}[\emptyset]}}\big(\mathbf{z}_{t(\tau)}^{(i)} \neq \mathbf{m}\big) + \Big(1 - p_{\theta,\hat{\alpha}_{\mathcal{F}[\emptyset]}}\big(\mathbf{z}_{t(\tau)}^{(i)} \neq \mathbf{m}\big)\Big) \frac{\hat{\alpha}_{\mathcal{F}[\emptyset]}^{(i)}(t(\tau-1)) - \hat{\alpha}_{\mathcal{F}[\emptyset]}^{(i)}(t(\tau))}{1 - \hat{\alpha}_{\mathcal{F}[\emptyset]}^{(i)}(t(\tau))}
\end{aligned}$$

Using the induction hypothesis $p_{\theta,\hat{\alpha}_{\mathcal{F}[\emptyset]}}\big(\mathbf{z}_{t(\tau)}^{(i)} \neq \mathbf{m}\big) = \hat{\alpha}_{\mathcal{F}[\emptyset]}^{(i)}(t(\tau))$, we obtain

$$\begin{aligned}
p_{\theta,\hat{\alpha}_{\mathcal{F}[\emptyset]}}\big(\mathbf{z}_{t(\tau-1)}^{(i)} \neq \mathbf{m}\big) &= \hat{\alpha}_{\mathcal{F}[\emptyset]}^{(i)}(t(\tau)) + \Big(1 - \hat{\alpha}_{\mathcal{F}[\emptyset]}^{(i)}(t(\tau))\Big) \frac{\hat{\alpha}_{\mathcal{F}[\emptyset]}^{(i)}(t(\tau-1)) - \hat{\alpha}_{\mathcal{F}[\emptyset]}^{(i)}(t(\tau))}{1 - \hat{\alpha}_{\mathcal{F}[\emptyset]}^{(i)}(t(\tau))} \\
&= \hat{\alpha}_{\mathcal{F}[\emptyset]}^{(i)}(t(\tau-1)).
\end{aligned}$$

This completes the backward induction, and the claim holds for every $\tau \in \{0, \ldots, T\}$. $\qquad\square$

We now proceed to the proof of Proposition 3.3. Throughout the proof, we omit $(\mathbf{x}, T)$ in $\mathcal{S}_{\mathrm{arm}}(\mathbf{x}, T)$, $\mathcal{S}_{\mathrm{absorb}}(\mathbf{x}, T)$, and $\mathcal{S}_{\mathrm{rest}}(\mathbf{x}, T)$ for brevity.

**Proposition 3.3** (Autoregressive models as a special case of OeMDM). *If $\mathbf{x}_\theta$ is time-agnostic as typical ARMs, there exists $\alpha_{\mathrm{arm},\epsilon} \in \mathcal{F}[\emptyset]$ that makes $p_{\theta,\hat{\alpha}_{\mathcal{F}}}$ becomes approximately equal to ARMs. Formally, the generative distribution induced by the reverse kernel $p_{\theta,\alpha_{\mathrm{arm},\epsilon}}(\mathbf{z}_s|\mathbf{z}_t)$ satisfies:*

$$p_{\theta,\alpha_{\mathrm{arm},\epsilon}}(\mathbf{x}) = \prod_{i=1}^{L} \langle \mathbf{x}_\theta^{(i)}(\mathbf{y}_i), \mathbf{x}^{(i)} \rangle + O(\epsilon),$$

*where $\mathbf{y}_i = [\mathbf{x}^{(1:i-1)} : \mathbf{m}^{L-i+1}]$. In continuous-time,*

$$\mathcal{L}_{\mathrm{OeMDM}}(\mathbf{x}, \theta, \alpha_{\mathrm{arm},\epsilon}, \alpha_{\mathrm{arm},\epsilon}) = -\log \prod_{i=1}^{L} \langle \mathbf{x}_\theta^{(i)}(\mathbf{y}_i), \mathbf{x}^{(i)} \rangle + O(\epsilon),$$

*such that OeMDM converges to ARM closely as $\epsilon \to 0+$.*

*Proof of the first statement:* $p_{\theta,\alpha_{arm,\epsilon}}(\mathbf{x}) = \prod_{i=1}^{L} \langle \mathbf{x}_\theta^{(i)}(\mathbf{y}_i), \mathbf{x}^{(i)} \rangle + O(\epsilon)$.

Define $s(\tau) := t(\tau-1)$ for $\tau \in [T]$, so each reverse step is $(t(\tau) \to s(\tau))$. The path distribution yields

$$p_{\theta,\alpha_{\mathrm{arm},\varepsilon}}(\mathbf{x}) = \sum_{\mathbf{z}_{t(0:T)} \in \mathcal{S}_{\mathrm{absorb}}} p_{\theta,\alpha_{\mathrm{arm},\varepsilon}}(\mathbf{x} \mid \mathbf{z}_{t(0)}) \Big(\prod_{\tau=1}^{T} p_{\theta,\alpha_{\mathrm{arm},\varepsilon}}(\mathbf{z}_{s(\tau)} \mid \mathbf{z}_{t(\tau)})\Big) p_{\theta,\alpha_{\mathrm{arm},\varepsilon}}(\mathbf{z}_{t(T)}). \tag{65}$$

Under $\alpha_{\mathrm{arm},\varepsilon}$ we have $\alpha_{\mathrm{arm},\varepsilon}^{(i)}(1) = 0$, hence $\mathbf{z}_{t(T)} = \mathbf{m}^L$ deterministically. Therefore $p_{\theta,\alpha_{\mathrm{arm},\varepsilon}}(\mathbf{z}_{t(T)}) = 1$, and Eq. 65 simplifies to

$$p_{\theta,\alpha_{\mathrm{arm},\varepsilon}}(\mathbf{x}) = \sum_{\mathbf{z}_{t(0:T)} \in \mathcal{S}_{\mathrm{absorb}}} p_{\theta,\alpha_{\mathrm{arm},\varepsilon}}(\mathbf{x} \mid \mathbf{z}_{t(0)}) \prod_{\tau=1}^{T} p_{\theta,\alpha_{\mathrm{arm},\varepsilon}}(\mathbf{z}_{s(\tau)} \mid \mathbf{z}_{t(\tau)}) \tag{66}$$

$$= \sum_{\mathbf{z}_{t(0:T)} \in \mathcal{S}_{\mathrm{arm}}} p_{\theta,\alpha_{\mathrm{arm},\varepsilon}}(\mathbf{x} \mid \mathbf{z}_{t(0)}) \prod_{\tau=1}^{T} p_{\theta,\alpha_{\mathrm{arm},\varepsilon}}(\mathbf{z}_{s(\tau)} \mid \mathbf{z}_{t(\tau)}) + R_\varepsilon, \tag{67}$$

where $\mathbf{z}_{t(T)} = \mathbf{m}^L$ and

$$R_\varepsilon := \sum_{\mathbf{z}_{t(0:T)} \in \mathcal{S}_{\mathrm{rest}}} p_{\theta,\alpha_{\mathrm{arm},\varepsilon}}(\mathbf{x} \mid \mathbf{z}_{t(0)}) \prod_{\tau=1}^{T} p_{\theta,\alpha_{\mathrm{arm},\varepsilon}}(\mathbf{z}_{s(\tau)} \mid \mathbf{z}_{t(\tau)}). \tag{68}$$

Introduce the reverse-time path measure $\mathbb{P}_{\theta,\varepsilon}$ induced by the Markov chain $\mathbf{z}_{t(T)} \to \mathbf{z}_{t(T-1)} \to \cdots \to \mathbf{z}_{t(0)} \to \mathbf{x}$ with transitions $p_{\theta,\alpha_{\mathrm{arm},\varepsilon}}(\mathbf{z}_{s(\tau)} \mid \mathbf{z}_{t(\tau)})$, $p_{\theta,\alpha_{\mathrm{arm},\varepsilon}}(\mathbf{x}^{(i)} \mid \mathbf{z}_{t(0)}) = \mathrm{Cat}\left(\mathbf{x}_\theta^{(i)}(\mathbf{z}_{t(0)}, t(0))\right)$, and initialization $\mathbf{z}_{t(T)} = \mathbf{m}^L$. By the definition of a path probability,

$$R_\varepsilon \leq \mathbb{P}_{\theta,\varepsilon}\Big((\mathbf{z}_{t(0)}, \ldots, \mathbf{z}_{t(T)}, \mathbf{x}) \in \{\mathbf{x}, \mathbf{z}_{t(0:T)} | \mathbf{x} \in \mathcal{V}^L, \mathbf{z}_{t(0:T)} \in \mathcal{S}_{\mathrm{rest}}(\mathbf{x}, L, T)\}\Big). \tag{69}$$

Note that Markov chain is formed by reversing the time interval through $t(T) \to t(T-1) \to \ldots t(0)$. Then, for each $i \in [L-1]$, define the "late" event that position $i$ remains masked after the start of its designated window, and the "early" event that some strictly-right position $i$ is unmasked before the end of the $i$-th window:

$$\mathcal{E}_i^{\mathrm{late}} := \left\{\mathbf{z}_{t(\tau_i^{\mathrm{start}}-1)}^{(i)} = \mathbf{m}\right\}, \qquad \mathcal{E}_i^{\mathrm{early}} := \left\{\mathbf{z}_{t(\tau_i^{\mathrm{end}}+1)}^{(i)} \neq \mathbf{m}\right\}. \tag{70}$$

If a trajectory lies in $\mathcal{S}_{\mathrm{rest}} = \mathcal{S}_{\mathrm{absorb}} \setminus \mathcal{S}_{\mathrm{arm}}$, then either some $i$ is remains masked after the start of its own window (i.e., $\mathcal{E}_i^{\mathrm{late}}$), or some $i$ is unmasked before the end of $i$-th window (i.e., $\mathcal{E}_i^{\mathrm{early}}$). Consequently,

$$\{\mathbf{x}, \mathbf{z}_{t(0:T)} | \mathbf{x} \in \mathcal{V}^L, \mathbf{z}_{t(0:T)} \in \mathcal{S}_{\mathrm{rest}}(\mathbf{x}, T)\} \subseteq \bigcup_{i=1}^{L-1} \left(\mathcal{E}_{i+1}^{\mathrm{late}} \cup \mathcal{E}_i^{\mathrm{early}}\right), \tag{71}$$

where the late event only occurs in $i \in [2, \ldots, L]$ and the early event only occurs in $i \in [1, \ldots, L-1]$. We now bound these events under $\mathbb{P}_{\theta,\varepsilon}$. By Lemma D.3 and $S(1) = 1$,

$$\mathbb{P}_{\theta,\varepsilon}(\mathcal{E}_i^{\mathrm{early}}) = \alpha_{\mathrm{arm},\varepsilon}^{(i)}(t(\tau_i^{\mathrm{end}}+1)) = 1 - \varepsilon t(\tau_i^{\mathrm{end}}+1) - (1-\varepsilon) = \varepsilon(1 - t(\tau_i^{\mathrm{end}}+1)) \leq \varepsilon.$$

Similarly, by Lemma D.3,

$$\mathbb{P}_{\theta,\varepsilon}(\mathcal{E}_i^{\mathrm{late}}) = 1 - \alpha_{\mathrm{arm},\varepsilon}^{(i)}(t(\tau_i^{\mathrm{start}}-1)) = 1 - (1 - \varepsilon t(\tau_i^{\mathrm{start}}-1)) = \varepsilon t(\tau_i^{\mathrm{start}}-1)) \leq \epsilon$$

Combining Eq. 69 and Eq. 71 with the union bound gives

$$R_\varepsilon \leq \sum_{i=1}^{L-1} \left(\mathbb{P}_{\theta,\varepsilon}(\mathcal{E}_{i+1}^{\mathrm{late}}) + \mathbb{P}_{\theta,\varepsilon}(\mathcal{E}_i^{\mathrm{early}})\right) \leq \sum_{i=1}^{L-1}(\varepsilon + \varepsilon) = O(\varepsilon),$$

where the implicit constant depends only on $L$. Hence we conclude

$$p_{\theta,\alpha_{\mathrm{arm},\varepsilon}}(\mathbf{x}) = \sum_{(\mathbf{z}_{t(0)}, \ldots, \mathbf{z}_{t(T)}) \in \mathcal{S}_{\mathrm{arm}}} p_{\theta,\alpha_{\mathrm{arm},\varepsilon}}(\mathbf{x} \mid \mathbf{z}_{t(0)}) \prod_{\tau=1}^{T} p_{\theta,\alpha_{\mathrm{arm},\varepsilon}}(\mathbf{z}_{s(\tau)} \mid \mathbf{z}_{t(\tau)}) + O(\epsilon). \tag{72}$$

Finally, since the trajectory comes from $\mathcal{S}_{\text{arm}}$, every $\mathbf{z}_t$ is a left-most-masked sequence $\mathbf{z}_t = [\mathbf{x}^{(1:k)} : \mathbf{m}^{L-k}]$. Recall that the reverse kernel of OeMDM (with $\hat{\alpha}_{\mathcal{F}} = \alpha_{\text{arm},\varepsilon}$) is coordinate-wise

$$p_{\theta,\alpha_{\text{arm},\varepsilon}}(\mathbf{z}_s^{(i)} \mid \mathbf{z}_t) = \begin{cases} \text{Cat}(\mathbf{z}_t^{(i)}), & \text{if } \mathbf{z}_t^{(i)} \neq \mathbf{m}, \\ \text{Cat}\left( \frac{(1-\alpha_{\text{arm},\varepsilon}^{(i)}(s))\mathbf{m}+(\alpha_{\text{arm},\varepsilon}^{(i)}(s)-\alpha_{\text{arm},\varepsilon}^{(i)}(t))\mathbf{x}_\theta^{(i)}(\mathbf{z}_t,t)}{1-\alpha_{\text{arm},\varepsilon}^{(i)}(t)} \right), & \text{if } \mathbf{z}_t^{(i)} = \mathbf{m}. \end{cases} \tag{73}$$

We only need to consider the step in $\mathcal{S}_{\text{arm}}$ where the *first unmasking* of position $i$ occurs. By the first case of Eq. 73, once $\mathbf{z}_t^{(i)} \neq \mathbf{m}$, the value is carried over deterministically in all subsequent steps. Hence, along any trajectory in $\mathcal{S}_{\text{arm}}$, each token $\mathbf{x}^{(i)}$ is sampled *exactly once* at the unique step where $\mathbf{z}_t^{(i)} = \mathbf{m}$ and $\mathbf{z}_s^{(i)} \neq \mathbf{m}$. Conditioned on this unmasking event, the second case of Eq. 73 implies that the induced conditional distribution over the sampled token is given by the model prediction $\mathbf{x}_\theta^{(i)}(\mathbf{z}_t,t)$ (since the $\alpha$-dependent prefactor only controls *whether* unmasking happens, not *which token* is drawn once unmasking occurs). Finally, the reconstruction distribution is given as $p_{\theta,\alpha_{\text{arm},\varepsilon}}(\mathbf{x}^{(i)} \mid \mathbf{z}_{t(0)}) = \text{Cat}\left( \mathbf{x}_\theta^{(i)}(\mathbf{z}_{t(0)}, t(0)) \right)$.

By the assumption in Proposition 3.3, $\mathbf{x}_\theta$ is time-agnostic, so $\mathbf{x}_\theta^{(i)}(\mathbf{z}_t,t) = \mathbf{x}_\theta^{(i)}(\mathbf{z}_t)$. Moreover, in $\mathcal{S}_{\text{arm}}$, right before unmasking position $i$ we necessarily have the canonical state $\mathbf{z}_t = \mathbf{y}_i = [\mathbf{x}^{(1:i-1)} : \mathbf{m}^{L-i+1}]$. Therefore,

$$\mathbb{P}\left( \mathbf{x}^{(i)} \mid \mathbf{x}^{(1:i-1)}, (\mathbf{z}_{t(0)}, \dots, \mathbf{z}_{t(T)}) \in \mathcal{S}_{\text{arm}} \right) = \langle \mathbf{x}_\theta^{(i)}(\mathbf{y}_i), \mathbf{x}^{(i)} \rangle.$$

Multiplying these one-time unmasking conditionals over $i = 1, \dots, L$ gives the probability mass assigned to $\mathbf{x}$ by trajectories in $\mathcal{S}_{\text{arm}}$, and combining with Eq. 72 yields

$$p_{\theta,\alpha_{\text{arm},\epsilon}}(\mathbf{x}) = \prod_{i=1}^{L} \langle \mathbf{x}_\theta^{(i)}(\mathbf{y}_i), \mathbf{x}^{(i)} \rangle + O(\epsilon),$$

as desired.

$\square$

*Proof of the second statement:* $\mathcal{L}_{OeMDM}(\theta, \alpha_{\mathcal{F}}, \hat{\alpha}_{\mathcal{F}}) = -\log \prod_{i=1}^{L} \langle \mathbf{x}_\theta^{(i)}(\mathbf{y}_i), \mathbf{x}^{(i)} \rangle + O(\epsilon)$. Recall that the NELBO of OeMDM is

$$\mathcal{L}_{OeMDM}(\mathbf{x}, \theta, \alpha_{\mathcal{F}}, \hat{\alpha}_{\mathcal{F}}) = \int_0^1 \mathbb{E}_{q_{\alpha_{\mathcal{F}}}} \left[ \sum_{i=1}^{L} \langle \mathbf{z}_t^{(i)}, \mathbf{m} \rangle \left\{ -A^{(i)} \log\langle \mathbf{x}_\theta^{(i)}(\mathbf{z}_t,t), \mathbf{x}^{(i)} \rangle + \mathcal{L}_{\text{velocity}} \right\} \right] dt.$$

Set $\alpha_{\mathcal{F}} = \hat{\alpha}_{\mathcal{F}} = \alpha_{\text{arm},\varepsilon}$. Then $A^{(i)} = \hat{A}^{(i)}$ for all $i$, hence $\mathcal{L}_{\text{velocity}} = 0$ and

$$\mathcal{L}_{OeMDM}(\mathbf{x}, \theta, \alpha_{\text{arm},\varepsilon}, \alpha_{\text{arm},\varepsilon}) = \int_0^1 \mathbb{E}_{q_{\alpha_{\text{arm},\varepsilon}}} \left[ \sum_{i=1}^{L} -\langle \mathbf{z}_t^{(i)}, \mathbf{m} \rangle A_{\text{arm},\varepsilon}^{(i)}(t) \log\langle \mathbf{x}_\theta^{(i)}(\mathbf{z}_t,t), \mathbf{x}^{(i)} \rangle \right] dt. \tag{74}$$

By the definition of velocity, $A^{(i)}(t) = -\frac{\partial_t \alpha^{(i)}(t)}{1-\alpha^{(i)}(t)}$; thus

$$\left(1 - \alpha_{\text{arm},\varepsilon}^{(i)}(t)\right) A_{\text{arm},\varepsilon}^{(i)}(t) = -\partial_t \alpha_{\text{arm},\varepsilon}^{(i)}(t). \tag{75}$$

Moreover, under $q_{\alpha_{\text{arm},\varepsilon}}(\mathbf{z}_t \mid \mathbf{x})$ we have

$$q_{\alpha_{\text{arm},\varepsilon}}(\mathbf{z}_t^{(i)} = \mathbf{m} \mid \mathbf{x}) = 1 - \alpha_{\text{arm},\varepsilon}^{(i)}(t), \qquad q_{\alpha_{\text{arm},\varepsilon}}(\mathbf{z}_t^{(i)} = \mathbf{x}^{(i)} \mid \mathbf{x}) = \alpha_{\text{arm},\varepsilon}^{(i)}(t),$$

and the coordinates factorize. Using conditional expectation in Eq. 74 and Eq. 75,

$$\mathcal{L}_{OeMDM}(\mathbf{x}, \theta, \alpha_{\text{arm},\varepsilon}, \alpha_{\text{arm},\varepsilon}) = \sum_{i=1}^{L} \int_0^1 \left( -\partial_t \alpha_{\text{arm},\varepsilon}^{(i)}(t) \right) \underbrace{\mathbb{E}_{q_{\alpha_{\text{arm},\varepsilon}}} \left[ -\log\langle \mathbf{x}_\theta^{(i)}(\mathbf{z}_t,t), \mathbf{x}^{(i)} \rangle \mid \mathbf{z}_t^{(i)} = \mathbf{m} \right]}_{=:g_i(t)} dt. \tag{76}$$

We now evaluate $g_i(t)$ for the ARM scheduler. Fix $i$ and take $t \in [t_i^{\text{start}}, t_i^{\text{end}}]$. For any $j < i$, we have $t < t_j^{\text{start}}$, hence $S(\frac{t - t_j^{\text{start}}}{\Delta}) = 0$ and $\alpha_{\text{arm},\varepsilon}^{(j)}(t) = 1 - \varepsilon t$, so $q(\mathbf{z}_t^{(j)} = \mathbf{m} \mid \mathbf{x}) = 1 - \alpha_{\text{arm},\varepsilon}^{(j)}(t) = \varepsilon t \leq \varepsilon$. For any $j > i$, we have $t \geq t_j^{\text{end}}$, hence $S(\frac{t - t_j^{\text{start}}}{\Delta}) = 1$ and $\alpha_{\text{arm},\varepsilon}^{(j)}(t) = \varepsilon(1 - t)$, so $q(\mathbf{z}_t^{(j)} \neq \mathbf{m} \mid \mathbf{x}) = \alpha_{\text{arm},\varepsilon}^{(j)}(t) \leq \varepsilon$. Therefore, conditioned on $\mathbf{z}_t^{(i)} = \mathbf{m}$, the event $\mathbf{z}_t = \mathbf{y}_i = [\mathbf{x}^{(1:i-1)} : \mathbf{m}^{L-i+1}]$ holds with probability $1 - O(\varepsilon)$ (union bound over $j \neq i$). Using the time-agnostic assumption $\mathbf{x}_\theta(\mathbf{z}_t, t) = \mathbf{x}_\theta(\mathbf{z}_t)$, we obtain

$$g_i(t) = -\log\langle \mathbf{x}_\theta^{(i)}(\mathbf{y}_i), \mathbf{x}^{(i)}\rangle + O(\varepsilon), \qquad t \in [t_i^{\text{start}}, t_i^{\text{end}}], \tag{77}$$

where the $O(\varepsilon)$ term is uniform in $t$ (assuming the integrand is finite, $i.e.$, $\langle \mathbf{x}_\theta^{(i)}(\cdot), \mathbf{x}^{(i)}\rangle > 0$ on the relevant states).

Outside the $i$-th window, we have $S'(\cdot) = 0$, hence $\partial_t \alpha_{\text{arm},\varepsilon}^{(i)}(t) = -\varepsilon$ and so

$$\int_{[0,1]\setminus[t_i^{\text{start}}, t_i^{\text{end}}]} \left(-\partial_t \alpha_{\text{arm},\varepsilon}^{(i)}(t)\right) g_i(t)\, dt = O(\varepsilon).$$

Combining with Eq. 77 in Eq. 76 yields

$$\int_0^1 \left(-\partial_t \alpha_{\text{arm},\varepsilon}^{(i)}(t)\right) g_i(t)\, dt = \left(-\log\langle \mathbf{x}_\theta^{(i)}(\mathbf{y}_i), \mathbf{x}^{(i)}\rangle\right) \int_0^1 \left(-\partial_t \alpha_{\text{arm},\varepsilon}^{(i)}(t)\right) dt + O(\varepsilon)$$

$$= -\log\langle \mathbf{x}_\theta^{(i)}(\mathbf{y}_i), \mathbf{x}^{(i)}\rangle \cdot \left(\alpha_{\text{arm},\varepsilon}^{(i)}(0) - \alpha_{\text{arm},\varepsilon}^{(i)}(1)\right) + O(\varepsilon)$$

$$= -\log\langle \mathbf{x}_\theta^{(i)}(\mathbf{y}_i), \mathbf{x}^{(i)}\rangle + O(\varepsilon),$$

since $\alpha_{\text{arm},\varepsilon}^{(i)}(0) = 1$ and $\alpha_{\text{arm},\varepsilon}^{(i)}(1) = 0$. Summing over $i = 1, \ldots, L$ proves

$$\mathcal{L}_{\text{OeMDM}}(\mathbf{x}, \theta, \alpha_{\text{arm},\varepsilon}, \alpha_{\text{arm},\varepsilon}) = \sum_{i=1}^L -\log\langle \mathbf{x}_\theta^{(i)}(\mathbf{y}_i), \mathbf{x}^{(i)}\rangle + O(\varepsilon) = -\log\prod_{i=1}^L \langle \mathbf{x}_\theta^{(i)}(\mathbf{y}_i), \mathbf{x}^{(i)}\rangle + O(\varepsilon),$$

which completes the proof. $\square$

Furthermore, we can extend the above theoretical results for the auto-regressive modeling of any fixed order:

**Corollary D.4.** *Let $\pi$ be an arbitrary but fixed permutation of $[L] := \{1, \ldots, L\}$, and denote the induced generation order by $(\pi(1), \ldots, \pi(L))$. For fixed sequence $\mathbf{x}$, let $\mathbf{y}_{\pi,i}$ be the masked sequence with $L - i + 1$ masks following $\pi$ permutation ordering:*

$$\mathbf{y}_{\pi,i} = (\mathbf{x}^{(\pi(1))}, \ldots, \mathbf{x}^{(\pi(i-1))}, \underbrace{\mathbf{m}, \ldots, \mathbf{m}}_{L-i+1})$$

*Define auto-regressive modeling of $p_{\theta,\pi}$ with time-agnostic model $\theta$ and its corresponding negative log-likelihood as follows:*

$$p_{\theta,\pi}(\mathbf{x}) = \prod_{i=1}^L \langle \mathbf{x}_\theta^{(i)}(\mathbf{y}_{\pi,i}), \mathbf{x}^{(i)}\rangle, \qquad \mathcal{L}_\pi = -\log\prod_{i=1}^L \langle \mathbf{x}_\theta^{(i)}(\mathbf{y}_{\pi,i}), \mathbf{x}^{(i)}\rangle$$

*Then, there exists $\alpha_{\pi,\epsilon} \in \mathcal{F}[\emptyset]$ that satisfies*

$$p_{\theta,\alpha_{\pi,\epsilon}}(\mathbf{x}) = p_{\theta,\pi}(\mathbf{x}) + O(\epsilon), \qquad \mathcal{L}_{OeMDM}(\theta, \alpha_{\pi,\epsilon}, \alpha_{\pi,\epsilon}) = \mathcal{L}_\pi + O(\epsilon).$$

The proof can be easily done by replacing indices $1, \ldots, L$ with $\pi(1), \ldots, \pi(L)$ in the proof of Proposition 3.3.

### D.2. OeMDM Can Express Block Diffusion Models

In this section, we show that OeMDM can also express block diffusion generation schemes. Arriola et al. (2025) propose block discrete denoising diffusion models (BD3LMs), which interpolate between autoregressive and discrete diffusion language models by (i) factorizing a sequence distribution autoregressively over blocks, and (ii) modeling each block-conditional via a discrete denoising diffusion process restricted to that block.

**Brief explanation of BD3LMs.** Let the length-$L$ sequence be partitioned into $B$ disjoint blocks $\{\mathcal{B}_b\}_{b=1}^{B}$ and write $\mathbf{x}^b := \mathbf{x}^{\mathcal{B}_b}$ and $\mathbf{x}^{<b} := \mathbf{x}^{1:b-1}$. Let $|\mathcal{B}_i| = L'$ for all $i \in [B]$ such that $L = BL'$ such that $\mathcal{B}_b = \{(b-1)L' + 1, \ldots, bL'\} \subset [L]$. BD3LMs define

$$p_{\theta,\text{bd3lm}}(\mathbf{x}) = \prod_{b=1}^{B} p_{\theta,\text{bd3lm}}(\mathbf{x}^b \mid \mathbf{x}^{<b}), \qquad \text{equivalently} \qquad \log p_{\theta,\text{bd3lm}}(\mathbf{x}) = \sum_{b=1}^{B} \log p_{\theta,\text{bd3lm}}(\mathbf{x}^b \mid \mathbf{x}^{<b}). \tag{78}$$

With a time-agnostic model $\mathbf{x}_\theta(\mathbf{z})$ defined under SUBS parametrization, BD3LM defines the reverse transition restricted to block $b$ and conditional reverse process as follows:

$$p_{\theta,\text{bd3lm}}(\mathbf{z}_s^b \mid \mathbf{z}_t^b, \mathbf{x}^{<b}) := \sum_{\tilde{\mathbf{x}}^b} q_{\alpha_{\text{mdlm}}}(\mathbf{z}_s^b \mid \mathbf{z}_t^b, \tilde{\mathbf{x}}^b)\, p_{\theta,\text{bd3lm}}(\tilde{\mathbf{x}}^b \mid \mathbf{z}_t^b, \mathbf{x}^{<b}),$$

$$p_{\theta,\text{bd3lm}}(\mathbf{x}^b \mid \mathbf{z}_t^b, \mathbf{x}^{<b}) = \prod_{i \in \mathcal{B}_b} p_{\theta,\text{bd3lm}}(\mathbf{x}^{(i)} \mid \mathbf{z}_t^b, \mathbf{x}^{<b}) = \prod_{i \in \mathcal{B}_b} \langle \mathbf{x}_\theta^{(i)}(\mathbf{z}_t^b, \mathbf{x}^{<b}), \mathbf{x}^{(i)} \rangle.$$

where the forward process and true posterior are defined the same as in MDLM; yet both act only on tokens in block $b$. Consequently, the block-conditional generative distribution is the marginal of the reverse Markov chain:

$$p_{\theta,\text{bd3lm}}(\mathbf{x}^b \mid \mathbf{x}^{<b}) := \sum_{\mathbf{z}_{t(0:T)}^b} p_{\theta,\text{bd3lm}}(\mathbf{z}_{t(T)}^b) \Big( \prod_{\tau=1}^{T} p_{\theta,\text{bd3lm}}(\mathbf{z}_s^b \mid \mathbf{z}_t^b, \mathbf{x}^{<b}) \Big) p_{\theta,\text{bd3lm}}(\mathbf{x}^b \mid \mathbf{z}_{t(0)}^b). \tag{79}$$

Sampling is performed sequentially over blocks: for $b = 1, 2, \ldots, B$, run a diffusion sampler for the conditional $p_\theta(\mathbf{x}^b \mid \mathbf{x}^{<b})$ and append the generated block, *i.e.*, it iterates the diffusion process, $t = 1 \to t = 0$, $B$ times sequentially.

In continuous-time limit (with $T \to \infty$), BD3LM derive the simplified NELBO:

$$\mathcal{L}_{\text{bd3lm}}(\mathbf{x}; \theta) := \sum_{b=1}^{B} \mathbb{E}_{t \sim \text{Unif}[0,1]}\, \mathbb{E}_{\mathbf{z}_t^b \sim q_{\alpha_{\text{mdlm}}}(\cdot | \mathbf{x}^b)} \left[ \frac{\alpha_{\text{mdlm}}(t)}{1 - \alpha_{\text{mdlm}}(t)} \log p_{\theta,\text{bd3lm}}(\mathbf{x}^b \mid \mathbf{z}_t^b, \mathbf{x}^{<b}) \right] \tag{80}$$

$$= \sum_{b=1}^{B} \mathbb{E}_{t \sim \text{Unif}[0,1]}\, \mathbb{E}_{\mathbf{z}_t^b \sim q_{\alpha_{\text{mdlm}}}(\cdot | \mathbf{x}^b)} \left[ \frac{\alpha_{\text{mdlm}}(t)}{1 - \alpha_{\text{mdlm}}(t)} \sum_{i \in \mathcal{B}_b} \log \langle \mathbf{x}_\theta^{(i)}(\mathbf{z}_t^b, \mathbf{x}^{<b}), \mathbf{x}^{(i)} \rangle \right] \tag{81}$$

**OeMDM can express BD3LM.** As detailed above, BD3LM interpolates autoregressive modeling and diffusion modeling to express block-wise L2R and block-within random ordering. We now explain how our OeMDM can also simulate such a generation scheme only with the diffusion framework. Define $\alpha_{\text{bd3lm},\epsilon}$ and corresponding trajectory sets as follows:

**Definition D.5** (BD3LM scheduler $\alpha_{\text{bd3lm},\varepsilon}$). Define block-level time windows

$$t_b^{\text{start}} := 1 - \frac{b}{B}, \qquad t_b^{\text{end}} := 1 - \frac{b-1}{B}, \qquad \Delta := t_b^{\text{end}} - t_b^{\text{start}} = \frac{1}{B}, \quad b \in [B]. \tag{82}$$

Let $S : \mathbb{R} \to [0, 1]$ be the $C^1$ smoothstep

$$S(u) := \begin{cases} 0, & u \leq 0, \\ 3u^2 - 2u^3, & 0 < u < 1, \\ 1, & u \geq 1. \end{cases} \tag{83}$$

For $\varepsilon \in (0, 1)$, define $\alpha_{\text{bd3lm},\varepsilon} \in \mathcal{F}[\emptyset]$ coordinate-wise by

$$\alpha_{\text{bd3lm},\varepsilon}^{(i)}(t) := 1 - \varepsilon t - (1 - \varepsilon) S\Big( \frac{t - t_{b(i)}^{\text{start}}}{\Delta} \Big), \qquad t \in [0, 1],\ i \in [L]. \tag{84}$$

where $b(i)$ refers to the unique block index of $i$ such that $b(i) := \lceil \frac{i}{L'} \rceil \in [B]$.

**Definition D.6** (Block-window index sets). For each block $b \in [B]$, define the window

$$t_b^{\text{start}} := 1 - \frac{b}{B}, \qquad t_b^{\text{end}} := 1 - \frac{b-1}{B}, \qquad \Delta := \frac{1}{B}.$$

Define the corresponding grid-index set

$$W_b := \{\tau \in [T] : t(\tau) \in [t_b^{\text{start}}, t_b^{\text{end}}]\}, \qquad \tau_b^{\text{start}} := \min W_b, \quad \tau_b^{\text{end}} := \max W_b.$$

Assume $T \gg B$ so that $W_b \neq \emptyset$ for all $b \in [B]$.

**Definition D.7** (BD3LM trajectory sets conditioned on $\mathbf{x}$). Let $\mathcal{S}_{\text{absorb}}(\mathbf{x}, T)$ be the monotone-masking set (Definition B.2), i.e., the set of trajectories $(\mathbf{z}_{t(0)}, \ldots, \mathbf{z}_{t(T)})$ whose masked set grows monotonically and any unmasked token equals the endpoint $\mathbf{x}$. Define the BD3LM trajectory set as the subset of $\mathcal{S}_{\text{absorb}}(\mathbf{x}, T)$, whose masking times fall inside the designated block windows:

$$\mathcal{S}_{\text{bd3lm}}(\mathbf{x}, T) = \left\{ \mathbf{z}_{t(0:T)} \in \mathcal{S}_{\text{absorb}}(\mathbf{x}, T) \,\Big|\, \exists (\kappa_i)_{i=1}^L \text{ s.t. } \kappa_i \in W_{b(i)}, \, \mathbf{z}_{t(\tau)}^{(i)} = \mathbf{x}^{(i)} \mathbb{1}\{\tau < \kappa_i\} + \mathbf{m}\mathbb{1}\{\tau \geq \kappa_i\}, \, \forall i, \forall \tau \right\},$$

where $b(i) \in [B]$ is the (unique) block index of coordinate $i$. Let $\mathcal{S}_{\text{rest}}(\mathbf{x}, T) := \mathcal{S}_{\text{absorb}}(\mathbf{x}, T) \setminus \mathcal{S}_{\text{bd3lm}}(\mathbf{x}, T)$.

Since the proof that OeMDM simulates BD3LM follows the same trajectory-set decomposition as in Proposition 3.3 (ARM case), we provide a brief proof sketch. Consider OeMDM instantiated with $\alpha_{\mathcal{F}} := \alpha_{\text{bd3lm},\varepsilon}$ and $\hat{\alpha}_{\mathcal{F}} := \alpha_{\text{bd3lm},\varepsilon}$.

- **Equivalence of generative distribution.** By construction of $\alpha_{\text{bd3lm},\varepsilon}$, for each coordinate $i$ the forward process masks $i$ almost surely *within* its designated block window, and the event that $i$ is masked outside its window has probability $O(\varepsilon)$. Taking a union bound over $i \in [L]$ yields $\mathbb{P}_{\theta,\alpha_{\text{bd3lm},\varepsilon}}(\bigcup_{\mathbf{x}} \mathcal{S}_{\text{rest}}(\mathbf{x}, T)) = O(\varepsilon)$. Hence, with probability $1 - O(\varepsilon)$ the trajectory lies in $\mathcal{S}_{\text{bd3lm}}$ and the reverse-time denoising unmasks blocks sequentially in block-wise left-to-right order, while the ordering within each block is unconstrained (and thus effectively random due to symmetry among indices in the same block). Therefore, $p_{\theta,\alpha_{\text{bd3lm},\varepsilon}}(\mathbf{x})$ matches the BD3LM block-factorized generation in Eq. 78 up to an $O(\varepsilon)$ error.

- **Equivalence of NELBO.** For the same reason, the NELBO of OeMDM decomposes block-wise up to $O(\varepsilon)$. Since we set $\alpha_{\mathcal{F}} = \hat{\alpha}_{\mathcal{F}}$, $\mathcal{L}_{\text{velocity}}$ term in $\mathcal{L}_{\text{OeMDM}}$ vanishes, and only the reconstruction term remains. For each $(i)$-th token and conditional expectation of reconstruction loss for $(i)$-th token, during the time window of block $b(i)$, the probability of BD3LM-pattern masked sequence becomes $1 - O(\epsilon)$, so the resulting reconstruction term coincides with the BD3LM conditional likelihood $\log p_{\theta,\text{bd3lm}}(\mathbf{x}^{(i)} \mid \mathbf{z}_t^{(i)}, \mathbf{x}^{<b(i)})$ with $O(\epsilon)$ error. Outside the window, the time integral of $\mathcal{L}_{\text{main}}$ also falls into $O(\epsilon)$. Altogether, $\mathcal{L}_{\text{OeMDM}}(\mathbf{x}; \theta, \alpha_{\text{bd3lm},\varepsilon}, \alpha_{\text{bd3lm},\varepsilon}) = \mathcal{L}_{\text{bd3lm}}(\mathbf{x}; \theta) + O(\varepsilon)$.

### D.3. Interpreting GenMD4 through OeMDM

In this section, we compare GenMD4 to our LoMDM. We first recall the forward process, true posterior, parametrized reverse process, and continuous-time NELBO of OeMDM:

$$q_{\alpha_{\mathcal{F}}}(\mathbf{z}_t^{(i)} \mid \mathbf{x}) = \text{Cat}\left( \alpha_{\mathcal{F}}^{(i)}(u, t)\mathbf{x}^{(i)} + (1 - \alpha_{\mathcal{F}}^{(i)}(u, t))\mathbf{m} \right),$$

$$q_{\alpha_{\mathcal{F}}}(\mathbf{z}_s^{(i)} \mid \mathbf{z}_t, \mathbf{x}) = \begin{cases} \text{Cat}(\mathbf{z}_t^{(i)}), & \text{if } \mathbf{z}_t^{(i)} \neq \mathbf{m}, \\ \text{Cat}\left( \frac{(1 - \alpha_{\mathcal{F}}^{(i)}(u,s))\mathbf{m} + (\alpha_{\mathcal{F}}^{(i)}(u,s) - \alpha_{\mathcal{F}}^{(i)}(u,t))\mathbf{x}^{(i)}}{1 - \alpha_{\mathcal{F}}^{(i)}(u,t)} \right), & \text{if } \mathbf{z}_t^{(i)} = \mathbf{m}, \end{cases}$$

$$p_{\theta,\hat{\alpha}_{\mathcal{F}}}(\mathbf{z}_s^{(i)} \mid \mathbf{z}_t) = \begin{cases} \text{Cat}(\mathbf{z}_t^{(i)}), & \text{if } \mathbf{z}_t^{(i)} \neq \mathbf{m}, \\ \text{Cat}\left( \frac{(1 - \hat{\alpha}_{\mathcal{F}}^{(i)}(\hat{u},s))\mathbf{m} + (\hat{\alpha}_{\mathcal{F}}^{(i)}(\hat{u},s) - \hat{\alpha}_{\mathcal{F}}^{(i)}(\hat{u},t))\mathbf{x}_{\theta}^{(i)}(\mathbf{z}_t,t)}{1 - \hat{\alpha}_{\mathcal{F}}^{(i)}(\hat{u},t)} \right), & \text{if } \mathbf{z}_t^{(i)} = \mathbf{m}. \end{cases}$$

$$\mathcal{L}_{\text{OeMDM}} = \int_0^1 \mathbb{E}_{q_{\alpha_{\mathcal{F}}}} \left[ \sum_{i=1}^L \langle \mathbf{z}_t^{(i)}, \mathbf{m} \rangle \left\{ -A^{(i)} \log\langle \mathbf{x}_\theta^{(i)}(\mathbf{z}_t, t), \mathbf{x}^{(i)} \rangle + A^{(i)}(\log A^{(i)} - \log \hat{A}^{(i)}) - (A^{(i)} - \hat{A}^{(i)}) \right\} \right] dt.$$

In contrast, GenMD4 (Shi et al., 2024) defines vocabulary-wise scheduler $\alpha_{\text{Gen}} : \mathcal{T} \to [0,1]^{V+1}$ where each dimension represents the different amount of noise for corresponding word. In this regard, forward process adds different amount

of noise to each position depending on which *word* it owes, *e.g.*, if $\alpha_{\text{Gen}}(t)_{[\text{"dog"}]} > \alpha_{\text{Gen}}(t)_{[\text{"cat"}]}$[3], scheduler noise "cat" more than "dog". To investigate further, we provide the forward process, true posterior, parametrized reverse process, and continuous-time NELBO of GenMD4 with our notations. We denote by $\mathbf{1} \in \mathbb{R}^{V+1}$ the all-ones vector, and use $\odot$ for element-wise product. For element-wise division we simply write it as a fraction. Then,

$$q_{\alpha_{\text{Gen}}}(\mathbf{z}_t^{(i)} \mid \mathbf{x}) = \text{Cat}\Big( \langle \alpha_{\text{Gen}}(t), \mathbf{x}^{(i)} \rangle \mathbf{x}^{(i)} + \langle \mathbf{1} - \alpha_{\text{Gen}}(t), \mathbf{x}^{(i)} \rangle \mathbf{m} \Big),$$

$$q_{\alpha_{\text{Gen}}}(\mathbf{z}_s^{(i)} \mid \mathbf{z}_t, \mathbf{x}) = \begin{cases} \text{Cat}(\mathbf{z}_t^{(i)}), & \text{if } \mathbf{z}_t^{(i)} \neq \mathbf{m}, \\ \text{Cat}\Big( \Big\langle \frac{\mathbf{1} - \alpha_{\text{Gen}}(s)}{\mathbf{1} - \alpha_{\text{Gen}}(t)}, \mathbf{x}^{(i)} \Big\rangle \mathbf{m} + \Big( \frac{\alpha_{\text{Gen}}(s) - \alpha_{\text{Gen}}(t)}{\mathbf{1} - \alpha_{\text{Gen}}(t)} \Big) \odot \mathbf{x}^{(i)} \Big), & \text{if } \mathbf{z}_t^{(i)} = \mathbf{m}, \end{cases}$$

$$p_{\theta,\alpha_{\text{Gen}}}(\mathbf{z}_s^{(i)} \mid \mathbf{z}_t) = \begin{cases} \text{Cat}(\mathbf{z}_t^{(i)}), & \text{if } \mathbf{z}_t^{(i)} \neq \mathbf{m}, \\ \text{Cat}\Big( \Big\langle \frac{\mathbf{1} - \alpha_{\text{Gen}}(s)}{\mathbf{1} - \alpha_{\text{Gen}}(t)}, \mathbf{x}_\theta^{(i)}(\mathbf{z}_t, t) \Big\rangle \mathbf{m} + \Big( \frac{\alpha_{\text{Gen}}(s) - \alpha_{\text{Gen}}(t)}{\mathbf{1} - \alpha_{\text{Gen}}(t)} \Big) \odot \mathbf{x}_\theta^{(i)}(\mathbf{z}_t, t) \Big), & \text{if } \mathbf{z}_t^{(i)} = \mathbf{m}. \end{cases}$$

$$\mathcal{L}_{\text{Gen}} = \int_0^1 \mathbb{E}_{q_{\alpha_{\text{Gen}}}} \left[ \sum_{i=1}^L \langle \mathbf{z}_t^{(i)}, \mathbf{m} \rangle \left\{ \left\langle \frac{\partial_t \alpha_{\text{Gen}}(t)}{\mathbf{1} - \alpha_{\text{Gen}}(t)}, \Big( \log\langle \mathbf{x}_\theta^{(i)}(\mathbf{z}_t, t), \mathbf{x}^{(i)} \rangle \Big) \mathbf{x}^{(i)} + \mathbf{x}^{(i)} - \mathbf{x}_\theta^{(i)}(\mathbf{z}_t, t) \right\rangle \right\} \right] dt \quad (85)$$

where $\alpha_{\text{Gen}} : \mathcal{T} \to [0,1]^{V+1}$ is a vocabulary-wise scheduler, so the noising velocity is determined only by single token. More specifically, in the forward process, the term $\langle \mathbf{1} - \alpha_{\text{Gen}}(t), \mathbf{x}^{(i)} \rangle$ represents the probability that the $i$-th token is replaced by a mask. When $\mathbf{x}^{(i)}$ is a one-hot vector corresponding to the $j$-th vocabulary token, this masking probability simplifies to $1 - \alpha_{\text{Gen}}(t)_j$. Note that the equations might appear different from Shi et al. (2024) since we explicitly use categorical distribution notation, but they are algebraically equivalent.

**Interpretation.** GenMD4 and our OeMDM both consider learnable schedulers, but they differ fundamentally in "what it dependent to" and in the expressive freedom of the scheduler. In GenMD4, the scheduler is a *vocabulary-wise* vector $\alpha_{\text{Gen}}(t) \in [0,1]^{V+1}$, so the noising velocity is determined only by the token identity (i.e., which vocabulary entry the clean token belongs to), and is shared across all contexts and positions. In contrast, our framework treats the scheduler as a *free-form object* $\alpha_{\mathcal{F}}^{(i)}(u, t)$ with minimal constraints, allowing it to depend on rich context $u$ (e.g., the entire sequence $\mathbf{x}$, the current state $\mathbf{z}_t$, and the position $i$). As a result, LoMDM can represent a broad family of generation orderings within a single framework (e.g., ARMs, MDMs, BD3LMs) by appropriate choices of $\alpha_{\mathcal{F}}$, while GenMD4 corresponds to a specific restricted parametrization. Moreover, we show below that GenMD4 *does* fall into the OeMDM framework in the continuous-time (infinitesimal) limit.

**Expressing the GenMD4 forward process and true posterior within OeMDM.** For a fixed position $i$, let

$$\alpha_{\mathcal{F}}^{(i)}(u, t) := \langle \alpha_{\text{Gen}}(t), \mathbf{x}^{(i)} \rangle \in [0, 1],$$

where $u$ contains $\mathbf{x}^{(i)}$ and $\alpha_{\text{Gen}}(t)$. Since $\mathbf{x}^{(i)}$ is one-hot, we have $\langle \mathbf{1} - \alpha_{\text{Gen}}(t), \mathbf{x}^{(i)} \rangle = 1 - \langle \alpha_{\text{Gen}}(t), \mathbf{x}^{(i)} \rangle = 1 - \alpha_{\mathcal{F}}^{(i)}(u, t)$. Thus the GenMD4 forward process is *exactly* the OeMDM forward process with scheduler defined as above:

$$q_{\alpha_{\text{Gen}}}(\mathbf{z}_t^{(i)} \mid \mathbf{x}) = \text{Cat}\Big( \langle \alpha_{\text{Gen}}(t), \mathbf{x}^{(i)} \rangle \mathbf{x}^{(i)} + \langle \mathbf{1} - \alpha_{\text{Gen}}(t), \mathbf{x}^{(i)} \rangle \mathbf{m} \Big)$$

$$= \text{Cat}\Big( \alpha_{\mathcal{F}}^{(i)}(u, t) \mathbf{x}^{(i)} + (1 - \alpha_{\mathcal{F}}^{(i)}(u, t)) \mathbf{m} \Big) = q_{\alpha_{\mathcal{F}}}(\mathbf{z}_t^{(i)} \mid \mathbf{x}).$$

Moreover, the GenMD4 true posterior also reduces to the OeMDM true posterior with the same scalar scheduler $\alpha_{\mathcal{F}}^{(i)}$. Indeed, for the mask case $\mathbf{z}_t^{(i)} = \mathbf{m}$, one-hotness implies

$$\left\langle \frac{\mathbf{1} - \alpha_{\text{Gen}}(s)}{\mathbf{1} - \alpha_{\text{Gen}}(t)}, \mathbf{x}^{(i)} \right\rangle = \frac{1 - \langle \alpha_{\text{Gen}}(s), \mathbf{x}^{(i)} \rangle}{1 - \langle \alpha_{\text{Gen}}(t), \mathbf{x}^{(i)} \rangle} = \frac{1 - \alpha_{\mathcal{F}}^{(i)}(u, s)}{1 - \alpha_{\mathcal{F}}^{(i)}(u, t)},$$

and

$$\left( \frac{\alpha_{\text{Gen}}(s) - \alpha_{\text{Gen}}(t)}{\mathbf{1} - \alpha_{\text{Gen}}(t)} \right) \odot \mathbf{x}^{(i)} = \left\langle \frac{\alpha_{\text{Gen}}(s) - \alpha_{\text{Gen}}(t)}{\mathbf{1} - \alpha_{\text{Gen}}(t)}, \mathbf{x}^{(i)} \right\rangle \mathbf{x}^{(i)} = \frac{\alpha_{\mathcal{F}}^{(i)}(u, s) - \alpha_{\mathcal{F}}^{(i)}(u, t)}{1 - \alpha_{\mathcal{F}}^{(i)}(u, t)} \mathbf{x}^{(i)}.$$

---

[3]Here, $\alpha_{\text{Gen}}(t) \in [0,1]^{V+1}$ is indexed by vocabulary tokens, and we simply write $\alpha_{\text{Gen}}(t)_{[\text{"dog"}]}$ as the entry corresponding to the one-hot index of the token "dog".

Plugging these identities into $q_{\alpha_{\text{Gen}}}(\mathbf{z}_s^{(i)} \mid \mathbf{z}_t, \mathbf{x})$ shows that the GenMD4 true posterior coincides with

$$q_{\alpha_{\mathcal{F}}}(\mathbf{z}_s^{(i)} \mid \mathbf{z}_t, \mathbf{x}) = \text{Cat}\left(\frac{1 - \alpha_{\mathcal{F}}^{(i)}(u, s)}{1 - \alpha_{\mathcal{F}}^{(i)}(u, t)}\,\mathbf{m} + \frac{\alpha_{\mathcal{F}}^{(i)}(u, s) - \alpha_{\mathcal{F}}^{(i)}(u, t)}{1 - \alpha_{\mathcal{F}}^{(i)}(u, t)}\,\mathbf{x}^{(i)}\right), \qquad \text{when } \mathbf{z}_t^{(i)} = \mathbf{m},$$

and the non-mask case $\mathbf{z}_t^{(i)} \neq \mathbf{m}$ is identical in both formulations. Therefore, GenMD4's forward process and true posterior are fully captured by OeMDM via the context-dependent scalar scheduler $\alpha_{\mathcal{F}}^{(i)}(u, t) = \langle \alpha_{\text{Gen}}(t), \mathbf{x}^{(i)} \rangle$. What remains is to compare the *model-parametrized* reverse process.

**Expressing GenMD4 reverse process within the OeMDM framework via reparametrization.** In the discrete-time reverse kernel of GenMD4, the token component is reweighted as $\left(\frac{\alpha_{\text{Gen}}(s) - \alpha_{\text{Gen}}(t)}{1 - \alpha_{\text{Gen}}(t)}\right) \odot \mathbf{x}_\theta^{(i)}(\mathbf{z}_t, t)$, which depends on the pair $(s, t)$ through element-wise weights. For a general vector-valued scheduler $\alpha_{\text{Gen}}(\cdot)$ this token-wise reweighting cannot be written in the plain scalar-mixture form $\text{Cat}\big(\beta\,\mathbf{m} + (1 - \beta)\mathbf{x}_\theta^{(i)}(\mathbf{z}_t, t)\big)$ of the discrete OeMDM reverse process without allowing an additional $(s, t)$-dependent transformation inside velocity. Hence, exact discrete-time equality for arbitrary $s < t$ is in general not possible.

However, in continuous time, the inclusion becomes exact at the level of the infinitesimal reverse transition. Let $s := t - \text{d}t$ and define the GenMD4 velocity

$$A_{\text{Gen}}(t) := -\frac{\partial_t \alpha_{\text{Gen}}(t)}{1 - \alpha_{\text{Gen}}(t)} \in \mathbb{R}_+^{V+1}, \qquad \text{(element-wise division)}, \qquad A_{\text{Gen}}(t) > \mathbf{0}.$$

A first-order expansion yields

$$\frac{\alpha_{\text{Gen}}(t - \text{d}t) - \alpha_{\text{Gen}}(t)}{1 - \alpha_{\text{Gen}}(t)} = A_{\text{Gen}}(t)\,\text{d}t + o(\text{d}t), \qquad \frac{1 - \alpha_{\text{Gen}}(t - \text{d}t)}{1 - \alpha_{\text{Gen}}(t)} = \mathbf{1} - A_{\text{Gen}}(t)\,\text{d}t + o(\text{d}t).$$

Plugging these into the GenMD4 reverse kernel (mask case) gives

$$p_{\theta, \alpha_{\text{Gen}}}\left(\mathbf{z}_{t-\text{d}t}^{(i)} \mid \mathbf{z}_t\right) = \text{Cat}\left(\left(1 - \langle A_{\text{Gen}}(t), \mathbf{x}_\theta^{(i)}(\mathbf{z}_t, t) \rangle\,\text{d}t\right)\mathbf{m} + \left(A_{\text{Gen}}(t) \odot \mathbf{x}_\theta^{(i)}(\mathbf{z}_t, t)\right)\text{d}t\right) + o(\text{d}t),$$

where we used $\langle \mathbf{1}, \mathbf{x}_\theta^{(i)} \rangle = 1$ and $\mathbf{z}_t^{(i)} = \mathbf{m}$.

Define the reparametrized model output

$$\tilde{\mathbf{x}}_\theta^{(i)}(\mathbf{z}_t, t) := \frac{A_{\text{Gen}}(t) \odot \mathbf{x}_\theta^{(i)}(\mathbf{z}_t, t)}{\left\langle A_{\text{Gen}}(t), \mathbf{x}_\theta^{(i)}(\mathbf{z}_t, t) \right\rangle} \in \Delta^{V+1}, \qquad \text{(note that } x_{\theta, \mathbf{m}}^{(i)} = 0 \Rightarrow \tilde{x}_{\theta, \mathbf{m}}^{(i)} = 0\text{)}.$$

Then

$$A_{\text{Gen}}(t) \odot \mathbf{x}_\theta^{(i)}(\mathbf{z}_t, t) = \left\langle A_{\text{Gen}}(t), \mathbf{x}_\theta^{(i)}(\mathbf{z}_t, t) \right\rangle \tilde{\mathbf{x}}_\theta^{(i)}(\mathbf{z}_t, t),$$

and moreover the scalar hazard can be written using only $(A_{\text{Gen}}, \tilde{\mathbf{x}}_\theta)$ as

$$\left\langle A_{\text{Gen}}^{\odot -1}(t), \tilde{\mathbf{x}}_\theta^{(i)}(\mathbf{z}_t, t) \right\rangle = \left\langle A_{\text{Gen}}^{\odot -1}(t), \frac{A_{\text{Gen}}(t) \odot \mathbf{x}_\theta^{(i)}(\mathbf{z}_t, t)}{\left\langle A_{\text{Gen}}(t), \mathbf{x}_\theta^{(i)}(\mathbf{z}_t, t) \right\rangle} \right\rangle$$

$$= \frac{\langle \mathbf{1}, \mathbf{x}_\theta^{(i)}(\mathbf{z}_t, t) \rangle}{\left\langle A_{\text{Gen}}(t), \mathbf{x}_\theta^{(i)}(\mathbf{z}_t, t) \right\rangle} = \frac{1}{\left\langle A_{\text{Gen}}(t), \mathbf{x}_\theta^{(i)}(\mathbf{z}_t, t) \right\rangle},$$

where $\odot^{-1}$ denotes the element-wise inverse operator, applied to each coordinate of $A_{\text{Gen}}(t)$. Hence,

$$\left\langle A_{\text{Gen}}(t), \mathbf{x}_\theta^{(i)}(\mathbf{z}_t, t) \right\rangle = \left\langle A_{\text{Gen}}^{\odot -1}(t), \tilde{\mathbf{x}}_\theta^{(i)}(\mathbf{z}_t, t) \right\rangle^{-1}.$$

Substituting into the infinitesimal kernel yields the scalar-mixture form

$$p_{\theta, \alpha_{\text{Gen}}}\left(\mathbf{z}_{t-\text{d}t}^{(i)} \mid \mathbf{z}_t\right) = \text{Cat}\left(\left(1 - \hat{A}^{(i)}(\hat{u}, t)\,\text{d}t\right)\mathbf{m} + \hat{A}^{(i)}(\hat{u}, t)\,\text{d}t\,\mathbf{x}_{\theta, \text{OeMDM}}^{(i)}(\mathbf{z}_t, t)\right) + o(\text{d}t), \qquad \mathbf{z}_t^{(i)} = \mathbf{m},$$

by choosing

$$\mathbf{x}_{\theta,\mathrm{OeMDM}}^{(i)}(\mathbf{z}_t, t) := \tilde{\mathbf{x}}_{\theta}^{(i)}(\mathbf{z}_t, t), \qquad \hat{A}^{(i)}(\hat{u}, t) := \left\langle A_{\mathrm{Gen}}^{\odot-1}(t), \mathbf{x}_{\theta,\mathrm{OeMDM}}^{(i)}(\mathbf{z}_t, t) \right\rangle^{-1},$$

where $\hat{u}$ contains $(\mathbf{z}_t, t)$ and the scheduler (equivalently $A_{\mathrm{Gen}}$). Therefore, GenMD4 is exactly expressed by the continuous-time (velocity-form) OeMDM reverse parameterization.

**NELBO equivalence.** We now show that the continuous-time NELBO of GenMD4 coincides with the OeMDM NELBO under the above identifications. First, since the forward process (and hence the true posterior) is identical under $\alpha_{\mathcal{F}}^{(i)}(u, t) = \langle \alpha_{\mathrm{Gen}}(t), \mathbf{x}^{(i)} \rangle$, the trajectory distributions agree, so the expectations are taken over the same law: $\mathbb{E}_{q_{\alpha_{\mathcal{F}}}}[\cdot] = \mathbb{E}_{q_{\alpha_{\mathrm{Gen}}}}[\cdot]$. Next, consider the OeMDM integrand inside $\mathcal{L}_{\mathrm{OeMDM}}$ on masked positions. With the infinitesimal reparametrization established above, we substitute $\mathbf{x}_{\theta,\mathrm{OeMDM}}^{(i)}(\mathbf{z}_t, t) = \tilde{\mathbf{x}}_{\theta}^{(i)}(\mathbf{z}_t, t)$ and $\hat{A}^{(i)}(\hat{u}, t) = \langle A_{\mathrm{Gen}}^{\odot-1}(t), \mathbf{x}_{\theta,\mathrm{OeMDM}}^{(i)}(\mathbf{z}_t, t) \rangle^{-1} = \langle A_{\mathrm{Gen}}(t), \mathbf{x}_{\theta}^{(i)}(\mathbf{z}_t, t) \rangle$. Moreover, for the forward (true) scalar velocity we take the GenMD4-induced hazard $A^{(i)} = \langle A_{\mathrm{Gen}}(t), \mathbf{x}^{(i)} \rangle$ (which is consistent with the above choice of $\alpha_{\mathcal{F}}^{(i)}$). Using one-hotness of $\mathbf{x}^{(i)}$, we have

$$\left\langle \tilde{\mathbf{x}}_{\theta}^{(i)}(\mathbf{z}_t, t), \mathbf{x}^{(i)} \right\rangle = \frac{\langle A_{\mathrm{Gen}}(t) \odot \mathbf{x}_{\theta}^{(i)}(\mathbf{z}_t, t), \mathbf{x}^{(i)} \rangle}{\langle A_{\mathrm{Gen}}(t), \mathbf{x}_{\theta}^{(i)}(\mathbf{z}_t, t) \rangle} = \frac{\langle A_{\mathrm{Gen}}(t), \mathbf{x}^{(i)} \rangle \langle \mathbf{x}_{\theta}^{(i)}(\mathbf{z}_t, t), \mathbf{x}^{(i)} \rangle}{\langle A_{\mathrm{Gen}}(t), \mathbf{x}_{\theta}^{(i)}(\mathbf{z}_t, t) \rangle}.$$

Substituting this identity together with $A^{(i)} = \langle A_{\mathrm{Gen}}(t), \mathbf{x}^{(i)} \rangle$ and $\hat{A}^{(i)} = \langle A_{\mathrm{Gen}}(t), \mathbf{x}_{\theta}^{(i)}(\mathbf{z}_t, t) \rangle$ into the OeMDM integrand $-A^{(i)} \log \langle \mathbf{x}_{\theta,\mathrm{OeMDM}}^{(i)}, \mathbf{x}^{(i)} \rangle + A^{(i)}(\log A^{(i)} - \log \hat{A}^{(i)}) - (A^{(i)} - \hat{A}^{(i)})$:

$$\begin{aligned}
&- A^{(i)} \log \left\langle \mathbf{x}_{\theta,\mathrm{OeMDM}}^{(i)}(\mathbf{z}_t, t), \mathbf{x}^{(i)} \right\rangle + A^{(i)} \big( \log A^{(i)} - \log \hat{A}^{(i)} \big) - (A^{(i)} - \hat{A}^{(i)}) \\
&= -A^{(i)} \log \left\langle \tilde{\mathbf{x}}_{\theta}^{(i)}(\mathbf{z}_t, t), \mathbf{x}^{(i)} \right\rangle + A^{(i)} \big( \log A^{(i)} - \log \hat{A}^{(i)} \big) - (A^{(i)} - \hat{A}^{(i)}) \\
&= -A^{(i)} \log \left( \frac{\langle A_{\mathrm{Gen}}(t), \mathbf{x}^{(i)} \rangle \langle \mathbf{x}_{\theta}^{(i)}(\mathbf{z}_t, t), \mathbf{x}^{(i)} \rangle}{\langle A_{\mathrm{Gen}}(t), \mathbf{x}_{\theta}^{(i)}(\mathbf{z}_t, t) \rangle} \right) + A^{(i)} \big( \log A^{(i)} - \log \hat{A}^{(i)} \big) - (A^{(i)} - \hat{A}^{(i)}) \\
&= -A^{(i)} \Big( \log \langle A_{\mathrm{Gen}}(t), \mathbf{x}^{(i)} \rangle + \log \langle \mathbf{x}_{\theta}^{(i)}(\mathbf{z}_t, t), \mathbf{x}^{(i)} \rangle - \log \langle A_{\mathrm{Gen}}(t), \mathbf{x}_{\theta}^{(i)}(\mathbf{z}_t, t) \rangle \Big) \\
&\qquad + A^{(i)} \log A^{(i)} - A^{(i)} \log \hat{A}^{(i)} - A^{(i)} + \hat{A}^{(i)} \\
&= -A^{(i)} \log \langle A_{\mathrm{Gen}}(t), \mathbf{x}^{(i)} \rangle - A^{(i)} \log \langle \mathbf{x}_{\theta}^{(i)}(\mathbf{z}_t, t), \mathbf{x}^{(i)} \rangle + A^{(i)} \log \langle A_{\mathrm{Gen}}(t), \mathbf{x}_{\theta}^{(i)}(\mathbf{z}_t, t) \rangle \\
&\qquad + A^{(i)} \log A^{(i)} - A^{(i)} \log \hat{A}^{(i)} - A^{(i)} + \hat{A}^{(i)}.
\end{aligned}$$

Now substitute $A^{(i)} = \langle A_{\mathrm{Gen}}(t), \mathbf{x}^{(i)} \rangle$ and $\hat{A}^{(i)} = \langle A_{\mathrm{Gen}}(t), \mathbf{x}_{\theta}^{(i)}(\mathbf{z}_t, t) \rangle$. Then the terms $-A^{(i)} \log \langle A_{\mathrm{Gen}}(t), \mathbf{x}^{(i)} \rangle + A^{(i)} \log A^{(i)}$ cancel, and the terms $A^{(i)} \log \langle A_{\mathrm{Gen}}(t), \mathbf{x}_{\theta}^{(i)}(\mathbf{z}_t, t) \rangle - A^{(i)} \log \hat{A}^{(i)}$ also cancel, yielding

$$-A^{(i)} \log \left\langle \mathbf{x}_{\theta}^{(i)}(\mathbf{z}_t, t), \mathbf{x}^{(i)} \right\rangle - A^{(i)} + \hat{A}^{(i)} = -\langle A_{\mathrm{Gen}}(t), \mathbf{x}^{(i)} \rangle \log \langle \mathbf{x}_{\theta}^{(i)}(\mathbf{z}_t, t), \mathbf{x}^{(i)} \rangle - \langle A_{\mathrm{Gen}}(t), \mathbf{x}^{(i)} \rangle + \langle A_{\mathrm{Gen}}(t), \mathbf{x}_{\theta}^{(i)}(\mathbf{z}_t, t) \rangle.$$

Finally, using $\frac{\partial_t \alpha_{\mathrm{Gen}}(t)}{1 - \alpha_{\mathrm{Gen}}(t)} = -A_{\mathrm{Gen}}(t)$, the OeMDM integrand becomes

$$\begin{aligned}
&-\langle A_{\mathrm{Gen}}(t), \mathbf{x}^{(i)} \rangle \log \langle \mathbf{x}_{\theta}^{(i)}(\mathbf{z}_t, t), \mathbf{x}^{(i)} \rangle - \langle A_{\mathrm{Gen}}(t), \mathbf{x}^{(i)} \rangle + \langle A_{\mathrm{Gen}}(t), \mathbf{x}_{\theta}^{(i)}(\mathbf{z}_t, t) \rangle \\
&= \left\langle \frac{\partial_t \alpha_{\mathrm{Gen}}(t)}{1 - \alpha_{\mathrm{Gen}}(t)}, (\log \langle \mathbf{x}_{\theta}^{(i)}(\mathbf{z}_t, t), \mathbf{x}^{(i)} \rangle) \mathbf{x}^{(i)} + \mathbf{x}^{(i)} - \mathbf{x}_{\theta}^{(i)}(\mathbf{z}_t, t) \right\rangle
\end{aligned}$$

which matches the GenMD4 integrand in Eq. 85.

## E. Training Stabilization Techniques

### E.1. REINFORCE Leave-One-Out

In Section 4.3, we have stated that $\mathbb{E}_{q_{\alpha_{\phi}}}[\nabla_{\phi} \log q_{\alpha_{\phi}} \cdot (\mathcal{L}_{\mathrm{main}} + \mathcal{L}_{\mathrm{velocity}})]$ in $\nabla_{\phi} \mathcal{L}_{\mathrm{LoMDM}} = \mathbb{E}_{q_{\alpha_{\phi}}}[\nabla_{\phi} \log q_{\alpha_{\phi}} \cdot (\mathcal{L}_{\mathrm{main}} + \mathcal{L}_{\mathrm{velocity}})] + \mathbb{E}_{q_{\alpha_{\phi}}}[\nabla_{\phi}(\mathcal{L}_{\mathrm{main}} + \mathcal{L}_{\mathrm{velocity}})]$ is a high-variance estimator. Therefore, we use a low-variance estimator

of such optimization problems proposed by Kool et al. (2019). Before deriving a low-variance estimator, we treat $\mathbb{E}_{q_{\alpha_\phi}}[\nabla_\phi \log q_{\alpha_\phi} \cdot (\mathcal{L}_{\text{main}} + \mathcal{L}_{\text{velocity}})]$ as follows:

$$\mathbb{E}_{q_{\alpha_\phi}}[\nabla_\phi \log q_{\alpha_\phi} \cdot (\mathcal{L}_{\text{main}} + \mathcal{L}_{\text{velocity}})] = \nabla_\phi \mathbb{E}_{q_{\alpha_\phi}}[\text{Sgd}(\mathcal{L}_{\text{main}} + \mathcal{L}_{\text{velocity}})] = \nabla_\phi \mathbb{E}_{\mathbf{z}_t \sim q_{\alpha_\phi}}[\text{Sgd}(\mathcal{L}_{\mathbf{z}_t})],$$

where we denote stop-gradient as Sgd and $\mathcal{L}_{\text{main}} + \mathcal{L}_{\text{velocity}}$ for $\mathbf{z}_t$ as $\mathcal{L}_{\mathbf{z}_t}$ as in main paper. Then, the low-variance estimator (Kool et al., 2019) is given as follows:

$$\nabla_\phi \mathbb{E}_{\mathbf{z}_t \sim q_{\alpha_\phi}}\left[\text{Sgd}(\mathcal{L}_{\mathbf{z}_t})\right] \approx \frac{1}{k} \sum_{i=1}^{k} \nabla_\phi \log q_{\alpha_\phi}(\mathbf{z}_t^i|\mathbf{x}) \left(\mathcal{L}_{\mathbf{z}_t^i} - \frac{1}{k-1} \sum_{j \neq i} \mathcal{L}_{\mathbf{z}_t^j}\right) \tag{86}$$

$$= \frac{1}{k-1} \sum_{i=1}^{k} \nabla_\phi \log q_{\alpha_\phi}(\mathbf{z}_t^i|\mathbf{x}) \left(\mathcal{L}_{\mathbf{z}_t^i} - \frac{1}{k} \sum_{j=1}^{k} \mathcal{L}_{\mathbf{z}_t^j}\right). \tag{87}$$

Here we set $k = 2$ so that,

$$\nabla_\phi \mathbb{E}_{\mathbf{z}_t \sim q_{\alpha_\phi}}\left[\text{Sgd}(\mathcal{L}_{\mathbf{z}_t})\right] \approx \frac{1}{2}\left(\nabla_\phi \log q_{\alpha_\phi}(\mathbf{z}_t^1|\mathbf{x}) - \nabla_\phi \log q_{\alpha_\phi}(\mathbf{z}_t^2|\mathbf{x})\right)\left(\mathcal{L}_{\mathbf{z}_t^1} - \mathcal{L}_{\mathbf{z}_t^2}\right). \tag{88}$$

**Final gradient estimation** Let

$$\widehat{\mathcal{L}}_{\text{LoMDM}}(\mathbf{z}_t^1, \mathbf{z}_t^2) := \frac{1}{2}(\mathcal{L}_{\mathbf{z}_t^1} + \mathcal{L}_{\mathbf{z}_t^2}) + \mathcal{L}_{\text{rloo}}(\mathbf{z}_t^1, \mathbf{z}_t^2), \qquad \text{where} \quad \mathcal{L}_{\text{rloo}}(\mathbf{z}_t^1, \mathbf{z}_t^2) := \frac{1}{2} \log \frac{q_{\alpha_\phi}(\mathbf{z}_t^1|\mathbf{x})}{q_{\alpha_\phi}(\mathbf{z}_t^2|\mathbf{x})}\left(\text{Sgd}(\mathcal{L}_{\mathbf{z}_t^1}) - \text{Sgd}(\mathcal{L}_{\mathbf{z}_t^2})\right),$$

such that $\mathbb{E}_{\mathbf{z}_t^1, \mathbf{z}_t^2 \sim q_{\alpha_\phi}}[\nabla_\phi \mathcal{L}_{\text{rloo}}(\mathbf{z}_t^1, \mathbf{z}_t^2)]$ is the two sample gradient estimator of reinforce term. Furthermore, since $\mathcal{L}_{\text{rloo}}$ is invariant to the gradient of $\psi, \theta$, final gradient is given as folows:

$$\nabla_\phi \mathcal{L}_{\text{LoMDM}} = \mathbb{E}_t[\mathbb{E}_{\mathbf{z}_t^1, \mathbf{z}_t^2 \sim q_{\alpha_\phi}}[\nabla_\phi \widehat{\mathcal{L}}_{\text{LoMDM}}(\mathbf{z}_t^1, \mathbf{z}_t^2)]],$$

$$\nabla_\psi \mathcal{L}_{\text{LoMDM}} = \mathbb{E}_t[\mathbb{E}_{\mathbf{z}_t^1, \mathbf{z}_t^2 \sim q_{\alpha_\phi}}[\nabla_\psi \widehat{\mathcal{L}}_{\text{LoMDM}}(\mathbf{z}_t^1, \mathbf{z}_t^2)]],$$

$$\nabla_\theta \mathcal{L}_{\text{LoMDM}} = \mathbb{E}_t[\mathbb{E}_{\mathbf{z}_t^1, \mathbf{z}_t^2 \sim q_{\alpha_\phi}}[\nabla_\theta \widehat{\mathcal{L}}_{\text{LoMDM}}(\mathbf{z}_t^1, \mathbf{z}_t^2)]],$$

where the two-sample estimator $\mathbb{E}_t[\mathbb{E}_{\mathbf{z}_t^1, \mathbf{z}_t^2 \sim q_{\alpha_\phi}}[\nabla_\phi \widehat{\mathcal{L}}_{\text{LoMDM}}(\mathbf{z}_t^1, \mathbf{z}_t^2)]]$ gives lower variance than the naive estimator $\nabla_\phi \mathcal{L}_{\text{LoMDM}} = \mathbb{E}_{\mathbf{z}_t \sim q_{\alpha_\phi}}[(\nabla_\phi \log q_{\alpha_\phi}) \cdot \mathcal{L}_{\mathbf{z}_t}] + \mathbb{E}_{q_{\alpha_\phi}}[\nabla_\phi \mathcal{L}_{\mathbf{z}_t}]$.

# F. Details about LoMDM parametrization

## F.1. Rationale behind the LoMDM parametrization

In this section, we explain why we parametrize LoMDM as in Eq. 4–Eq. 7. Unlike discrete diffusion, which has only recently begun to be explored, the continuous domain (e.g., image diffusion) has a longer history of studying *learnable* noise schedulers. However, the image and text domains differ substantially, making it nontrivial to directly transfer such techniques. We therefore first briefly review how learnable schedulers are typically introduced in continuous diffusion.

**Learnable schedulers in continuous diffusion: variational diffusion models.** A representative example is the variational diffusion model (VDM), which introduces a single learnable scheduler shared across the entire image (Kingma et al., 2021). Since the scheduler must be monotone decreasing in time, VDM learns it via an additional monotonic neural network that is separate from the diffusion model. In contrast, such monotonic-network-based parametrizations are difficult to apply in our case, modeling a different generation order in a text space.

First, since the diffusion model itself cannot be constrained to be monotone in $t$, a monotonic neural network (NN) must be decoupled from the diffusion model. In VDM, the monotonic NN learns a single scheduler shared across all pixels; because

this mapping is low-dimensional and input-agnostic, it is easy to train and can be implemented with a small network. In contrast, in text generation, inducing nontrivial generation orders requires a scheduler that depends on the given context and outputs *token-wise* scheduling values. This would necessitate a large, standalone monotonic NN separate from the diffusion model, which is computationally and statistically inefficient.

Second, the quantity that is learned in our formulation is not $\alpha(\cdot, t)$ itself, but rather

$$A(\cdot, t) \;=\; \partial_t \alpha(\cdot, t) \oslash \big(1 - \alpha(\cdot, t)\big).$$

In particular, the OeMDM NELBO is expressed as a weighted sum of token-wise reconstruction losses with weights given by $A^{(i)}(\cdot, t)$. Under such a nontrivial functional relationship between $\alpha$ and $A$, naively parameterizing $\alpha$ via a monotonic NN as in VDM provides no direct mechanism to ensure that tokens with lower reconstruction loss receive smaller effective training weights. Consequently, VDM-style monotonic-network parametrizations are ill-suited for discrete diffusion with token-wise scheduling, motivating other parametrizations for texts.

**Learnable schedulers in continuous diffusion: multivariate learned adaptive noise.** Sahoo et al. (2024b) further advance this line of work by proposing *multivariate learned adaptive noise* (MULAN). In a text-conditioned image diffusion model, MULAN takes the conditioning text as input and produces *pixel-wise* schedulers, allowing different pixels to follow different noise schedules. However, in light of the limitations of monotonic NNs discussed above, MULAN adopts a different design: rather than learning a full scheduler via a standalone monotonic network, it parameterizes the scheduler as a quintic polynomial controlled by three parameters. For image dimension $d = L \times H$, MULAN define neural networks $\mathbf{a}_\vartheta(\mathbf{x}) : \mathcal{V}^L \to \mathbb{R}^d$, $\mathbf{b}_\vartheta(\mathbf{x}) : \mathcal{V}^L \to \mathbb{R}^d$, and $\mathbf{c}_\vartheta(\mathbf{x}) : \mathcal{V}^L \to \mathbb{R}^d$ with parameters $\vartheta$. Let $f_\vartheta : \mathcal{V}^L \times [0, 1] \to \mathbb{R}^d$ be defined as

$$f_\vartheta(\mathbf{x}, t) = \frac{\mathbf{a}_\vartheta(\mathbf{x})^2}{5} \, t^5 \;+\; \frac{\mathbf{a}_\vartheta(\mathbf{x}) \, \mathbf{b}_\vartheta(\mathbf{x})}{2} \, t^4 \;+\; \frac{\mathbf{b}_\vartheta(\mathbf{x})^2 + 2 \, \mathbf{a}_\vartheta(\mathbf{x}) \, \mathbf{c}_\vartheta(\mathbf{x})}{3} \, t^3 \;+\; \mathbf{b}_\vartheta(\mathbf{x}) \, \mathbf{c}_\vartheta(\mathbf{x}) \, t^2 \;+\; \mathbf{c}_\vartheta(\mathbf{x})^2 \, t, \qquad (89)$$

where the multiplication and division operations are elementwise. The noise schedule $\alpha_\vartheta(\mathbf{x}, t)$ is given by

$$\alpha_\vartheta(\mathbf{x}, t) = \mathrm{Sigmoid}\bigg( - \Big(\gamma_{\min} + \big(\gamma_{\max} - \gamma_{\min}\big) \, \frac{f_\vartheta(\mathbf{x}, t)}{f_\vartheta(\mathbf{x}, t = 1)}\Big)\bigg), \qquad (90)$$

where it always guarantees that $\alpha$ is monotone decreasing.

However, it remains difficult to apply this approach directly to text. As in VDM, the quantity that is directly learned is the velocity $A$, rather than $\alpha$ itself. Concretely, for tokens that are poorly reconstructed, training should reduce the corresponding weights $A^{(i)}$. Yet, since $\alpha_\vartheta$ is already a complex function, and $A$ becomes even more intricate as it is induced through $\alpha_\vartheta$, aligning $A^{(i)}$ with reconstruction quality becomes highly challenging. We empirically attempted to train with this parametrization; yet, when we measured the Pearson correlation between $A_\phi^{(i)}(\mathbf{x}, t)$, $A_\psi^{(i)}(\mathbf{z}_t, t)$, and $\langle \mathbf{x}_\theta^{(i)}(\mathbf{z}_t, t), \mathbf{x}^{(i)} \rangle$ throughout training, we observed that none of these correlations increased meaningfully, contrast to that of our LoMDM shown in in Figure 4.

**Parametrizion of LoMDM.** From the continuous-diffusion case above, we draw one key lesson: the velocity $A$ should admit a *simple form* that enables direct optimization. While continuous diffusion often allows highly flexible parameterizations of $\alpha$, in text space we instead require a suitably regularized and simple functional form. Motivated by the simplest scheduler used in MDLM, $\alpha_{\mathrm{mdlm}}(t) = 1 - t$, we have proposed the LoMDM parametrization. To recap,

$$\alpha_\phi^{(i)}(\mathbf{x}, t) := 1 - t^{\,c_1 + c_2 \cdot \big[\mathrm{NormSig}(g_\phi(f(\mathbf{x})))\big]_i}, \qquad A_\phi^{(i)}(\mathbf{x}, t) = \frac{c_1 + c_2 \cdot \big[\mathrm{NormSig}(g_\phi(f(\mathbf{x})))\big]_i}{t}, \qquad (91)$$

$$\hat{\alpha}_\psi^{(i)}(\mathbf{z}_t, t) := 1 - t^{\,c_1 + c_2 \cdot \big[\mathrm{NormSig}(g_\psi(f(\mathbf{z}_t)))\big]_i}, \qquad \hat{A}_\psi^{(i)}(\mathbf{z}_t, t) = \frac{c_1 + c_2 \cdot \big[\mathrm{NormSig}(g_\psi(f(\mathbf{z}_t)))\big]_i}{t}. \qquad (92)$$

where we can modulate the overall denoising velocity by $c_1$ and the variance of generation order priority by $c_2$. We also experimented with softmax; however, it became overly biased toward a single position, preventing training from stabilizing.

Another thing we should consider is that learning a large, standalone scheduler network for text space would be infeasible. Fortunately, Hong et al. (2025) have shown that we can post-train unmasking ordering with one transformer layer attached to the diffusion backbone model. They empirically show that the post-trained unmasking policy model boosts the performance of MDM even without learning MDM itself; *i.e.*, MDM is frozen while training. Inspired by this work, we set $f(\cdot) = \mathrm{Sgd}\big(\theta_{\mathrm{TF}}(\cdot)\big)$ as the feature extractor in the above equations where $\theta_{\mathrm{TF}}$ is the backbone of diffusion model.

*Table 8.* Hyperparameter ablation on $(c_1, c_2)$ for LoMDM on LM1B with sentence packing: we train separate LoMDM models for each $(c_1, c_2)$ for 1M steps and report the resulting test PPL ($\downarrow$). When $c_2 = 0$, the scheduler is non-learnable and reduces to the polynomial form $\alpha(t) = 1 - t^{c_1}$. In particular, $(c_1, c_2) = (1, 0)$ recovers MDLM.

| $(c_1, c_2)$ | $(0.7, 0.0)$ | $(0.7, 0.65)$ | $(1.0, 0.0)$ | $(1.0, 0.95)$ | $(1.3, 0.0)$ | $(1.3, 1.25)$ |
|---|---|---|---|---|---|---|
| Test PPL $\downarrow$ | 31.7 | **27.2** | 31.8 | 28.7 | 31.4 | 28.9 |

### F.2. Ablation study on $c_1$ and $c_2$

We here provide and discuss an ablation study on our LoMDM that was omitted in the main paper, where we train the model under various choices of $c_1$ and $c_2$. To recap, $\alpha_\phi$ and $\alpha_\psi$ are given by:

$$\alpha_\phi^{(i)}(\mathbf{x}, t) := 1 - t^{c_1 + c_2 \cdot [\mathrm{NormSig}(g_\phi(f(\mathbf{x})))]_i}, \qquad \hat{\alpha}_\psi^{(i)}(\mathbf{z}_t, t) := 1 - t^{c_1 + c_2 \cdot [\mathrm{NormSig}(g_\psi(f(\mathbf{z})))]_i}.$$

In particular, when $(c_1, c_2) = (1, 0)$, our scheduler reduces exactly to that of MDLM. More generally, if $c_2 = 0$ and $c_1 > 0$ is arbitrary, then the identical polynomial scheduler is applied at every position, *i.e.*, $\alpha^{(i)}(t) = 1 - t^{c_1}$ for all $i \in [L]$.

Before presenting the experiments, we provide a more detailed interpretation of $c_1$ and $c_2$. A larger $c_1$ implies that $\alpha$ decays more slowly (since $t < 1$), meaning that a substantial amount of masking occurs only when $t$ is close to 1. Conversely, a smaller $c_1$ implies a faster decay of $\alpha$, so masking is concentrated near $t \approx 0$. When $c_2 = 0$, although the decay rate varies with $c_1$, the NELBO becomes effectively independent of the scheduler after an appropriate change of variables with respect to $\alpha$; hence all such cases yield the same objective, which is also mentioned by Sahoo et al. (2024a). We now turn to the case $c_2 \neq 0$. The parameter $c_2$ controls the position-dependent strength determined by the scheduler models $g_\phi$ and $g_\psi$. In other words, $c_2$ governs the variance of generation-order priorities across positions: as $c_2$ increases, some indices can be masked much earlier (i.e., with a much faster decay) while others can be masked much later (i.e., with a much slower decay).

Through the ablation over $(c_1, c_2)$, we aim to answer two questions. First, we ask whether the performance gains of our LoMDM are driven by the choice of $c_1$—that is, by using a polynomial time scheduler unlike MDLM—or whether the gains truly come from learning position-dependent schedules via $g_\phi$ and $g_\psi$. Second, we investigate how to set $(c_1, c_2)$ so that our LoMDM achieves its best performance.

The results are reported in Table 8. First, for every tested value of $c_1$, setting $c_2 \neq 0$ yields substantially lower PPL than $c_2 = 0$, confirming that the context-aware scheduler in our LoMDM is indeed beneficial for learning. Furthermore, when $c_2 = 0$, the PPL remains nearly constant at $\sim 31.8$ across different choices of $c_1$. As discussed above, this is expected because $c_2 = 0$ makes the scheduler non-learnable, and a change of variables with respect to $\alpha$ renders the NELBO invariant to the specific choice of $c_1$; thus, varying $c_1$ mainly changes *which* masked states are visited, but not the overall amount of training signal allocated to the diffusion model.

Second, among $c_2 \neq 0$ settings, performance improves as $c_1$ decreases. Unlike the $c_2 = 0$ case, $c_2 \neq 0$ yields a context- and position-dependent scheduler, allowing the objective to reweight learning across tokens according to learned generation priorities. Since heavily masked states provide limited contextual information, many masked positions in such states can yield weak or noisy gradients. Our method alleviates this by concentrating the training signal on the subset of tokens that remain *learnable* under the current context via learned, position-wise priorities. With smaller $c_1$, training visits more highly corrupted states, and the learned scheduler can allocate more of the learning budget within those states to the most informative (learnable) tokens, offering a plausible explanation for the consistent gains at smaller $c_1$ when $c_2 \neq 0$.

## G. Experimental Details

### G.1. Experimental Settings

We detokenize the One Billion Words dataset following Lou et al. (2024); Sahoo et al. (2024a), whose official code can be found. We tokenize LM1B using the BERT-BASE-UNCASED tokenizer, consistent with He et al. (2023). We then concatenate and pack the sequences to a fixed length of 128 (Raffel et al., 2020). For OpenWebText (OWT), we use the GPT-2 tokenizer and similarly concatenate and pack sequences to length 1,024; during packing, we insert an eos token between consecutive documents. As OWT does not provide an official validation split, we reserve the last 100k documents for validation. We parameterize the diffusion backbone of our LoMDM with the modified diffusion transformer architecture (Peebles & Xie, 2023) from Lou et al. (2024); Sahoo et al. (2024a). We use 12 layers, a hidden dimension of 768, 12 attention heads, and for

LoMDM, one more same size of transformer layer for each $\phi, \psi$. For diffusion backbone, we use the AdamW optimizer with a batch size of 512, constant learning rate warmup from 0 to a learning rate of 3e-4 for 2,500 steps following prior works (Sahoo et al., 2024a; Lou et al., 2024; Sahoo et al., 2025). For schedulers $\phi$ and $\psi$, we use the AdamW optimizer with constant learning rate warmup from 0 to a learning rate of 1e-5 for 2,500 steps. We use a dropout rate of 0.1. As specified in the main paper, we use $c_1 = 0.7, c_2 = 0.65$ for the parametrization of $\phi$ and $\psi$. For both LM1B with/without sentence packing, we utilized 2 H200 GPUs for training. For OWT, we utilized 8 H100 GPUs for training.

### G.2. Qualitative Examples

We present qualitative examples in this section. Figure 6 compares the generation trajectories of MDLM and LoMDM trained for 1M steps on LM1B. One interesting property is that LoMDM tends to generate structural elements first and fill in semantic content later. More specifically, unlike MDLM, LoMDM first generates words that play structural roles in a sentence, such as prepositions and articles. It then fills in words that carry substantive information, such as nouns and verbs. We do not observe a left-to-right bias in the learned generation trajectories of LoMDM. We speculate that, for unconditional generation, a coarse-to-fine generation order may be more effective than a left-to-right order. In contrast, what generation order is preferable for conditional generation tasks, such as QA, remains an interesting direction for future work.

We also provide additional generation examples from LoMDM trained on OWT in the following section.

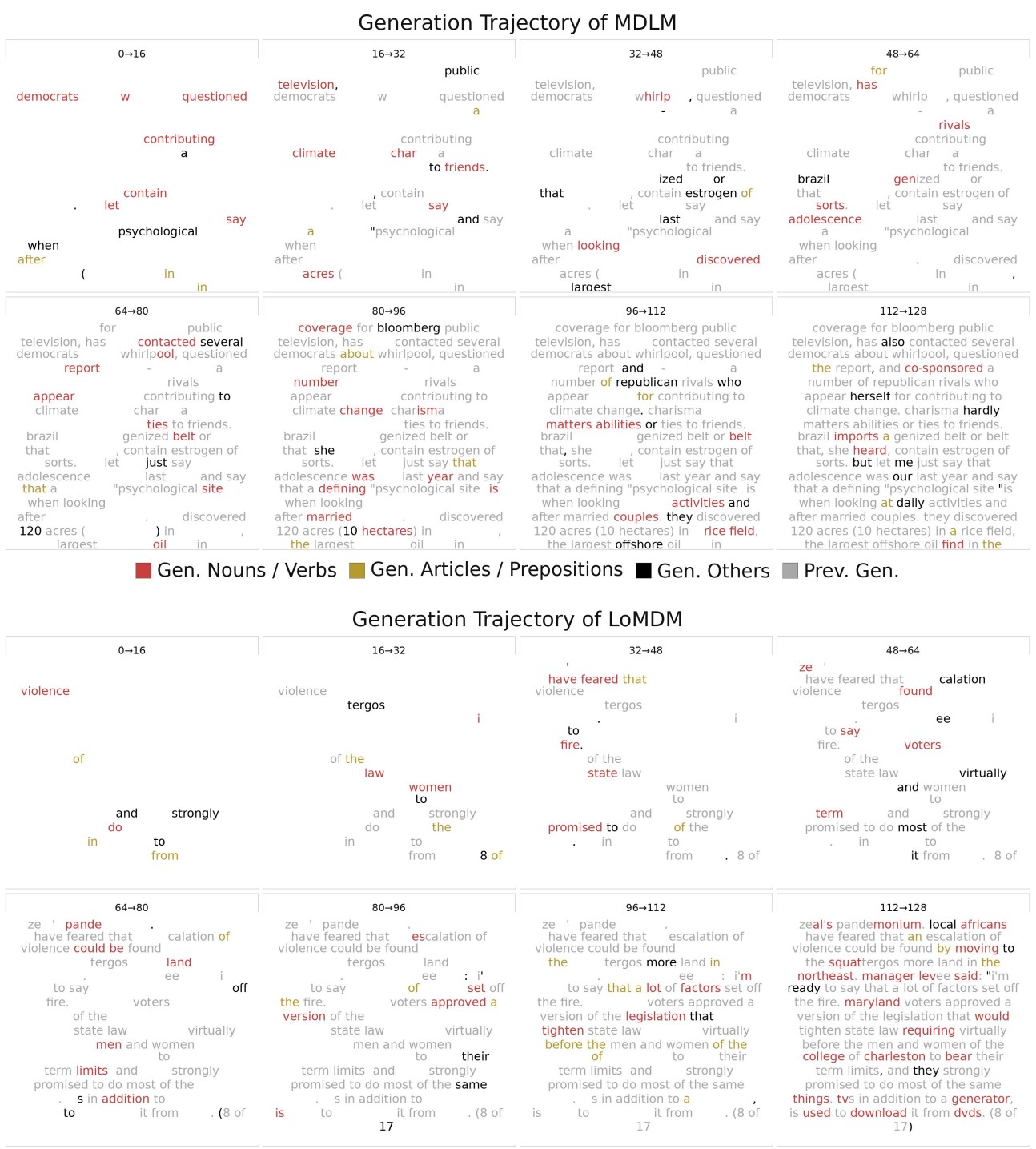

*Figure 6.* **Generation trajectories of MDLM (top) and LoMDM (bottom).** Both models were trained on LM1B for 1M steps, and each figure shows a length-128 sentence generated over 128 decoding steps. Each panel visualizes which tokens are generated at different stages of decoding. Newly generated nouns and verbs are shown in **red**, newly generated articles and prepositions are shown in **yellow**, other newly generated tokens are shown in **black**, and previously generated tokens are shown in **gray**. MDLM tends to fill semantic content, such as nouns and verbs, relatively early, whereas LoMDM tends to first establish structural words, such as prepositions and articles, and then fill in semantic content later.

Text sample generated by LoMDM trained on OpenWebText with 1024 sampling steps.

<|endoftext|>, Indian states have dampened the productivity of some core industries. Most have been acquired by Western companies and a few new industries, supported by outsourcing opportunities. Nevertheless structural changes and Indian forces in recent years brought economists thinking that some company reforms that were made possible with better work conditions --- better wages and unpaid IBM engineers --- made industry jobs much easier to work.

Rathan's opinion in Global European, judged by economist Ernst Karelis hardly ignores workers, the technologies that have bought them information, and the basic human dignity of the world's largest companies. Nor does he argue that the economic and social growth of world industrial and economic powers are mutually beneficial in the near future. Mostly, however, it does not deny that India can further improve its productivity, as is indeed the case today.

India is a leading economic power and one of the world's leading ``empower'' countries (see OECD). Moreover, it is a large and developed country with its own market, in the shape of the European Economic Area, an area of aggregating services and goods in the labour market. Only 4.4 percent of the tariff-free transport network will be shipped from India. But with the fast-liberalisation of India, which is sure to attract workers from small company sizes, it will be possible to make wage cuts targeting GST.

India's share in exports is quite large. Actually, imports generate the largest share, because imports are brought by low-wage workers like Indian workers. That is thanks to India's production of many large steel plants. Most of its neighbours, and Germany, would have enjoyed the same kind of work at their industrial plants. Of course, erasing the global income gap requires massive capital controls.

Sure, it could tailor its economic policies to its trade policy, including trade with Japan. But closing the larger gap means one much bigger problem. Firms can't rise above borders and capital controls generally exist outside the country. Trying to foresee how India might achieve structural changes may discount rising income levels of workers in countries like Japan. Since unions are a larger part of the workforce, additional labour measures would require India to take far less in one move. Nor is it worth mentioning that only Sweden, Brazil, Germany, and Argentina are three foreign countries which don't have any trade with India but manage Delhi's existing exports.

Nor, is it possible that a worker, independent of bringing his or her social status in once and all, will benefit from capital gains from India's policies.

In the countries where the ratio of workers in population has increased by some sort in recent years, the standard labor standard of workers has fallen in India, as for even industrial countries as advanced as the United States. For decades, the improvement in average-adjusted life expectancy in developed nations was greater than at its 30 years prior, partly as a result of growth in productivity. The recent recession has prevented a full recovery, partly because an increasing sustained supply of workers has only increased, and some even fewer workers died from disability. One of the most notable examples coming in the Second World War, in which, the Soviet Union created a new type of economy that raised the full-time wages of some 200 million, lowered the rise in life expectancy for workers who retired over 15-month tenure, and vastly reduced its labour force by hundreds of workers for economic reasons.

In India, export the majority of heavy goods and its labour force has expanded incrementally. Since 1950, industrial productivity has shot rise to 7.9 percent, just a little higher for an Indian citizen. The average-adjusted productivity (which had been rising since 1942) continued for several decades. It is the middle-age expectation that India will be able to match the rising productivity to 11 percent in five decades, and so should have a strong advantage. To meet the needs of its civilian workers, world industrial growth has called on India to increase its number of highly skilled workers.

Consider that the steel sector, like any other part of Japan, is a modern industry that, like Japanese, has remarkable rates of growth. Politicians from all 160 member states would respond well to a takeover of industry and population against a province with half the exports of the industrialized world.

In recent years, however, a wave of entrepreneurs have begun to expand drastically, spread to the manufacturing of leather (in China), buring bags, and other new technologies done in developed countries. While a new (and more compact) global order must, it could best be avoided. In points of view, globalization is part of a process that stemmed from the fall of empires; that no modern country or even some of the modern states can handle the consequences of the modern industry.'s inability to protect the masses if necessary or rail on other kinds of horrors on a new day.

The country blurring our attention is India. It is the politically advanced country that needs to improve its income, its productivity and its vested interests who intend to plunder it.

Ameliorating Indian<|endoftext|>

Text sample generated by LoMDM trained on OpenWebText with 1024 sampling steps.

In an interview with the Houston Chronicle, Hummel Brock said it was an ideal club for Ryan's season. And coming into his decision to stay in Texas he indicated that that was the correct decision.

``And I said, `I'm looking forward to enjoying myself here,''' he said, according to the Chronicle. ``And I said, `I'm crazy' haven't played here after less than four weeks. I'm more than happy to sit down, grow this team and teach myself and keep learning so that we can depend on like that.

``Now that I'm healthy and I can return to the home game which when I was a player, it'll be my starter home again.''

The Texans fought against competition, with players Alvin Nakamura, DB, and Eli Kennedy entering the open and ended up being the Kos' offseason target.

Bacette has been selected after a few stints. Ryan took the roster in Week 4 which gave Ryan a chance to establish some momentum and some could have expected the beast would move to Houston next offseason.<|endoftext|>With the crew got fired off on the air set uttering on Tuesday, Fox announced that it was making progress on its upcoming ``The X-Men'' film sequel.

The upcoming film stars J.J. Smith and Mark Shumalo, and the producers the studio is owned by Fox Vacation Pictures was imposing a real-life partnership with co-writing producers Simon Kinberg and Kosethel.

The two executives agreed that there were previously even fewer contributors, facing procedural issues the company of production may have overshadowed over Fox's top priorities --- such as five-page paperwork and video cameras.

Rather, Fox said they were there to hit ``the Network'' and FX production costs and began building a set of ``bases and buildings,'' which wound up curbing their runs and the weekend totals charge for years to come.

PHOTOS: The Official Story For The First X-Men Movie

Wayne Thomas and Chanel Wigehouse were the executive producers and the script for the film revolves around being done by a first-time movie producer for Kinberg and by Patti Jones for the showrunners.

``The producers found a success and gave us what we needed,'' said original executive producer Breanna Carrappen. ``At the same time, we were pressed to get things done, no one felt it was possible with the extraordinary looks and the production quality.''

Menial debut to two X-Men titles Smith worked on the animated ``Doctor Strange,'' with Chris Evans in charge with the role of Peter Parker and Peter X

Then produced the fully animated ``The X-Men'' with James Cameron attached as director. ``I shaped and wrote the script,'' Kosethel has said. ``I was Dell/the manager of the production design.''

Jones' own stint was a rocky one at first. She wrote of NBC Channel ``The X-Men'' film she was first directed by Nicholas Meyer and successfully slipped on her way to showrunner for ``Fox'' until late early 2012. Her replacement did not until 2013.

Kosethel had the experience and charm of collaborating on several X-Men titles with screen producers, like ``Watchmen'', S.E.K.D.L.C.E.L.D (``The Terrific X-Men'' and ``Best of Viewers''), which liked the director's choices to match their diverse demands.

X-Men

``Not every person was when they wanted to make it except the top directors of the time period,'' explained Thomas. ``It would never go that way, every time you do something completely different, it would go through these different subtleties.''

Kosethel also starkly conflicted the studio as to John being on a staff project and as to what the matter of the material was.

``I developed `Vning' these characters and there was no other movie that would relate to them, might have an impact here,'' said Thomas. ``I had to assemble the right pieces that they all agreed with and that there were all things that were right for it.''

McFarlane might be back, full time, and he will instill the journey with the returning cast by taking his leave.

``It has always been important for me to produce who have career highs,/or statuses, who have important roles to play. I have to understand that script and studio are two of the arenas to drive the business<|endoftext|>

