# OpenReview forum: "Unifying Masked Diffusion Models with Various Generation Orders and Beyond"
_ICML.cc/2026/Conference — ICML 2026 spotlight_

### Official Review · Reviewer_p4nG · 2026-03-08

**Soundness:** 3
**Presentation:** 3
**Significance:** 3
**Originality:** 3
**Overall Recommendation:** 4
**Confidence:** 3

**Summary:**

This paper studies how generation order affects masked diffusion language models. It introduces OeMDM, a unified framework with position-dependent noise scheduling, yielding a generalized NELBO with a velocity-weighted reconstruction term and a forward–reverse velocity mismatch term. The framework recovers standard MDMs, autoregressive models, block diffusion, and GenMD4 as special cases. Building on this, the paper proposes LoMDM, which jointly learns a context-dependent generation order and the diffusion backbone through a single objective using lightweight scheduler heads. Experiments on LM1B and OpenWebText show improved perplexity over prior diffusion baselines, and on OpenWebText LoMDM reaches MDLM’s 1M-step performance in about 0.18M steps.

**Compliance With Llm Reviewing Policy:**

Affirmed.

**Key Questions For Authors:**

1. Can you visualize a few actual generation trajectories learned by LoMDM? Figure 4 provides indirect evidence through correlation analysis, but qualitative examples of token-level generation order would help verify whether the model learns meaningful context-dependent orderings.

2. What is the concrete wall-clock slowdown of LoMDM relative to MDLM? The paper describes the additional passes qualitatively, but a precise runtime and memory comparison would help assess the practical cost of the method.

3. How robust are the results to the current scheduler design choices? In particular, did you try alternative scheduler parameterizations or training without stop-gradient between the scheduler and backbone, and if so, how did those variants perform?

**Limitations:**

The impact statement is reasonable. However, the paper does not include a dedicated limitations section. It would be helpful to explicitly discuss the restricted experimental scale, the reliance on perplexity-based evaluation, the sensitivity to the
(c1,c2) hyperparameters, and the limitations of the chosen scheduler form.

**Strengths And Weaknesses:**

**Strengths**

- OeMDM provides a meaningful theoretical unification: ARMs, standard MDMs, BD3LM, and GenMD4 are all recovered through scheduler choices within one framework, and the generalized NELBO cleanly decomposes into a velocity-weighted reconstruction term and a forward–reverse velocity mismatch term.

- LoMDM’s joint learning setup is technically coherent: it combines stop-gradient feature sharing, lightweight scheduler heads, and an RLOO-based estimator, and the training-dynamics analysis shows the learned scheduler correlates with reconstruction confidence.


**Weaknesses**

- The experimental scale is limited to the paper’s standard setup, so it remains unclear whether the gains persist at larger model scales; moreover, while the paper acknowledges extra computation, it does not provide a precise quantitative overhead breakdown.

- The evaluation includes test, zero-shot, and generative perplexity, plus correlation analysis, but it does not provide explicit visualizations of the learned generation orders, which would strengthen the paper’s central interpretability claim.

- The scheduler parameterization is relatively constrained and depends on tuned hyperparameters (c1,c2); appendix ablations show noticeable sensitivity, suggesting performance depends meaningfully on this design choice.

---

> ### Author Rebuttal · Authors · 2026-03-31
>
> We thank the reviewer for the valuable feedback! We conducted all of the requested experiments and will include them in the camera-ready version. We will also add a dedicated limitation section as suggested.
>
> ### **W1**. The experimental scale is limited to the paper’s standard setup, so it is unclear whether the method will also remain effective at a larger scale.
>
> For this rebuttal, we trained a model that is **2.8× larger than the original setting** on LM1B for 0.5M steps. More specifically, while the original model uses a Transformer with hidden size / number of blocks / number of heads equal to 768/12/12, we scaled this to 1024/24/16. Under this larger setting, **MDLM achieved a PPL of 27.5, whereas LoMDM achieved a much lower PPL of 24.6.**
>
> Meanwhile, we ask for the reviewer’s understanding that experiments in the paper are already as large as we can reasonably conduct in an academic setting. Even MDLM alone requires 1K> H100 GPU hours on OWT in the current setting. As the reviewer also acknowledged, within a standard academic setup, we have carried out the largest and most thorough experiments that were realistically possible.
>
> ### **W1/Q2**. Report the wall-clock slowdown and memory usage relative to MDLM during training.
>
> LoMDM runs about 1.6× slower than MDLM in terms of training steps per unit time, **but reaches the same performance level about 3× faster in wall-clock time.** In Figure 5, we have already shown test PPL per wall-clock time for both MDLM and LoMDM. More specifically, as detailed in the rebuttal, when trained on OWT with 8 H100 GPUs, LoMDM reaches in about ~50 hours the performance that MDLM attains only after about ~170 hours of training.  In addition, we have observed that LoMDM trains much faster under other hardware as well; please refer to BRJQ:W3 for further discussion.
>
> In addition, LoMDM uses about 1.4× more memory when training, as it requires additional backprop graphs. However, in language modeling, training is commonly performed with gradient accumulation, and we did not observe a practical issue.
>
> ### **W2/Q1**. No visualization of the learned order.
>
> We visualized the generation paths of LoMDM and found interesting properties of the learned order. In particular, **LoMDM tends to generate structural elements first and fill in semantic content later**. As the ICML rebuttal policy allows anonymous links to figures, we provide below a link to figures illustrating the learned order of LoMDM: https://anonymous.4open.science/r/OeMDM
>
> Since the figures alone may be difficult to interpret, we also provide quantitative results together with our interpretation, and we kindly ask the reviewer to refer to X7Et:Q1 for details.
>
> ### **W3**. In Table 4, the performance is strongly affected by the choice of $(c_1, c_2)$.
>
> In Table 4, even though the performance varies with $(c_1, c_2)$, LoMDM still achieves substantially lower PPL than the baseline MDLM across all tested settings. In addition, our chosen setting works well consistently across different setups, which suggests that this issue is not severe in practice. Furthermore, through the rebuttal, we further confirm that the proposed LoMDM and its hyperparameters remain robustly better than MDLM even when the model size changes substantially. For details, please refer to 54z1:Q2/Q3.
>
> ### **Q3**. Have you tried other schedulers? (e.g., without stop-gradient, or alternative parametrization)
>
> We have already experimented with both end-to-end gradient flow and Softmax parameterization, and neither performed well.
>
> **End-to-end gradient**. On LM1B, after 0.5M training steps, LoMDM with gradient detachment achieved a perplexity of 29.2, whereas the end-to-end gradient version of LoMDM achieved a perplexity of 36.3. To interpret this result, by design, $\phi,\psi,\theta$ in LoMDM are independent parameters that serve different roles, so separating their gradients is conceptually cleaner. We conjecture that training $\theta$ to handle both token prediction and order-priority learning at the same time makes optimization more difficult. In particular, although the correlations among $<x_\theta^i,x^i>, A^i_\phi$, and $A_\psi^i$ do increase during training, the resulting dynamics were much messier.
>
> **Softmax parametrization**. We already noted in the “Parametrization of LoMDM” paragraph in Section F.1 that “We also experimented with softmax; it became overly biased toward a single position, preventing training from stabilizing.” More specifically, we experimented with the parameterization $A_\phi^i(x,t) = \frac{c_1 + c_2 \cdot [SoftMax^i(g_\phi))]}{t}$ (and likewise for $A_\psi$) under various choices of $c_1,c_2$. The results after training for 500K steps on LM1B are as follows.
> | Parametrization | $(c_1, c_2)$ | PPL |
> | --- | --- | --- |
> | NormSig | (0.7, 0.65) | 29.2 |
> | Softmax | (0.7, 1) | 34.3 |
> |  | (0.7, 3) | 34.1 |
> | | (0.7, 10) | 34.0 |
>
> We have also tested $c_1=1.0/1.3$ and $c_2=1/3/10$ but all give worse results.

---

> > ### Author Rebuttal · Reviewer_p4nG · 2026-04-02
> >
> > Thanks for the rebuttal. It addresses my concerns. Considering the overall quality of the responses, I will keep my current score.

---

> > > ### Author Response · Authors · 2026-04-06
> > >
> > > We are glad that the reviewer’s concerns have been fully addressed! We hope that our method will inspire further work within the MDM framework.

---

### Official Review · Reviewer_BRJQ · 2026-03-12

**Soundness:** 3
**Presentation:** 3
**Significance:** 2
**Originality:** 2
**Overall Recommendation:** 4
**Confidence:** 4

**Summary:**

This paper examines a key weakness of masked diffusion models (MDMs): their generation quality is highly sensitive to the order in which tokens are revealed. Conventional approaches typically rely on either random or manually designed unmasking sequences, but these strategies often lead to suboptimal performance, To address this issue, the authors introduce the Order-expressive Masked Diffusion Model (OeMDM), a framework that reframes the noise scheduler as a core modeling element. Through this perspective, MDMs, autoregressive models (ARMs), and block diffusion can all be unified within a single formulation based on a generalized Negative Evidence Lower Bound (NELBO). Extending this idea further, the work proposes the Learnable-order Masked Diffusion Model (LoMDM), which simultaneously learns both the token generation order and the diffusion backbone during training. Rather than relying on fixed schedules, LoMDM employs a context-dependent scheduler that decides which positions should be revealed next by leveraging information from the entire sequence.

**Compliance With Llm Reviewing Policy:**

Affirmed.

**Final Justification:**

I have decided to raise my score to 4. The authors' rebuttal and the supplementary experiments—specifically the theoretical grounding for few-step generation and the parameterization improvements for text—show practical merit.

That said, I remain reserved about the distinction from LO-ARM. The underlying mechanism of using a weighted loss to learn generation order is functionally very similar. I am willing to support the paper, but this is contingent on the authors delivering a detailed and objective comparison in the final version. It is important that they go beyond architectural differences to clearly define the unique technical boundaries of this work.

**Key Questions For Authors:**

1. The work used $c_1=0.7$ and $c_2=0.65$ for all datasets. How was this specific ratio discovered, and does the model remain stable if $c_2$ approaches or exceeds $c_1$?
2. The model detach the gradient from the scheduler flowing into the diffusion backbone. Did you experiment with end-to-end gradient flow? If so, did it lead to "collapsing" orderings where the model only learns a trivial L2R path?
3. The scheduler uses 1 Transformer layer and 1 MLP. How does the complexity of this "ordering head" affect the quality of the learned velocity? Is a simpler MLP-only head sufficient?
4. While the model outperform other diffusion models, it still lags behind the Autoregressive Transformer baseline in primary benchmarks like OWT (20.4 vs 17.5 PPL). In your view, what is the "missing ingredient" to finally close this gap?

**Strengths And Weaknesses:**

**Strengths**

- The OeMDM framework provides a principled way to view different discrete generative processes (ARMs, MDMs, BD3LM) as special cases of a single formulation by varying the noise scheduler.
- The paper presents compelling evidence that LoMDM is a faster learner, matching 1M-step MDLM performance in only 18% of the steps (~180K steps) on the Open WebText dataset.
- LoMDM  outperforms various discrete diffusion baselines in test perplexity across multiple benchmarks (LM1B, OWT) and zero-shot evaluations.

**Weaknesses**

- While the authors propose a unified framework, the shift from vocabulary-dependent scheduling in **GenMD4**  to context-dependent ordering is a natural progression that has been conceptually explored in previous work, specifically **Learning-order Autoregressive Models (LO-ARM)**. The manuscript fails to cite or discuss LO-ARM, which is a significant omission given that the proposed OeMDM framework appears to be largely a continuous-time relaxation of the LO-ARM objective. This lack of differentiation makes it difficult to assess the true technical leap beyond existing discrete-time or autoregressive ordering methods.

[1]. Shi J, Han K, Wang Z, et al. Simplified and generalized masked diffusion for discrete data.

[2]. Wang Z, Shi J, Heess N, et al. Learning-order autoregressive models with application to molecular graph generation.

- The specific form of the scheduler velocity ($A_{\phi}$ and $\hat{A}_{\psi}$) feels heavily engineered with specific constants ($c_1, c_2$) and normalization layers (NormSig) to ensure stability and finiteness of the NELBO. The sensitivity to these hyperparameters (as seen in Table 4) suggests the model needs some further discussion
- Although LoMDM is sample-efficient, it requires an additional forward pass of the backbone and two more forward/backward passes of the scheduler modules per step. This overhead is mentioned but not fully analyzed in terms of total wall-clock time versus total performance across a wider variety of hardware.
- The paper tracks correlations but omits a visualization or quantitative breakdown of the *actual* learned orderings. It is unclear if a meaningful generative strategy is being learned or simply a form of importance sampling.

---

> ### Author Rebuttal · Authors · 2026-03-31
>
> We thank the reviewer for the valuable feedback! We conducted all of the requested experiments and will include them in the camera-ready version.
>
> **W1**. LO-ARM is not cited and discussed.
>
> Thank you for pointing us to this valuable work! We will cite and discuss it in more detail in the camera-ready version. LO-ARM and our method share the broad goal of jointly learning generation order and token prediction. However, **LO-ARM and OeMDM are fundamentally different in their modeling design.** (1) Sequence modeling: LO-ARM is based on AR factorization, whereas OeMDM is built on the MDM framework. (2) Order modeling: LO-ARM models it through variational inference, whereas OeMDM models it through a scheduler.
>
> **This modeling difference leads to a fundamental distinction in the derived NELBO.** In the LO-ARM objective, the token-prediction loss itself is not directly scaled by order priority, as they are decomposed through a logarithm. By contrast, in OeMDM, order priority appears as a **direct scaling factor** on the token-prediction loss: it explicitly biases training toward tokens with higher order priority. As stated in our paper, we view this as a fundamental recipe for order-aware MDM training, and through OeMDM, we can understand how order shapes inference and training of MDMs. We believe it is the notable point that distinguishes OeMDM from LO-ARM.
>
> **W2**. LoMDM introduces several hyperparameters, so additional discussion, such as Table 4, would be helpful.
>
> Please refer to our responses to your Q1, Q2, and Q3. We have also shown that current hyperparameters perform well stably under various model sizes in 54z1:Q3.
>
> **W3**. Report the wall-clock time vs. performance across a range of hardware.
>
> LoMDM runs about 1.6× slower than MDLM in terms of training steps per unit time, **but reaches the same performance level about 3× faster in wall-clock time.**
>
> In Figure 5, we have already shown test PPL per wall-clock time for both MDLM and LoMDM. More specifically, as detailed in the rebuttal, when trained on OWT with 8 H100 GPUs, LoMDM reaches in about ~50 hours the performance that MDLM attains only after about ~170 hours of training. In addition, the LM1B experiment reported in the paper was conducted on 2 H200 GPUs, where LoMDM surpassed in 17 hours the performance of MDLM obtained after 50 hours of training (1M steps).
>
> We also trained LM1B on 2 RTX 4090 for this rebuttal. Please note that we used a smaller size (hidden size/# of blocks/# of heads set to 512/8/8) due to the limited time and speed of 4090. Results are as follows:
>
> | Hours | 20 | 40 | 60 |
> | --- | --- | --- | --- |
> | MDLM | 43.1 | 40.7 | **39.7** |
> | LoMDM | **39.3** | 36.2 | 35.1 |
>
> **W4**. No discussion of the learned ordering.
>
> We visualized the generation paths of LoMDM and found interesting properties of the learned order. In particular, **LoMDM tends to generate structural elements first and fill in semantic content later**. Due to the limited space, please refer to p4nG:Q3 for a detailed response.
>
> **Q1**. How were $c_1$ and $c_2$ chosen, and is the model still stable when $c_2$ becomes close to or larger than $c_1$?
>
> We selected these values primarily based on Table 4 (ablation on LM1B). Furthermore, we have observed that settings such as $(c_1,c_2)=(0.7,0.69)$ give marginally better performance. However, we were concerned that such more aggressive choices might be less robust across different datasets. Since the diffusion language modeling is very costly, e.g., MDLM requires 1K> H100 GPU hours to train on OWT, we chose a reasonably stable $(c_1,c_2)$ for global setting rather than tuning more aggressively.
>
> When $c_1<c_2$, training diverges at an early stage. This is consistent with Proposition C.3, which already proves that the finiteness condition of the NELBO is guaranteed when $c_1>c_2$.
>
> **Q2**. Have you tested end-to-end gradient flow? If so, does it collapse to a trivial L2R path?
>
> We did try this, but the performance was worse (PPL increased by 7.1). Due to the limited space, please refer to X7Et/Q1 for a detailed response. In addition, paths appeared nearly uniform when averaged and showed no clear relation to an L2R.
>
> **Q3**. LoMDM without a Transformer layer.
>
> We introduced the Transformer layer to make the generation order context-aware, whereas an MLP layer is not fundamentally designed to analyze linguistic context. We trained LoMDM with only 1 MLP on LM1B for 0.5M steps, and the results show that performance is substantially better when the Transformer layer is included.
>
> |  | LoMDM | LoMDM w/o TF layer | MDLM |
> | --- | --- | --- | --- |
> | PPL | 29.2 | 31.3 | 32.5 |
>
> **Q4**. What is the key missing ingredient for LoMDM to close the remaining gap to ARM?
>
> We believe the next step is efficient training. ARM uses the fixed L2R order, which is already a strong heuristic, whereas LoMDM must additionally learn the generation order itself. Due to the limited space, please refer to X7Et:W1 for a more detailed discussion.

---

> > ### Author Rebuttal · Reviewer_BRJQ · 2026-04-03
> >
> > We thank the authors for their rebuttal. However, the concern regarding the distinction between OeMDM and LO-ARM remains unresolved.
> >
> > In your response to W1, you argue that LO-ARM is fundamentally different because its token-prediction loss is not '*directly scaled by order priority*' and is '*decomposed through a logarithm.*'
> >
> > However, looking at the inner term $F\_\theta(\mathbf{z}\_{<i}, \mathbf{x})$ of the LO-ARM objective:
> >
> > $$
> > F\_\theta(\mathbf{z}\_{<i}, \mathbf{x}) = \sum\_{j=1}^L q\_\theta(z\_i = j | \mathbf{z}\_{<i}, \mathbf{x}) \cdot \left[ \log \frac{p\_\theta(z\_i = j | \dots) p\_\theta(x\_j | \dots)}{q\_\theta(z\_i = j | \dots)} \right]
> > $$
> >
> > Since the unmasking index $j$ ranges over a finite set, this is a tractable, explicit weighted sum where the policy $q_\theta$ acts as a direct per-token scaling factor for the reconstruction log-likelihood (and it's directly computed by $q$ network in one pass in their implementation). This seems functionally identical to how the velocity $A^{(i)}$ biases the training signal in OeMDM.
> >
> > Given this, could you clarify:
> >
> > 1. Is there a specific qualitative difference in the resulting gradients between your 'velocity-scaled' loss and LO-ARM’s 'policy-weighted' sum?
> >
> > 2. Beyond the continuous-time vs. discrete-time formulation, does LoMDM provide a mathematical advantage in how it 'concentrates' the learning budget compared to this tractable weighting in LO-ARM?

---

> > > ### Author Response · Authors · 2026-04-06
> > >
> > > We thank the reviewer for the detailed comments. The explicit form of the loss provided by the reviewer helped deepen our understanding of LO-ARM. We agree with the reviewer that OeMDM and LO-ARM both (1) learn generation order and token prediction, and (2) **optimize order-weighted CE loss that are intuitively similar**. We promise to add a separate detailed section on this relationship in the camera-ready.
> > >
> > > We still believe that **OeMDM makes distinct contributions**. OeMDM and LO-ARM have different foundations: LO-ARM is built on autoregressive (AR) factorization, whereas OeMDM is built on the MDM framework. This distinction leads to the following contributions.
> > >
> > > ### Theoretical contributions
> > > - **Unified MDM framework with various generation orders.** The MDM framework has recently become an important direction in text modeling. **A key contribution of OeMDM** is that it provides a unified view of L2R ARM (Prop. 3.3), fixed random-order ARM (Cor. D.4), BD3LM (Sec. D.2), GenMD4 (Sec. D.3), and MDLM within the MDM framework. This can help future researchers better understand how generation order operates within MDM. LO-ARM can also represent L2R ARM, fixed random-order ARM, and any-order ARM, but it does not provide a unified view of multiple MDM classes.
> > > - **OeMDM supports a wide range of future work within MDM.** Our definition of a free-form scheduler imposes only very general mathematical conditions, from which we derive a generalized NELBO. Thus, beyond unifying generation orders in MDM, OeMDM provides a flexible foundation for future work on order modeling. One can even design an MDM whose generation order varies with $t$ for the same masked sequence $z$. LO-ARM may likewise inspire future work in the AR setting, but we believe OeMDM will be especially useful for researchers interested in MDM order modeling.
> > > - **Few-step generation.** One advantage of MDM over ARM is that it supports few-step generation. MDM has attracted strong interest from both academia and industry as it allows fast generation. This distinction also extends to OeMDM and LO-ARM. In OeMDM, at timestep $t$, $\hat A^{(i)}(z,t)$ determines the relative transition probabilities of the $i$-th tokens and the probability that a masked token remains masked, which makes few-step sampling theoretically compatible. In contrast, in LO-ARM, $p_\theta(z^i \mid z^{<i})$ is normalized to 1 at each step and built under AR framework, which forces one token to be generated in one step. Although LO-ARM may still generate multiple tokens in an engineering sense, this is different from being theoretically grounded. In this regard, OeMDM opens the door to future research on a broad range of few-step samplers under different generation orders.
> > >
> > > ### Practical contributions
> > > - **SOTA in diffusion language modeling.** LoMDM is developed for language modeling, whereas LO-ARM is studied on molecular graphs. We believe it is a meaningful contribution to show that learnable-order generation is effective for language. LoMDM achieves SOTA PPL among diffusion language models and outperforms MDLM in few-step generation. Molecular graphs are inherently more structured than text, so their generation order may be more explicit. Therefore, introducing learnable order into text might require additional effort and a different approach as follows.
> > > - **Different parameterizations of LoMDM and LO-ARM.** To model order, LoMDM uses a normalized sigmoid parameterization, whereas LO-ARM uses the Plackett-Luce model with an exponential function, which is conceptually close to Softmax. We already stated in Section F.1 that a Softmax does not perform well in our setting, as Softmax tends to place overly strong mass on a single masked token. We conjecture that this behavior is particularly problematic for language modeling. In this rebuttal, we further show in p4nG:Q3 that Softmax continues to underperform across a range of hyperparameter choices.
> > > - **Different model architectures in LoMDM and LO-ARM.** We separate the token-prediction module from the order-generation module and, more specifically, add one Transformer and linear layer for order generation together with stop-gradient. In contrast, LO-ARM adds one linear layer for order generation and treats it as shared parameters or uses a fully independent model. The reviewer requested experiments that use only 1 MLP layer, or using an end-to-end gradient corresponds to LO-ARM architectures, and we showed experimentally that these do not work well. Furthermore, the fully independent-model case is impractical in language modeling, where computation is a major constraint.
> > >
> > > We again thank the reviewer for pointing us to the LO-ARM paper. While the two works share a similar philosophy, we hope the reviewer will consider that **they make parallel contributions in different domains**. We believe that unifying generation order in MDM and achieving SOTA in **discrete diffusion language models** are, by themselves, meaningful contributions.

---

### Official Review · Reviewer_54z1 · 2026-03-13

**Soundness:** 3
**Presentation:** 3
**Significance:** 4
**Originality:** 4
**Overall Recommendation:** 5
**Confidence:** 3

**Summary:**

This paper first presents a unified masked diffusion framework, OeMDM, that accommodates diverse generation orders. It then builds on this framework to develop LoMDM, and experiments show that LoMDM achieves state-of-the-art performance over existing discrete diffusion models on multiple language modeling benchmarks.

**Compliance With Llm Reviewing Policy:**

Affirmed.

**Final Justification:**

The authors' rebuttal and supplementary experiments have addressed most of my key concerns. I remain positive about the paper and will maintain my score.

**Key Questions For Authors:**

1. As shown in Table 2, Diffusion (Absorbing state) models  outperform autoregressive (AR) models on several datasets, with LoMDM further amplifying this advantage. However, on datasets such as LM1B, all diffusion-based models underperform AR models by a significant margin. Does this indicate that different datasets have varying compatibility with distinct text generation rules?

2. In baseline experiments, c1 is set to 0.7 instead of smaller values. Is this choice primarily based on the ablation results presented in Table 4? If other changes are made to the model architecture, will the principles for selecting c1 also shift?

3. When the number of layers or overall scale of the model is altered, will the values of c1 and c2 significantly affect the stability of model training and the final performance of the model?

**Limitations:**

yes

**Strengths And Weaknesses:**

## Strengths:
1. The declarations and conclusions have detailed formula derivations and arguments.

2. This paper states that the proposed OeMDM can express various generation orders, and proves that ARMs are a special case of OeMDM.

3. The proposed LoMDM achieves SoTA on multiple datasets.

## Weaknesses:

See questions.

---

> ### Author Rebuttal · Authors · 2026-03-31
>
> We thank the reviewer for the valuable feedback! We conducted all of the requested experiments and will include them in the camera-ready version.
>
> ### **Q1**. Why do several diffusion models outperform ARM in perplexity in Table 2, but not in Table 1? Does this indicate that different datasets have varying compatibility with distinct text generation rules?
>
> Thank you for this interesting question. First, we clarify that Table 1 reports test-set perplexity, where the training and test data come from the same domain, whereas Table 2 reports zero-shot perplexity, where the evaluation domain is different.
>
> We believe this difference may be related to the **generalization effect of discrete diffusion models through a form of data augmentation**, rather than the existence of distinct text generation rules for each dataset. More specifically, for a sequence of length L, a diffusion model can, in principle, generate up to $2^L$ different prediction problems. In this sense, the training data can be viewed as being augmented in many different ways. Indeed, prior work [1] has shown that discrete diffusion models can outperform ARM in limited-data settings. We therefore conjecture that this stronger augmentation effect may reduce overfitting, which could explain why diffusion models perform better on zero-shot datasets even when their in-domain test perplexity is worse.
>
> [1] Prabhudesai et al., “Diffusion Beats Autoregressive in Data-Constrained Settings,” NeurIPS’25.
>
> ### **Q2**. Why is $c_1$ set to 0.7? Is this choice based on the results in Table 4?
>
> As the reviewer noted, when designing our experiments, we selected the setting $c_1=0.7$ based on Table 4 and used it across all datasets. In fact, we did observe that using a smaller value leads to a marginal improvement. However, we were concerned that such more aggressive choices might be less robust across different datasets. Since the diffusion language modeling is very costly to study in an academic setting, e.g., MDLM requires 1K> H100 GPU hours to train on OWT, so we chose a reasonably stable $(c_1,c_2)$ for global hyperparameter rather than tuning more aggressively.
>
> ### **Q2/3**. Would the principle for choosing $c_1$ and $c_2$ change when the model changes? Also, would $c_1$ and $c_2$ affect training stability and final performance when the model changes?
>
> Thank you for your valuable feedback! We additionally trained both LoMDM and MDLM for 0.5M steps on the LM1B dataset under a smaller setting and a larger setting. Please note that the original model uses a Transformer with hidden size / # of blocks / # of heads set to 768 / 12 / 12. The results are as follows.
>
> | Hidden size / # of blocks / # of heads | PPL of MDLM | PPL of LoMDM $(c_1=0.7,c_2=0.65)$ | PPL of LoMDM $(c_1=1.0,c_2=0.95)$ | PPL of LoMDM $(c_1=1.3,c_2=1.25)$ |
> | --- | --- | --- | --- | --- |
> | 512/8/8 | 40.1 | 34.3 | 35.9 | 37.0 |
> | 1024/24/16 | 27.5 | 24.6 | N/A | N/A |
>
> From the results under the 512/8/8 setting, we observe that LoMDM with $(c_1,c_2)=(0.7,0.65)/(1.0,0.95)/(1.3,1.25)$ consistently outperforms MDLM, and that  $(c_1,c_2)=(0.7,0.65)$ gives the best performance among them. This suggests that even when the model changes, $(c_1,c_2)=(0.7,0.65)$ gives better performance. Furthermore, this also suggests that training under various hyperparameters is stabilized and gives better performance than MDLM. Due to the limited rebuttal period, we were not able to test all hyperparameter choices under the 1024/24/16 setting. However, we can still confirm that LoMDM with $(c_1,c_2)=(0.7,0.65)$ performs substantially better than MDLM in that setting as well.

---

> > ### Author Rebuttal · Reviewer_54z1 · 2026-04-03
> >
> > Thanks for the authors’ feedback.Some of my concerns have been resolved, but the cross-architecture behavior of smaller c1 values remains unclear.
> >
> > The authors note that they "did observe that using a smaller value leads to a marginal improvement, but such more aggressive choices might be less robust across different datasets". While this explanation is reasonable, it remains essentially an empirical speculation rather than direct evidence. Accordingly, I will retain my original overall score.

---

> > > ### Author Response · Authors · 2026-04-06
> > >
> > > We thank the reviewer for the positive assessment of our paper!
> > >
> > > We understand the reviewer’s concern regarding $c_1$, but we hope the reviewer will consider that both the experiments provided in Q3 and our existing results across multiple datasets suggest that the proposed LoMDM and its hyperparameters work robustly in practice. We will continue to pursue more reliable learnable-order MDMs in future work.

---

### Official Review · Reviewer_X7Et · 2026-03-14

**Soundness:** 3
**Presentation:** 3
**Significance:** 2
**Originality:** 3
**Overall Recommendation:** 4
**Confidence:** 4

**Summary:**

This paper studies how to incorporate generation order into masked diffusion language models. The key contribution is a unified framework, OeMDM, which generalizes standard masked diffusion by allowing position-dependent and context-dependent schedulers in both the forward and reverse processes. Under this formulation, the paper derives a generalized NELBO that decomposes into a reconstruction term and a velocity-matching term, thereby giving generation order a principled probabilistic interpretation rather than treating it as a heuristic decoding strategy. Building on this framework, the paper proposes LoMDM, which jointly learns the diffusion backbone together with learnable forward and reverse schedulers under a single objective. Empirically, the method achieves strong perplexity results across several language modeling benchmarks and demonstrates consistent improvements over prior discrete diffusion baselines.

**Compliance With Llm Reviewing Policy:**

Affirmed.

**Final Justification:**

The author has addressed my concerns, so I will keep my original score.

**Key Questions For Authors:**

1. In the parametrization of the reverse process, although I agree that the true reverse process suffers from a lack of access to global context, it would make the proposed reverse-process design more promising if the paper included additional experimental results on a small-scale setting.
2. How large is the gap in training cost between LoMDM and MDLM? A discussion of the trade-off between performance and cost would be appreciated.

**Limitations:**

yes

**Strengths And Weaknesses:**

Strengths:
1. I find the central idea of the paper both clean and meaningful: instead of treating generation order as an external or inference-only design choice, the paper incorporates it directly into the probabilistic formulation of masked diffusion.
2. The generalized NELBO is not merely a formal derivation; it plays an important role in explaining how generation order affects training.
3. The language modeling experiments are convincing overall. LoMDM consistently improves over prior discrete diffusion baselines on in-domain perplexity, zero-shot perplexity, and generative perplexity, which provides solid evidence that the learned order is beneficial in practice.
4. The paper is well structured, and the progression from MDLM limitations, to OeMDM, to LoMDM is easy to follow.

Weaknesses:
1. A particularly elegant aspect of the paper is that it shows OeMDM can approximate ARM, thereby theoretically bringing autoregressive models into the same framework. However, empirically, LoMDM still does not outperform the AR baseline; for example, on OWT, its test perplexity remains worse than that of the Transformer AR model.
2. Small fix suggestion: In the notation part, using \{0, 1\}^{V+1} instead of [0,1]^{V+1} to represent a one-hot vector.

---

> ### Author Rebuttal · Authors · 2026-03-31
>
> We thank the reviewer for the valuable feedback! We conducted all of the requested experiments and will include them in the camera-ready version.
>
> ### **W1.** Although OeMDM can theoretically approximate ARM, LoMDM is still unable to match ARM.
>
> We believe this is mainly a training-efficiency issue. Unlike ARM, which learns language prediction under a fixed L2R order, LoMDM must learn the 1) generation order and 2) language prediction in multiple learned orders. This makes the learning problem slower than ARM. That said, as discussed in our response to your Q2, LoMDM already learns more than 3× faster than MDLM in wall-clock time, bringing MDM one step closer to ARM. We therefore believe that further improvements in training techniques could help LoMDM close the remaining gap.
>
> Besides, LoMDM still has clear value. As noted in the “Generative perplexity” paragraph of Sec. 5.1, **the main advantage of MDM is its parallel decoding**. LoMDM also shares this advantage, and we have shown in Table 3 that LoMDM even outperforms MDLM in the few-step NFE regime.
>
> ### **W2**. Errata
>
> We will revise the cam-ready version accordingly.
>
> ### **Q2**. The trade-off between the training cost and performance of MDLM and LoMDM.
>
> LoMDM runs about 1.6× slower than MDLM in terms of training steps per unit time, **but reaches the same performance level about 3× faster in wall-clock time.** In Figure 5, we have already shown test PPL per wall-clock time for both MDLM and LoMDM. More specifically, as detailed in the rebuttal, when trained on OWT with 8 H100 GPUs, LoMDM reaches in about ~50 hours the performance that MDLM attains only after about ~170 hours of training.  We observe the same trend on other hardware; please refer to BRJQ:W3 for further discussion.
>
> ### **Q1**. While the proposed reverse process is reasonable, additional analysis would strengthen the paper.
>
> We interpret the question as having 2 parts: a) alternative reverse-process ablations and b) analysis of learned reverse behavior. For a), we kindly invite the reviewer to BRJQ:Q3 (different architecture) and p4nG:Q3 (end-to-end gradient flow / Softmax parametrization). There, we discuss alternatives and why they underperform our parametrization, which indirectly supports the current design.
>
> In this response, we focus on **b), we analyze the generation trajectories of LoMDM** and examine whether the learned reverse behavior exhibits meaningful patterns. We generated 256 length-128 sentences over 128 steps and analyzed them by part of speech (POS).
>
> **Finding:** LoMDM first generates structure and later fills in semantic content.
>
> **Trajectory Visualization**: https://anonymous.4open.science/r/OeMDM (ICML allows a link to figures.)
>
> - **Exp 1. Generation Trajectory of MDLM and LoMDM)**
> We measured the proportion of newly generated tokens in each POS category at each step:
>
>
>     |  | Steps | Nouns | Verbs | Articles | Prepositions |
>     | -- | -- | -- | -- | -- | -- |
>     | MDLM | $1\rightarrow16$ | 0.30 | 0.18 | 0.10 | 0.15 |
>     |  | $112\rightarrow128$ | 0.29 | 0.18 | 0.09 | 0.14 |
>     | LoMDM | $1\rightarrow16$ | 0.25 | 0.15 | 0.14 | 0.19 |
>     |  | $112\rightarrow128$ | 0.37 | 0.20 | 0.07 | 0.10 |
>
>     MDLM’s generation pattern remains similar throughout the entire process. In contrast, LoMDM tends to generate **structures (prepositions, articles) earlier, and more semantic content (nouns and verbs) later.** Our conjecture is that, when the sentence is almost fully masked, LoMDM first learns to establish a reliable sentence structure that is less likely to be wrong, thereby increasing the available context, and then fills in the more semantic content afterward.
>
> - **Exp 2. Masks that LoMDM focuses on)**
> We measured relative generation priority for each mask token, grouped by the POS of the revealed anchor token and the distance from that anchor. To quantify this, we report the mean Z-score of $A_\psi$ for masked positions at distance $d$ from the anchor token:
>     | Anchor POS | $d=1$ | $d=2$ | $d=3$ |
>     | --- | ---: | ---: | ---: |
>     | Noun | 1.79 | 1.00 | 0.47 |
>     | Verb | 0.29 | 0.43 | 0.05 |
>     | Article | -2.56 | -1.13 | 0.10 |
>     | Preposition | -2.47 | -1.45 | -0.10 |
>
>     From the results, we observe that the generation priority is substantially higher near semantic words. We interpret this in 2 ways: 1) Positions near articles or prepositions may be avoided because those positions often require more semantic content to be determined, whereas positions near nouns or verbs may be preferred because structural words are more naturally generated around them. 2) Having richer surrounding information makes a position easier to predict, and we conjecture that LoMDM learns to favor such paths.
>
>
> Findings 1 and 2 suggest that LoMDM is indeed a learnable-order MDM that generates tokens by considering the current context. We will include these analyses in the camera-ready version.

---

> > ### Author Rebuttal · Reviewer_X7Et · 2026-04-03
> >
> > The author has addressed my concerns, so I will keep my original score.

---

> > > ### Author Response · Authors · 2026-04-06
> > >
> > > We are glad that the reviewer’s concerns have been fully addressed! We hope that our method will inspire further work within the MDM framework.

---

### Decision · Program_Chairs · 2026-04-30

**Decision:**

Accept (spotlight)

**Comment:**

This paper proposes OeMDM, a unified masked diffusion framework that can express various generation orders, and LoMDM, which jointly learns the generation order and the diffusion model.

During rebuttal, the authors addressed the main concerns by providing additional experiments on training efficiency, scaling behavior, and learned generation order, as well as clarifying comparisons to related work such as LO-ARM. Following these clarifications, reviewers indicated that their concerns were largely resolved and maintained or raised their scores to weak accept or accept. Therefore, I recommend acceptance.

Based on my own reading, I find the paper presents a solid and innovative approach to learning generation order through training. In particular, it provides a principled way to better exploit the flexibility of generation order in discrete diffusion models during training. I believe this perspective is both technically insightful and likely to inspire further research in this direction.

I would also like to kindly point out two recent ICLR papers on learning generation order. These belong to post-training approaches, and therefore do not diminish the value of this work at all. I mention them as they are relevant but not discussed in the paper (they appeared online only a few months before the deadline, so it is understandable that the authors may not have been aware of them):

https://openreview.net/forum?id=rrD1U0Izt5

https://openreview.net/forum?id=XjcHRIu0iF